# Evaluation of the uncertainty of the spectral UV irradiance measured by double- and single-monochromator Brewer spectrophotometers

Carmen González<sup>1,2</sup>, José M. Vilaplana<sup>1</sup>, Alberto Redondas<sup>3</sup>, Javier López-Solano<sup>4</sup>, José M. San Atanasio<sup>5</sup>, Richard Kift<sup>6</sup>, Andrew R. D. Smedley<sup>6</sup>, Pavel Babal<sup>7</sup>, Ana Díaz<sup>5</sup>, Nis Jepsen<sup>8</sup>, Guisella Gacitúa<sup>8</sup>, and Antonio Serrano<sup>2</sup>

Correspondence to: Carmen González (cgonher@inta.es)

Abstract. Brewer instruments are robust, widely used instruments that have been monitoring global solar ultraviolet (UV) irradiance since the 1990s, playing a key role in UV research. Unfortunately, the uncertainties of these measurements are rarely evaluated due to the difficulties involved in the uncertainty propagation. This evaluation is essential to determine the quality of the measurements as well as their comparability to other measurements. In this study, eight double- and two single-monochromator Brewers are characterised and the uncertainty of their global UV measurements is estimated using the Monte Carlo method. This methodology is selected as it provides reliable uncertainty estimations and considers the nonlinearity of certain steps in the UV processing algorithm. The combined standard uncertainty depends on the Brewer, varying between 2.5 % and 4 % for the 300–350 nm region. For wavelengths below 300 nm, the differences between single-and double-monochromator Brewers increase, due to the uncertainty in stray light correction. For example, at 295 nm, the relative uncertainties of single Brewers range between 11–21 % while double Brewers have uncertainties of 4–7 %. These uncertainties arise primarily from radiometric stability, the application of cosine correction, and the uncertainty of the lamp used during calibration. As the measured UV irradiance decreases, the correction of dark signal, stray light (for single Brewers), and noise become the dominant sources of uncertainty. These results indicate that the overall uncertainty of a Brewer spectrophotometer could be reduced by increasing the frequency of radiometric calibration and improving the traditional entrance optics.

<sup>&</sup>lt;sup>1</sup>Área de Investigación e Instrumentación Atmosférica, Instituto Nacional de Técnica Aeroespacial (INTA), El Arenosillo, Huelva, Spain

<sup>&</sup>lt;sup>2</sup>Departamento de Física, Instituto del Agua, Cambio Climático y Sostenibilidad, Facultad de Ciencias, Universidad de Extremadura, Badajoz, Spain

<sup>&</sup>lt;sup>3</sup>Izaña Atmospheric Research Center, Agencia Estatal de Meteorología, Tenerife, Spain

<sup>&</sup>lt;sup>4</sup>Departamento de Ingeniería Industrial, Universidad de La Laguna, La Laguna, Tenerife, Spain

<sup>&</sup>lt;sup>5</sup>Servicio de Redes Especiales, Agencia Estatal de Meteorología, Madrid, Spain

<sup>&</sup>lt;sup>6</sup>Department of Earth and Environmental Sciences, University of Manchester, Manchester, UK

<sup>&</sup>lt;sup>7</sup>OTT Hydromet, Delftechpark 36, 2628XH, Delft, The Netherlands

<sup>&</sup>lt;sup>8</sup>National Centre for Climate Research, Danish Meteorological Institute, Copenhagen, Denmark

# 1 Introduction

60

Brewer spectrophotometers (Brewer, 1973; Kerr, 2010) were initially developed in the 1970s for supplementing the ozone measurements of the Dobson spectrophotometer (Dobson, 1931). These first Brewers were single-monochromator spectrometers and are usually referred to as single Brewers. Towards the 1980s, they were modified to also measure solar ultraviolet (UV) irradiance (Bais et al., 1996). During this time, double-monochromator Brewers (also known as double Brewers) were developed to reduce the stray light in UV measurements. Thanks to the robustness and stability of Brewer spectrophotometers, the early instruments continue in operation and the Brewer network has steadily increased in number. Today, more than 200 Brewers are deployed worldwide providing measurements of total ozone column (TOC), global UV irradiance, sulphur dioxide, aerosol optical depth in the UV, and nitrogen dioxide. All these measurements contribute to a better understanding of long-term UV variations (Fountoulakis et al., 2016a; Simic et al., 2008; Smedley et al., 2012; Zerefos et al., 2012) and the dynamics of the Earth's atmosphere (Arola et al., 2003; Estupiñán et al., 1996; Fioletov et al., 1998). Furthermore, Brewer spectral UV data have also been used to monitor the increase in surface UV-B levels due to the depletion of stratospheric ozone (Fioletov et al., 2001; Kerr and McElroy, 1993; Lakkala et al., 2003). Therefore, Brewer spectrophotometers have greatly contributed to the study of solar UV for more than 30 years.

To ensure the quality of Brewer UV measurements, quality assurance (QA) and quality control (QC) procedures must be performed. QC evaluates the uncertainty of the measurement by: (a) identifying the error sources, (b) determining the model that relates these sources with the output quantity (i.e. the UV irradiance), and (c) propagating the uncertainty using a numeric or analytical approach (e.g. Garane et al., 2006; González et al., 2023, 2024b; Webb et al., 1998). This is essential to determine the quality of the measurement and ensure its comparability to other measurements (BIPM et al., 2008a). On the other hand, QA can be performed using two methods (Webb et al., 2003). In the first one (inductive), the instrument's performance is assessed through intercomparison campaigns. As for the second (deductive) method, the user deduces the instrument's quality through a meticulous description of the calibration process as well as the instrumental characteristics, such as its linearity and angular response. For OA purposes, the general principles established by Webb et al. (1998) should be followed, expanded, and refined, so the user can report reliable uncertainties for any measurement, not limiting the analysis for a typical measurement at the station (Webb et al., 2003). Unfortunately, for Brewer measurements, there is no consensus on how QA/QC should be performed and as a result, the data processing, uncertainty evaluation, and calibration practices vary from station to station. In this context, EuBrewNet (European Brewer Network), originally developed through COST Action 1207 and currently operational thanks to AEMET (Spanish State Meteorological Agency), is working on harmonising and developing coherent practices for Brewer QA/QC (Redondas et al., 2018; Rimmer et al., 2018). All Brewers used in this study are part of EuBrewNet and as a result, their calibration and UV measurements are obtained following their protocols.

The QA performed for the instruments used in this work corresponds to the inductive method described by Webb et al., (2003). It is carried out during the campaigns performed by the Regional Brewer Calibration Center–Europe (RBCC-E)

where Brewer spectrophotometers are compared to the European reference spectroradiometer, the QASUME unit (e.g. Gröbner et al., 2010; Lakkala et al., 2008). These intercomparison campaigns meet the main requirements laid out by Webb et al. (2003), i.e. transparency and objective comparison algorithms (see the campaign reports at the PMOD/WRC website, <a href="https://www.pmodwrc.ch/en/world-radiation-center-2/wcc-uv/qasume-site-audits/">https://www.pmodwrc.ch/en/world-radiation-center-2/wcc-uv/qasume-site-audits/</a>, the report of the 18th intercomparison campaign, Hülsen, 2023, and an overview of the EuBrewNet's algorithms, López-Solano, 2024). This QA procedure is not currently implemented in terms of uncertainty estimates. This is not surprising as QC is a pending task for the Brewer network. In fact, it remains one of the main challenges for Brewer sites measuring ozone (Fioletov et al., 2008) and UV irradiance. Although the main errors and uncertainties affecting spectral UV irradiance measurements are well-known (Bais, 1997; Bernhard and Seckmeyer, 1999; Webb et al., 1998), their proper characterisation is difficult and requires specialised equipment (such as tuneable lasers, portable unit systems, or devices to measure angular response) that is not available to most UV site operators. Furthermore, the calculation of the uncertainty propagation is complicated due to the nonlinearity of the UV irradiance model.

Brewer sites performing QC for UV measurements usually approach the uncertainty propagation following the recommendations of the Guide to the Expression of Uncertainty in Measurement (GUM) (BIPM et al., 2008a). In particular, the GUM uncertainty framework (hereafter "GUF") is applied by assuming that the UV irradiance model is linear (e.g. Garane et al., 2006). Although this assumption is valid for Brewer spectrophotometers, the GUF provides less accurate estimations than other uncertainty propagation techniques (González et al., 2024b), such as the Monte Carlo method (hereafter "MCM") and the Unscented transformation (hereafter "UT"). The UT is an efficient technique that evaluates the uncertainty by applying the nonlinear model to a reduced set of points, referred to as sigma points (Julier et al., 1995; Julier and Uhlmann, 1997). These sigma points are carefully chosen using several parameters to ensure their statistics (first and second order) match those of the measurand. However, if these points are not selected properly, the UT fails to obtain a correct estimate and its corresponding uncertainty. While recommended values usually work, they may not be optimal in some cases (Straka et al., 2012; Turner and Rasmussen, 2010; Wang and Ding, 2020). Although the UT method provided accurate results for a specific double Brewer (González et al., 2024b), it is unclear whether those results can be extended to single Brewers and other double Brewers. Therefore, in this work the MCM was selected as it has a broader range of validity than the UT and GUF, provided that a sufficient number of simulations are performed (usually 10<sup>6</sup> iterations) (BIPM et al., 2008b).

The original contribution of this article is the characterisation and uncertainty evaluation of ten single and double Brewer spectrophotometers (MkIV and MkIII type, respectively) using the methodology set by González et al. (2023). In this way, the MCM is implemented to evaluate the uncertainty of the Brewer UV network. All the necessary uncertainty sources considered by Webb et al. (1998) have been included in the uncertainty evaluation presented in this work, plus some highly recommended and additional sources (such as stray light, alignment, or wavelength accuracy). Moreover, a sensitivity analysis has also been performed to examine the influence of every uncertainty source on each Brewer spectrophotometer.


The UV scans used for the uncertainty analysis were recorded during the XVIII RBCC-E intercomparison campaign at the El Arenosillo Observatory (37.1° N, 6.7° W, 41 m a.s.l., Huelva, Spain).

The paper is organised as follows. First, the specifications of single and double Brewer spectrophotometers as well as an overview of the intercomparison campaign are given in Section 2. Next, Section 3 describes the uncertainty evaluation, i.e. the characterisation of the uncertainty sources, the UV irradiance model, and the MCM approach. Then, in Section 4 the results obtained from the sensitivity and uncertainty analysis are discussed. Finally, Section 5 summarises the main conclusions of this study.

# 2 Instrumentation and data






# 2.1 Brewer spectrophotometers

In this study, the uncertainty of the spectral UV measurements performed by ten Brewer instruments have been evaluated. Two different types of Brewer spectrophotometers have been considered, MkIV (single monochromator) and MkIII (double monochromator). The serial number and information of each instrument is shown in Table 1.

The optical path is similar for both Brewer types, i.e. global UV irradiance enters through the entrance optics, consisting of a Teflon diffuser covered by a quartz dome, and is redirected into the fore-optics using two prisms (UV-B and zenith prisms). The incoming radiation is then focused and collimated by the Iris diaphragm. Then, the intensity of the beam is adjusted before entering the spectrometer, using two filter wheels. The first filter wheel has an open hole (open position) for UV measurements, a ground-quartz disk (pos. 1) for direct-Sun measurements and an opaque disk (pos. 2) for dark signal tests (Kipp & Zonen, 2007, 2018). The second filter wheel contains five neutral density filters to adjust the intensity level of the incoming light. After passing the filter wheels, the light is focused onto the spectrometer. This spectrometer is a modified Ebert grating type that disperses the light into a spectrum using a diffraction grating. At the exit of the spectrometer, there is a cylindrical slit mask. For ozone, dead time, and dark signal observations, the diffraction grating is fixed while the slit mask rotates, selecting in this way the wavelength. On the other hand, for the measuring of UV irradiance, the slit mask remains fixed, and the diffraction grating rotates (using a micrometre) to select the wavelength. In MkIV Brewers, the emerging light passes through a third filter wheel, which has several filters to block undesired radiation: (1) in the ozone mode, a UG-11/NiSO4 filter combination is used, (2) in the UV mode, the filter switches to a UG-11 filter, and (3) in the NO2 mode, a BG-12 filter is used to block UV radiation (Kipp & Zonen, 2007). Finally, the photons are detected by a photomultiplier tube (PMT). A diagram of the MkIV and MkIII Brewers can be found in Kerr (2010) and González et al. (2023), respectively.

Table 1. Information for each Brewer spectrophotometer used in this study.

| Brewer | Type (monochromator) | Entrance optics (diffuser) | Institute (Country)                                |
|--------|----------------------|----------------------------|----------------------------------------------------|
| #117   | MkIV (single)        | Traditional (flat)         | State Meteorological Agency – AEMET (Spain)        |
| #150   | MkIII (double)       | CMS-Schreder (shaped)      | National Institute of Aerospace Technology (Spain) |

| #151 | MkIV (single)  | Traditional (flat)  | State Meteorological Agency – AEMET (Spain)      |
|------|----------------|---------------------|--------------------------------------------------|
| #158 | MkIII (double) | Traditional (flat)  | OTT Hydromet (The Netherlands)                   |
| #172 | MkIII (double) | Traditional (flat)  | University of Manchester (UK)                    |
| #185 | MkIII (double) | CMS-Schreder (flat) | Izaña Atmospheric Research Center, AEMET (Spain) |
| #186 | MkIII (double) | Traditional (flat)  | State Meteorological Agency – AEMET (Spain)      |
| #202 | MkIII (double) | Traditional (flat)  | Danish Meteorological Institute (Denmark)        |
| #228 | MkIII (double) | Traditional (flat)  | Danish Meteorological Institute (Denmark)        |
| #256 | MkIII (double) | CMS-Schreder (flat) | Izaña Atmospheric Research Center, AEMET (Spain) |

The difference between the two types of Brewers lies in the number of monochromators and the type of diffraction gratings. MkIV Brewers are single monochromators and the diffraction grating of the spectrometer is operated in the third order and has a line density of 1200 lines mm<sup>-1</sup>. On the other hand, the spectrometer system in MkIII Brewers consists of a pair of monochromators and gratings to reduce the stray light of the system. Both diffraction gratings have a line density of 3600 lines mm<sup>-1</sup> and are operated in the first order.

The entrance optics for both types of Brewer spectrophotometer consists of a Teflon diffuser covered by a quartz dome. Traditionally, the diffuser was flat, but the angular response of such an entrance optic can deviate substantially from the ideal cosine response (Bais et al., 2005; Lakkala et al., 2018). Therefore, a correction is needed to mitigate such deviation, as will be described later in Section 3.1.3. On the other hand, new designs have been developed to improve the Brewer angular response (Gröbner, 2003). Brewer #150 has this novel entrance optic developed by CMS-Schreder (model UV-J1015) with a shaped Teflon diffuser. On the other hand, Brewers #185 and #256 have a flat diffuser mounted on the CMS-Schreder optic. The remaining seven Brewers have the standard flat diffuser originally mounted in these spectrometers.

For spectral UV irradiance measurements, the operational wavelength range of the Brewers under study is 290–363 nm with a step of 0.5 nm. The shape of the slit function is trapezoidal and the Full Width at Half Maximum (FWHM) ranges from 0.55 to 0.65 nm, depending on the instrument. Brewer spectrophotometers are operated within a weather-proof housing and have electrical heaters to prevent operation at low temperatures. If the internal temperature of the instrument falls below 10 °C or 20 °C (Kipp & Zonen. 2018), these heaters are automatically switched on. Nevertheless, the Brewer internal temperature is not fully stabilised and can fluctuate throughout the day. As for their calibration, the instrument is calibrated using reference lamps (usually 1000 W lamps) with the input optics positioned at zenith. The calibration and processing algorithms of the Brewer UV measurements used in this study are set by EuBrewNet and are described in the following sections.

#### 2.2 Intercomparison campaign at INTA/El Arenosillo






The data used are the UV scans performed during the 18th Regional Brewer Calibration Center for Europe (RBCC-E) intercomparison held at the El Arenosillo Observatory (37.1° N, 6.7° W, 41 m a.s.l.) in Huelva, Spain, from 5 to 15

September 2023 (WMO, 2023). All instruments were installed on the roof of the station, where the horizon is free up to solar zenith angles (SZAs) of 85°.

The first five days of the campaign, 6–10 September, were dedicated to assessing the status of the participating Brewers (by comparing their ozone measurements with those of the reference, Brewer #185), to performing the necessary maintenance, and, finally, to gathering data for the instruments' calibration. The angular characterisation and the UV comparison were carried out during the final four days of the campaign, 11–14 September. It should be mentioned that the angular response characterisation was performed for five MkIII Brewers (#150, #185, #186, #158, and #256). Four of them (#150, #185, #186, and #256) were characterised using the Brewer Angular Tester (BAT), described later in Section 3.1.3. The remaining Brewer (#158) was characterised in the laboratory of its operating site, using a lamp mounted on an arm that turns by step of 5°.

The protocol to measure global spectral UV irradiance consisted of scanning one UV spectrum from sun rise to sun set every 30 minutes. The wavelength increment and time step were set to 0.5 nm and 3 s respectively. This setting was chosen to obtain simultaneous measurements between the 16 Brewer spectrophotometers (8 MkIII, 6 MkIV, and 2 MkII) participating in the campaign and the travelling reference QASUME (Gröbner et al., 2005; Hülsen et al., 2016) from the Physikalisch-Meteorologisches Observatorium Davos/World Radiation Center (PMOD/WRC). The results from this intercomparison, along with other QASUME site audits, are published on the web page of PMOD/WRC (https://www.pmodwrc.ch/en/world-radiation-center-2/wcc-uv/gasume-site-audits/, last access: 2 February 2025).

In this work, the UV irradiances and the corresponding uncertainties of MkIII and MkIV Brewers (see Table 1) have been calculated using their raw UV signal, calibration files, slit functions, dead time, dark signal, reference lamp certificates, and angular response measurements. This information is available at EuBrewNet (Rimmer et al., 2018) except for the lamp certificates and some calibration records, which were provided by the Brewer and QASUME operators. In the following section, the UV processing algorithm and the uncertainty propagation techniques implemented are described.

#### 175 **3 Methodology**



The combined standard uncertainty of the global UV irradiance measured by MkIII and MkIV Brewers has been calculated following the guidelines established by the Guide to the expression of Uncertainty in Measurement (GUM). In particular, the uncertainty analysis has been carried out using a numerical propagation technique, the MCM (BIPM et al., 2008b). The UV measurements used for the uncertainty evaluation were performed during the 18th RBCC-E intercomparison campaign (described in detail in Section 2.2.). Furthermore, a sensitivity analysis has also been performed to identify the main sources of uncertainty in Brewer UV measurement procedure. To carry out the uncertainty evaluation, the uncertainty sources and the model relating these sources to the measured irradiance must first be characterised. In the absence of standard procedures for this task, guidelines established EuBrewNet followed the by have been (https://eubrewnet.aemet.es/dokuwiki/doku.php?id=codes:uvaccess, last access: 20 May 2025). An overview of these guidelines can also be found in López-Solano et al. (2024). The uncertainty sources and irradiance model are described in the following subsections.

#### 3.1 Characterisation of the uncertainty sources







The spectral irradiance measured by a Brewer spectrophotometer is affected by several error sources that need to be corrected (e.g. Kerr, 2010; Lakkala et al., 2008). Error sources are usually separated into random and systematic components. Random errors produce variations in repeated measurements and as such, are usually reduced by increasing the number of observations (BIPM et al., 2008a). On the other hand, systematic errors can be compensated by applying a correction factor to the irradiance measured. Even if all errors are appropriately characterised and corrected, there still remains a doubt, an uncertainty, about the accuracy of the reported result (BIPM et al., 2008b). In the following, the term "error" will denote the imperfections in a measurement result, while the term "uncertainty" will be used to reflect the existing doubt regarding the value of the measured spectral UV irradiance. In this way, it is important to differentiate between the source of error (for example cosine error) and the uncertainty associated with its correction.

In this study, the main uncertainty sources in solar radiometry have been considered (Webb et al., 1998). They have been characterised following the methodologies of Bernhard and Seckmeyer (1999) and González et al. (2023, 2024b). It should be noted that some of the uncertainties (such as those related to noise, stray light correction, or radiometric stability) have not been determined thoroughly, as the data used for their estimation are insufficient to obtain appropriate statistics. Nevertheless, in all these cases, uncertainty values have been given and included in the Monte Carlo simulation. The uncertainty sources affecting the Brewer UV measurement procedure can be divided into three groups, depending on whether they affect (1) the signal measured by the instrument (see Section 3.1.1.), (2) the radiometric calibration (see Section 3.1.2.), or (3) the uncorrected absolute irradiance (see Section 3.1.3.). All these sources and their corresponding corrections have been applied to the Brewer UV measurements during the 18th RBCC-E campaign, whenever possible. As will be described in the following, some of the participating Brewers were not fully characterised, lacking information regarding their temperature and cosine correction. The uncertainty sources included in the uncertainty evaluation of each Brewer are summarised in Table 2.

Table 2. Summary of the uncertainty sources considered for each Brewer under study. Red squares (–) represent the uncertainty sources not included in the evaluation, while green squares (×) indicate those uncertainty sources considered.

| Uncertainty sources considered |      | Brewer ID |      |      |      |      |      |      |      |      |  |
|--------------------------------|------|-----------|------|------|------|------|------|------|------|------|--|
| Oncertainty sources considered | #117 | #150      | #151 | #158 | #172 | #185 | #186 | #202 | #228 | #256 |  |
| Noise                          | ×    | ×         | ×    | ×    | ×    | ×    | ×    | ×    | ×    | ×    |  |
| Dark signal                    | ×    | ×         | ×    | ×    | ×    | ×    | ×    | ×    | ×    | ×    |  |
| Stray light                    | ×    | _         | ×    | _    | -    | _    | -    | -    | -    | _    |  |
| Dead time                      | ×    | ×         | ×    | ×    | ×    | ×    | ×    | ×    | ×    | ×    |  |

| Distance adjustment               | × | × | × | × | × | × | × | × | × | × |
|-----------------------------------|---|---|---|---|---|---|---|---|---|---|
| Uncertainty of the reference lamp | × | × | × | × | × | × | × | × | × | × |
| Radiometric stability             | × | × | × | × | × | × | × | × | × | × |
| Wavelength shift                  | × | × | × | × | × | × | × | × | × | × |
| Temperature correction            | _ | × | _ | _ | _ | _ | _ | _ | _ | - |
| Cosine correction                 | - | × | - | × | - | × | × | - | - | × |

# 3.1.1 Brewer signal






The signal measured by any Brewer spectrometer are affected by stray light, noise, dark signal, and dead time.

Stray light is the radiation coming from wavelengths other than the one being measured. This undesired light is able to reach the detector due to scatter inside the instrument and dust particles. The presence of stray light is larger in single Brewers (such as MkIV Brewers) and results in an overestimation of the measured global UV irradiance at short wavelengths (Bais et al., 1996; Davies, 1996). Therefore, EuBrewNet applies a correction only for single Brewers, as the stray light present in double Brewers is very low (Bais et al., 1996; Karppinen et al., 2015; Savastiouk et al., 2023). While this correction is implemented for ozone measurements (Redondas et al., 2018), for UV measurements further characterisation is needed. It is usual to estimate stray light as the average signal recorded below 292 nm (e.g. Arola and Koskela, 2004; Lakkala et al., 2008; Mäkelä et al., 2016) and the correction is carried out by subtracting this average value from the signal measured at all wavelengths. The uncertainty of this method was estimated by comparing the corrected irradiance to the QASUME from 290 to 292 nm. This estimation also includes the effects of temperature and cosine errors since the single Brewers under study are not corrected for these two sources of error. Nevertheless, since the effect of these two sources is expected to be small below 292 nm, the uncertainty determined might be only a slight overestimation. Furthermore, the standard deviation from the measurements of the five wavelengths (from 290 to 292 nm) was also derived and combined with the uncertainty obtained from the QASUME comparison.

Noise can be characterised by studying the signal-to-noise ratio (SNR) (e.g. Bernhard and Seckmeyer, 1999; Cordero et al., 2012). However, this ratio can only be determined if all conditions, such as the incoming radiation, remain stable. For CCD-array spectroradiometers, this is easy as the instrument can record several spectra in a few seconds. In contrast, Brewer spectrophotometers take approximately 5–6 minutes to measure a single UV spectrum. As a result, characterising the noise in Brewer measurements is only straightforward during the radiometric calibration, when the emitting lamp is stable. During the RBCC-E campaign, the measurement of the irradiance of the reference lamp was acquired four times. With this information, the SNR for the radiometric calibration is calculated. Since it was proportional to the root of the raw signal recorded, the methodology proposed by Bernhard and Seckmeyer (1999) could be applied and the SNR for the signal measured under the Sun was obtained as:

$$SNR^{S}(\lambda) = SNR^{R}(\lambda) \sqrt{\frac{c_{0}^{S}(\lambda)}{c_{n}^{R}(\lambda)}},$$
(1)

where  $C_0^S(\lambda)$  and  $C_0^R(\lambda)$  are the raw signal measured under the Sun and lamp (calibration), respectively, and  $SNR^S(\lambda)$  and  $SNR^R(\lambda)$  are the signal-to-noise of the outdoor and calibration measurements, respectively.

Dark signal is the counts recorded when no light is entering the instrument. Brewer spectrophotometers are programmed to measure the dark signal before every observation (UV scan, direct-Sun measurement, etc.) by blocking the incoming radiation. The dark signal of the Brewers under study increased as the internal temperature of the instrument rose. To estimate the uncertainty of the dark signal, the dark signal measured at each temperature reached by the instrument were averaged and the corresponding standard uncertainty was calculated. The number of available measurements depended on the instrument, but, in total, more than 2500 dark signal measurements were recorded by each Brewer during the intercomparison campaign.

Dead time is the time after a photon has been recorded during which the photomultiplier tube (PMT) is unable to detect a second one. This causes the nonlinearity in Brewer response as any photons arriving during the dead time of the PMT are not taken into account. Similarly to dark signal, dead time determination is also included in the schedule of Brewer spectrophotometers (DT test). Each instrument records it daily by measuring and comparing high and low intensities of an internal quartz-halogen 20 W lamp (standard lamp). Initially, the dead time of the instrument is stored in the B-files (instrument constants). Then, using the DT tests, this constant is frequently checked and updated during calibration audits. The dead time is characteristic of each Brewer as it depends on the type of PMT used. For the Brewers under study, the dead time varied from 25 to 36 ns and their uncertainties ranged from 0.3 to 3 ns. These values were derived from the instrument constant files and the DT tests carried out during the campaign. The dead time uncertainties found using the previous methodology are similar to those reported by Fountoulakis et al. (2016b). They determined standard deviations of 1–2 ns for the Brewer dead time using direct-Sun measurements. Moreover, the uncertainties estimated for dead time also agree with the ones applied in other uncertainty evaluation studies for Brewer spectrophotometers (Diémoz et al., 2014).

# 3.1.2 Radiometric calibration





Brewer spectrophotometerss were calibrated during the campaign, using one or two reference DXW-1000 W tungstenfilament incandescent halogen lamps. These lamps had been previously calibrated in the laboratories of PMOD/WRC and the Finnish Metrology Research Institute, belonging to Aalto University and MIKES. The calibrations performed by PMOD/WRC are traceable to the primary standard of the Physikalisch-Technische Bundesanstalt (PTB) (Gröbner and Sperfeld, 2005). On the other hand, the Metrology Research Institute is the national standard laboratory for optical quantities in Finland and is part of the CIPM Mutual Recognition Arrangement (CIPM MRA), a framework in which metrology institutes prove the international equivalence of their calibrations and certificates. These calibrations guarantee that the spectral irradiances of the lamps are the ones stated in their calibration certificate when operated at the specified distance and electrical current. To ensure the latter, the radiometric calibration was performed with a mobile system that places the lamp on top of the Brewer diffuser at the required distance. This setup also stabilised and monitored the electrical current at its nominal value using a multimeter. Then, the signal under the lamp is recorded several times and corrected for dark counts, dead time, and stray light (see Section 3.1.1.). The responsivity of the instrument is derived by dividing the corrected signal by the irradiance of the reference lamp. However, this responsivity is also affected by other sources of uncertainty produced during the radiometric calibration such as the distance adjustment between the lamp and the diffuser, the radiometric stability, and the uncertainty of the spectral irradiance emitted by the reference lamp. Small fluctuations in the electrical current of the lamp can also produce errors in the calibration. Based on the findings of Webb et al. (1994), the standard practice is to assume that a 1 % change in the current of the reference lamp leads to a 10 % change in the spectral irradiance measured by the instrument (e.g. Bernhard and Seckmeyer, 1999; Webb et al., 1998). According to the previous rule, the expected change in the irradiance of the Brewers under study would be of 0.125 %, as the electrical current was stabilised to within 0.0125 % during their calibration. However, this source could not be included in the Monte Carlo simulation, as it requires the determination of the lamp's operating temperature (Schinke et al., 2020) and this could not be carried out during the campaign.

In the RBCC-E campaigns hosted at the El Arenosillo, the distance between the plane of the reference lamp and the Brewer's diffuser is adjusted using a ruler with a precision of 1 mm. According to the GUM, this precision error translates in an uncertainty of 0.58 mm (BIPM et al., 2008a). Since 1000 W lamps are usually placed at 50 cm, for most Brewer spectrophotometers, the lamps were set at  $(500.0 \pm 0.6)$  mm. The exceptions are Brewers #158 and #150, their lamps were set at  $(412.5 \pm 0.6)$  mm and  $(500.0 \pm 0.6)$  mm, respectively. The reference lamp used to calibrate Brewer #158 needed to be placed at 412.5 mm, as indicated in its calibration certificate. Brewer #150, on the other hand, has an additional source of uncertainty since the position of its diffuser's reference plane needs to be determined as well (González et al., 2023), resulting in an uncertainty of 0.59 mm. This plane determination was carried out by placing an ultrastabilised lamp at several distances and measuring its emitted spectrum. The data showed that the diffuser's reference plane is placed  $(0.234 \pm 0.015)$  cm below the reference used for calibration, i.e. the metalling ring of the quartz dome covering the Brewer's diffuser (a schematic drawing of this reference can be found in González et al. (2023)).

Regarding the reference lamps, their emitted irradiances and their corresponding expanded uncertainties (coverage factor of k = 2) are specified in their calibration certificates. These uncertainties depend on the lamp and the wavelength, slightly decreasing from 2–3 % at 290 nm to 1–2 % at 360 nm.

Even if a Brewer spectrophotometer is regularly calibrated, the responsivity of the instrument varies with time. This drift is caused by several factors such as transportation, storage, and ageing of the instrument as well as the instability of the photomultiplier tube (PMT). To characterise the radiometric stability of every Brewer, several studies recommend studying the difference between consecutive calibration factors over a significant period (e.g. a year) (Bernhard and Seckmeyer, 1999; Webb et al., 1998). These methods require that the instrument is calibrated frequently to derive reliable statistics. Unfortunately, not all Brewers studied had enough calibration files. Some of them had been operating for less than 2 years,

others had undergone several modifications that greatly affected their responsivity (such as replacement of the PMT or change of entrance optics), and the rest were not calibrated frequently enough. As a result, Brewers #150 and #185 were the only ones characterised following the methodology of Bernhard and Seckmeyer (1999), i.e. by deriving the standard deviation of the difference between consecutive calibrations. For the remaining instruments, a 3 % uncertainty was assumed as several Brewer spectrophotometers (both double and single) exhibit uncertainties of that order (Garane et al., 2006; Lakkala et al., 2008). It should be noted that this value, derived from long-term monitoring, might not be a large overestimation for the newly calibrated Brewer (#256), as this instrument showed large instabilities during its first year of operation. On the other hand, the uncertainty assumed (3 %) is also similar to the average uncertainty found for Brewers #150 and #185, of 2.9 % and 2.4 %, respectively. For Brewer #150, the radiometric uncertainty was derived using the yearly calibration files from 2005 to 2023, while for Brewer #185 the uncertainty was calculated using the monthly calibration files recorded from 2021 to 2024. As mentioned earlier, no data from prior years could be used as the entrance optics of Brewers #150 and #185 were replaced in 2005 and 2021, respectively.

#### 3.1.3 Uncorrected absolute irradiance






Even if error sources affecting the Brewer signal and the radiometric calibration are characterised (as indicated in the sections above), there are still some uncertainty sources affecting the UV irradiance. Specifically, wavelength misalignment, temperature dependence, and cosine error.

Wavelength misalignment refers to a mismatch between the wavelength desired and the one being measured. It is usually corrected by determining the wavelength shift for every wavelength measured. Although this shift is small for Brewer spectrophotometers, it still leads to important errors, especially in the UV-B (280–315 nm) due to the marked variability in the solar irradiance at this region. For instance, a shift of less than 0.05 nm can produce an uncertainty in the UV irradiance of a few percent for wavelengths below 305 nm (Bais, 1997, González et al., 2024b).

Wavelength shifts can be determined using specific software like SHICrivm (Slaper et al., 1995) or MatSHIC (Hülsen et al., 2016). There are other options (as explained by Bernhard and Seckmeyer (1999)), but this one is the most straightforward and these detection algorithms also derive a shift for every wavelength and irradiance level measured. In this study, the wavelength shifts were obtained using SHICrivm as it is the algorithm implemented in EuBrewNet. This software estimates the wavelength shift by comparing the structure of the spectrum measured by the ground-based instrument with the extraterrestrial spectrum. The latter is simulated using the SUSIM Extraterrestrial spectrum (Slaper et al., 1995). Therefore, the shifts determined by SHICrivm include the wavelength misalignment produced by the precision of the micrometre, i.e. the system setting the wavelengths measured by a Brewer spectrophotometer. This precision is approximately 8 pm (Gröbner et al., 1998). These shifts were also estimated during the RBCC-E campaign and can be checked in the report elaborated by the PMOD/WRC (Hülsen, 2023).

Brewer spectrophotometers, no matter the type, have no temperature stabilisation system. Thus, the internal temperature fluctuates throughout the day, resulting in a temperature dependency in Brewer global UV measurements. This fact is well-

documented (Fountoulakis et al., 2017; Garane et al., 2006; Lakkala et al., 2008; Weatherhead et al., 2001), but there is no standard methodology for its characterisation. Therefore, EuBrewNet lets the Brewer operators characterise this source in the way they see fit. As a recommendation, the work of Lakkala et al. (2008) is indicated.

No temperature characterisation was performed during the campaign. As a result, this uncertainty source was only included in the uncertainty evaluation of Brewer #150. This instrument was characterised on three separate days in 2022, using 100 and 1000 W lamps. The instrument temperature increased gradually from 23 to 38 °C while it measured the irradiance emitted by the lamps. Then, the relationship between the internal temperature of Brewer #150 and its change in responsivity with respect to a reference value (31 °C) was studied. The results showed that the instrument's responsivity decreases linearly with temperature, as:



$$r(\lambda, T) = r(\lambda, T_{\text{ref}})[1 + c_{\text{T}}(T - T_{\text{ref}})], \qquad (2)$$

where  $r(\lambda, T)$  is the responsivity measured at wavelength  $\lambda$  and internal temperature T,  $r(\lambda, T_{ref})$  is the responsivity measured at the reference temperature  $T_{ref} = 31$  °C, and  $c_T$  is the slope of the linear fit. The latter is the temperature correction factor and for Brewer #150 it has a value of  $c_T = (-0.0016 \pm 0.0002)$  °C<sup>-1</sup> (González et al., 2023). Therefore, the UV measurements of Brewer #150 were corrected for temperature by considering this correction factor and the difference between the temperature of the UV scan and the reference temperature (31 °C), as indicated later in Eq. (9).

The angular response of a Brewer spectrophotometer deviates considerably from the ideal angular behaviour. This deviation is mainly caused by imperfections in the entrance optics and is called cosine error, after the ideal behaviour. For single Brewers, the cosine error varies between 8 and 12 % (Bais et al., 2005; Garane et al., 2006), while for double Brewers it ranges between 4–11 % (Antón et al., 2008; Bais et al., 2005; Lakkala et al., 2018). Although cosine correction is one of the most important uncertainty sources (Garane et al., 2006; González et al., 2024b), it is rarely characterised in RBCC-E campaigns (e.g. Lakkala et al., 2018). During the 18th RBCC-E campaign, five MkIII Brewer spectrophotometers characterised their angular response error. For this selection of Brewers, the cosine correction factor was calculated as (Gröbner et al., 1996):

$$f_{\rm g} = f_{\rm d} \left( 1 - \frac{DIR}{GLO} \right) + f_{\rm r} \frac{DIR}{GLO},\tag{3}$$

where  $f_d$  and  $f_r$  are the diffuse and direct cosine errors respectively,  $f_g$  the cosine correction factor, and DIR and GLO the direct and global irradiances.

In Eq. (3),  $f_r = C_R(\varphi, \theta, \lambda)/\cos\theta$ , where  $C_R(\varphi, \theta, \lambda)$  is the angular response of the Brewer diffuser. For most Brewers, this was measured using the Brewer Angular Tester (BAT), which measures the North-South and West-East planes using a 150 W Xe lamp placed at fixed angles (from -85° to 85° by steps of 5°). The standard uncertainty of the direct cosine error was derived from repeated measurements of Brewer #150, resulting in values varying from 0.002 at 5° to 0.006 at 85°.

The diffuse cosine error was estimated by assuming an isotropic sky radiance and integrating numerically the angular response  $f_{\rm d} = \frac{1}{\pi} \int_{\varphi=0}^{2\pi} \int_{\theta=0}^{\pi/2} C_{\rm R}(\varphi,\theta,\lambda) \sin\theta \ d\theta d\varphi$ . This factor is specific for each instrument, for the Brewers studied it

ranged from 0.908 (Brewer #186) to 0.986 (Brewer #150). Its uncertainty was estimated as indicated by Bernhard and Seckmeyer (1999), based on the findings of Gröbner et al. (1996):

$$u(f_{\rm d}) \approx \frac{|1-f_{\rm d}|}{|1-f_{\rm d}^{\rm G}|} \Delta D^{\rm G},\tag{4}$$

where  $f_d^G$  is 0.883 (the diffuse error found by Gröbner et al. (1996)) and  $\Delta D^G$  is the difference found by Gröbner et al. (1996) between  $f_d^G$  and the diffuse error derived for a inhomogeneous sky radiance distribution.

Finally, the ratio *DIR/GLO* was calculated using the radiative transfer model libRadtran (Emde et al., 2016) for a variety of SZAs. Other inputs to the model were the average Angström's turbidity coefficient (0.039) and Angström's exponent (1.371) derived from the El Arenosillo CIMEL measurements and mean TOC recorded by Brewer #150 during the campaign (295 DU), as well as the surface albedo (0.05), determined with a Li-Cor spectroradiometer. The uncertainty of this quantity at every SZA was estimated from all ratios measured within the desired SZA 1° (angular variability).

#### 3.2 UV processing algorithm



Spectral UV irradiance measured at wavelength  $\lambda$  was obtained following the standard processing of EuBrewNet, except for those Brewer with cosine correction. This uncertainty source was estimated using a different methodology, since the one currently implemented in EuBrewNet could not be easily included in the Monte Carlo simulation. The processing algorithm entails a series of corrections as the UV irradiance measured is affected by several error sources (see Section 3.1.). The first step is to correct the raw signal registered  $C_0(\lambda)$  for stray light, dark signal, and dead time (UV level 1 in EuBrewNet). Stray light is calculated using Eq. (1) and is then subtracted from the raw signal (only for single Brewers)

$$C_1(\lambda) = C_0(\lambda) - S(\lambda). \tag{5}$$

Then, dark signal and dead time are corrected according to the practices established by the manufacturer (Kipp & Zonen, 2018). Dark signal is simply subtracted from the UV signal

$$C_2(\lambda) = C_1(\lambda) - D,\tag{6}$$

where  $C_1(\lambda)$  are the counts corrected for stray light and D the dark signal.

On the other hand, dead time is corrected iteratively (n = 1...10) by assuming Poisson statistics:

$$C_3(\lambda, n+1) = C_2(\lambda) \cdot \exp(\tau C_3(\lambda, n)).$$
 (7)

In Eq. (7),  $C_2(\lambda)$  is the observed count rate (corrected for dark signal and stray light) at wavelength  $\lambda$ ,  $\tau$  the dead time, and  $C_3(\lambda)$  the true count rate. As a first guess,  $C_3(\lambda, 1) = C_2(\lambda)$ .

Then, the uncorrected absolute irradiance  $E_0^{\rm M}(\lambda)$  is obtained by dividing the corrected count rates by the response of the instrument:

$$E_0^{\mathcal{M}}(\lambda) = \frac{c_3^{\mathcal{M}}(\lambda)}{r(\lambda)},\tag{8}$$

where  $C_3^{\rm M}(\lambda)$  is the corrected signal (stray light, dark signal, and dead time) measured outdoors, and  $r(\lambda)$  the responsivity of the Brewer. The latter is determined by performing a radiometric calibration, (see Section 3.1.2).

The resulting UV irradiances need further processing to correct the temperature dependence, wavelength shifts, and cosine error.

First, they are corrected for temperature dependence by assuming a linear relationship:

$$E_1^{\mathcal{M}}(\lambda) = \frac{E_0^{\mathcal{M}}(\lambda)}{1 + c_{\mathcal{T}}(T - T_{\text{ref}})},\tag{9}$$

where  $E_0^{\rm M}(\lambda)$  is the uncorrected irradiance measured at wavelength  $\lambda$ ,  $E_1^{\rm M}(\lambda)$  is the irradiance corrected for temperature,  $c_{\rm T}$  is the temperature correction factor, and  $T_{\rm ref}$  is the reference temperature.

Secondly, the cosine correction is carried out using the methodology set by Gröbner et al. (1996). This derives a correction factor for each wavelength measured using Eq. (3) and corrects the irradiance by:

$$E_2^{\mathcal{M}}(\lambda) = E_1^{\mathcal{M}}(\lambda) / f_g(\lambda, \theta) , \tag{10}$$

where  $E_1^{\rm M}(\lambda)$  is the irradiance corrected for the temperature dependence,  $f_{\rm g}(\lambda,\theta)$  the cosine correction factor, and  $E_3^{\rm M}(\lambda)$  is the corrected irradiance for temperature and cosine error.

Finally, the irradiance is further corrected for the wavelength shifts using the SHICrivm software (Slaper et al., 1995). Only spectra recorded at SZAs smaller than 90° were used in this study. Larger SZA values have not been considered as the UV irradiance recorded in these conditions is small, close to the detection threshold of the Brewer spectrophotometers. Furthermore, since the "El Arenosillo" Observatory is at sea level, at large SZAs the instability of the atmosphere increases due to sea turbulence.

#### 3.3 Monte Carlo method

425

The Monte Carlo method estimates the uncertainty of the measurement by propagating the distribution of the input quantities, i.e. the uncertainty sources. Following the GUM guidelines, the uncertainty arising from random and systematic errors have been treated identically (BIPM et al., 2008b). These are determined by drawing from the probability density function (PDF) of the error sources. In this study, two types of PDFs have been considered: gaussian and rectangular. Following the recommendations of the GUM, gaussian distributions are assigned to those variables that can be characterised by a best estimate and a standard uncertainty, such as dead time, dark signal, or noise. On the other hand, rectangular distributions are appropriate for those sources that are best described by a lower and upper limit, i.e. the probability that the true value of the variable lies within the fixed interval is constant and is zero outside this interval. This is the case for the distance adjustment or the wavelength shift variables.

Once the distributions are known, the MCM can be implemented. To carry out this task, the number of times the model will be evaluated (Monte Carlo trials, M) must be first selected. In this study,  $M = 10^6$  since this value is expected to deliver a

95 % coverage interval for the spectral irradiance (BIPM et al., 2008b). Then, at every trial, the uncertainty sources are varied according to their PDFs, forming a M-sized vector for every source. For each of the M draws, the irradiance model is evaluated, obtaining a  $M \times \lambda$  matrix of the output irradiance. Then, the average and the standard deviation of these irradiances are taken as the best estimate and its standard uncertainty, respectively.

The procedure described above is the one implemented to calculate the combined standard uncertainty. However, the MCM can also estimate the contribution of each uncertainty source to the total uncertainty budget. This is performed by running the *M* trials while varying only one uncertainty source and fixing the rest at their best estimate (BIPM et al., 2008b). In this way, the dominant uncertainty sources can be identified.

#### 4 Results

- The uncertainty evaluation was performed for all the UV scans measured during the campaign under cloud-free conditions. However, the results obtained were very similar in all cases. Therefore, only the estimations corresponding to 13 September 2023 are shown in this section. This day was selected as most Brewers measured uninterruptedly (no maintenance or calibrations were performed) and under clear sky scenarios. Under cloudy conditions, the methodology for calculating the cosine correction and noise must be adapted accordingly. As the cosine correction depends on the cloudiness, the cloud cover must be considered when modelling the direct-to-global ratio. Furthermore, clouds strongly affect the surface UV irradiance and can lead to short-time variations. As a result, noise needs to be thoroughly characterised. For example, by studying the variability of groups of data measured very close in time. For Brewer spectrophotometers, this can be difficult as the instrument does not have enough temporal resolution to detect fast fluctuations of solar UV irradiance.
- To present the results, the Brewers studied have been separated into two groups. The first group includes the five Brewers whose angular responses were characterised (#150, #158, #185, #186, and #256). The remaining five Brewers were gathered in a second set as their characterisation is less elaborated. Therefore, the second group has two single (#117 and #151) and three double (#172, #202, and #228) Brewers. The uncertainty evaluation of the Brewer spectrophotometers in this second group is limited as it is missing one of the key uncertainty sources in solar radiometry, cosine correction. As a result, the uncertainties determined are likely an underestimation. Nevertheless, these estimations represent the uncertainty of the spectral irradiance reported by most of the participating Brewer spectrophotometers.

Figure 1: Spectral UV irradiance recorded at 14:00 UTC on 13 September 2023 by all the Brewer spectrophotometers studied. (a) First group formed by double Brewers with cosine correction (#150, #158, #185, #186, and #256). (b) Second group formed by two single (#117 and #151) and three double Brewers (#172, #202, and #228) with no cosine correction.

Figure 1 illustrates one of the UV spectra recorded on 13 September 2023. This will help understand the behaviour of the combined standard uncertainty presented in the following section. As can be seen from Fig. 1, the spectral UV irradiance increases rapidly between 290 and 310 nm (due to the decrease of the ozone absorption). Then, from 315 nm onwards, it levels off. For wavelengths shorter than 300 nm, the single Brewers (#117 and #151) are unable to measure UV irradiance with the same precision as the double monochromator instruments.

# 4.1 Combined standard uncertainty





The absolute combined standard uncertainty of all Brewer spectrophotometers depended on the wavelength and the solar zenith angle (SZA), in a similar way than the spectral UV irradiance (see Fig. 1). That is, the absolute uncertainty increases as wavelength grows and SZA decreases. Below 300 nm, the differences between double and single Brewers increase greatly, with single Brewers (#117 and #151) having absolute uncertainties that at least triple those of double Brewers. This was expected as single Brewers are affected greatly by stray light and the effect of its correction is more pronounced in the UV-B region. On the other hand, there are slight variations between the absolute uncertainties of double Brewers, mostly caused by the correction of (a) dark signal and noise at short wavelengths and (b) dead time and cosine error at larger wavelengths. The influence of these uncertainty sources will be studied in the following section (sensitivity analysis).

To better understand the magnitude of the combined standard uncertainty, it is interesting to study its relative values. The relative combined standard uncertainty (the absolute combined standard uncertainty divided by the UV irradiance measured) displayed different behaviours with wavelength and SZA depending on the instrument. For most Brewers, the relative

uncertainty values ranged from 2.5 % to 4 % for wavelengths between 300 and 360 nm and some Brewers showed almost no SZA dependency, as shown later in Figure 3.

Figure 2. Relative combined standard uncertainties of the UV irradiances shown in Fig. 1. (a) First group (double Brewers with cosine correction). (b) Second group (two single and three double Brewers with no cosine correction implemented).





To illustrate the wavelength dependency, the relative uncertainties of the UV scan performed on 13 September 2023 at 14:00 UTC (40° SZA) are shown in Figure 2. It should be noted that the relative uncertainties of all Brewers increase significantly below 300 nm as the UV irradiances measured are very small, close to 0 W m<sup>-2</sup> nm<sup>-1</sup>. Between 300 and 360 nm, the relative combined standard uncertainty of some Brewer spectrophotometers (a) decreases slightly with wavelength (#150 and #186), (b) increases gradually with wavelength (#202), (c) fluctuates significantly (#158 at short wavelengths and #151 at large wavelengths), and (d) is approximately constant (#117, #172, #185, #228, and #256). The reason for these behaviours will be described later in Section 4.3 (sensitivity analysis).

Regarding the angular dependency, Figure 3 represents all the relative uncertainty values derived on 13 September at 335 nm. This wavelength was selected to minimise the effect of the fluctuations found for Brewers #151 and #158 (see Figure 2). Figure 3 shows that the relative combined standard uncertainty of half of the Brewers (#117, #150, #151, #172, #228) has no angular dependency. On the other hand, the relative uncertainties of the remaining Brewer either increase (first group except for #150) or slightly decrease (#202) with SZA.

Figure 3. Relative combined standard uncertainties of all UV irradiances measured on 13 September 2023 at 335 nm. (a) First group (double Brewers with cosine correction). (b) Second group (two single and three double Brewers with no cosine correction implemented).

#### 4.2. Comparison against the QASUME



The corrections applied to the measured irradiance (described in Section 3.2) are recommended by numerous studies to improve the quality of the measurements (e.g. Fountoulakis et al., 2016b; Garane et al., 2006; Kerr, 2010; Lakkala et al., 2008, 2018). This was also verified during the 18th RBCC-E campaign, as the results show that including the cosine correction improves considerably the comparison to the QASUME (Hülsen, 2023). Although the campaign report shows the ratio of each participating Brewer to the QASUME (see Hülsen (2023)), it is interesting to represent the ratio of all studied Brewers together. In this way, Fig. 4 displays the global irradiance ratio to the QASUME obtained from dividing the irradiances shown in Fig. 1 to the irradiance recorded by the QASUME unit.

Figure 4 shows the effectiveness of the cosine correction, since Brewers with such correction implemented (Fig. 4a) report irradiances more similar to the one measured by the QASUME. Nevertheless, the agreement between all Brewer spectrophotometers and the QASUME is within  $\pm 10$  % for wavelengths above 310 nm.

Furthermore, the irradiance uncertainty found for each Brewer in the previous section can be used to derive the uncertainty of their ratio to the QASUME. Table 3 summarises the combined standard uncertainty of the average Brewer/QASUME ratio measured on 13 September at three different wavelengths. These uncertainties were computed by combining the irradiance uncertainty of each Brewer and the one from the QASUME, provided by Hülsen et al. (2016).

Table 3. Number of simultaneous scans, mean ratio to the QASUME and its combined standard uncertainty (both absolute and relative) determined between 310 and 360 nm on 13 September.

| Brewer ID | NI  | Ratio to the QASUME (310–360 nm) |                               |                                   |  |  |  |  |  |  |
|-----------|-----|----------------------------------|-------------------------------|-----------------------------------|--|--|--|--|--|--|
| Brewer ID | N _ | Mean value                       | Combined standard uncertainty | Relative standard uncertainty (%) |  |  |  |  |  |  |
| #117      | 19  | 0.927                            | 0.034                         | 3.7                               |  |  |  |  |  |  |
| #150      | 20  | 1.035                            | 0.035                         | 3.4                               |  |  |  |  |  |  |
| #151      | 24  | 0.914                            | 0.033                         | 3.6                               |  |  |  |  |  |  |
| #158      | 17  | 0.972                            | 0.036                         | 3.7                               |  |  |  |  |  |  |
| #172      | 19  | 0.947                            | 0.033                         | 3.5                               |  |  |  |  |  |  |
| #185      | 18  | 0.978                            | 0.030                         | 3.1                               |  |  |  |  |  |  |
| #186      | 15  | 1.003                            | 0.043                         | 4.3                               |  |  |  |  |  |  |
| #202      | 19  | 0.928                            | 0.033                         | 3.6                               |  |  |  |  |  |  |
| #228      | 19  | 0.937                            | 0.033                         | 3.5                               |  |  |  |  |  |  |
| #256      | 19  | 1.003                            | 0.037                         | 3.7                               |  |  |  |  |  |  |

Table 3 shows that only those Brewer spectrophotometers with a cosine correction implemented (#150, #158, #185, #186, and #256) include the ideal value of the ratio (unity) within their uncertainty interval. The remaining Brewers underestimate the UV irradiance and deviate from unity. This is likely caused by the cosine and temperature errors of the instruments, which couldn't be corrected (there was no available information regarding their characterisation). Therefore, to improve the performance of these uncorrected Brewers these two sources must be characterised and corrected.

# 4.3 Sensitivity analysis


505

510

To clarify which uncertainty sources are responsible for the different behaviours found for each Brewer, a sensitivity analysis has been performed. Generally, for wavelengths above 300 nm, the dominant uncertainty sources are radiometric

stability, cosine correction (if implemented), and the uncertainty of the reference lamp. As an example, the contribution of each uncertainty source to the combined standard uncertainty is shown in Fig. 5 for a single (#117) Brewer and in Fig. 6 for a double (#185) Brewer spectrophotometer. As the intensity of the incoming UV radiation decreases, i.e. as wavelength decreases and SZA rises, the uncertainty associated with the correction of dark signal, stray light (if the Brewer is a single monochromator), and noise begin to gain influence. In fact, for wavelengths below 295 nm, they become the dominant sources (see Figs. 5 and 6). However, there were exceptions to this behaviour (mainly Brewers #117, #151, #158, and #202), showing that the uncertainties in dead time correction, noise, and wavelength shift can also become dominant uncertainty sources in the UV-A region (315–400 nm). Regarding the calibration of the instrument (uncertainty sources affecting the responsivity and the signal measured under the reference lamp), it leads to irradiance uncertainties that range from 2.3 % (Brewer #185) to 3.8 % (Brewer #150).

Figure 5. Relative contribution of the uncertainty sources of a single monochromator Brewer (#117) to the combined standard uncertainty of the UV spectrum measured at three wavelengths (293. 320, and 360 nm) and two SZAs, (a) 35° and (b) 63°. Each contribution was calculated from the average over a  $\pm 1^{\circ}$  SZA band, with N being the number of measurements considered for the average.

Figure 6. Relative contribution of the uncertainty sources of a double monochromator Brewer (#185) to the combined standard uncertainty of the UV spectrum measured at three wavelengths (293. 320, and 360 nm) and two SZAs, (a) 33° and (b) 63°. Each contribution was calculated from the average over a  $\pm 1^{\circ}$  SZA band, with N being the number of measurements considered for the average.

As a summary, Table 4 shows the relative individual and combined standard uncertainties for each Brewer under study at SZAs below 80° and wavelengths larger than 300 nm. Larger SZAs and shorter wavelengths have not been included in this table as the relative uncertainties increase greatly since the UV irradiance measured approaches zero (see Fig. 1).

Table 4. Range of the irradiance uncertainties produced by each uncertainty source individually and the combined standard uncertainty for SZAs below 80° and wavelengths larger than 302 nm for each of the Brewers studied.


|      |        | Individual uncertainty (%) |       |       |                    |         |       |          |         |       |             |
|------|--------|----------------------------|-------|-------|--------------------|---------|-------|----------|---------|-------|-------------|
| ID   | Noise  | Dark                       | Stray | Dead  | Dist.              | Unc.    | Stab. | λ shift  | Temp.   | Cos.  | uncertainty |
| l No | Noise  | signal                     | Light | time  | Adjust.            | Lamp    | Stab. | λ SIIIIt | Corr.   | Corr. | (%)         |
| #117 | 0.070- | 0.060-                     | ≤36   | ≤0.17 | 0.23               | 1.2–1.3 | 3.0   | 0.004-   |         |       | 3.3–36      |
| #11/ | 5.3    | 8.6                        | ≥30   | ≤0.17 | 0.23   1.2–1.3   3 |         | 3.0   | 1.6      | _       | _     | 3.3–30      |
| #150 | 0.029- | 0.0044-                    |       | ≤0.84 | 0.24               | 0.61-   | 3.6   | 0.004-   | 0.0091- | 0.43- | 2.9–5.1     |
| #130 | 3.1    | 3.8                        | _     | ≥0.64 | 0.24               | 0.75    | 3.0   | 0.82     | 0.037   | 0.51  | 2.5–3.1     |
| #151 | 0.080- | 0.043-                     | ≤33   | ≤0.58 | 0.23               | 0.43-   | 3.0   | 0.014-   |         |       | 3.1–34      |
| #131 | 2.9    | 12                         | 233   | ≥0.56 | 0.23               | 0.61    | 3.0   | 3.8      | _       | _     | 3.1–34      |
| #158 | 0.048- | 0.043-                     |       | ≤0.66 | 0.28               | 0.85-   | 3.0   | 0.014-   |         | 0.80- | 3.3–5.5     |
| #138 | 2.6    | 2.9                        | _     | ≥0.00 | 0.28               | 1.4     | 3.0   | 2.4      | _       | 1.6   | 3.5–3.5     |
| #172 | 0.026- | 0.0013-                    | _     | ≤0.44 | 0.23               | 0.39-   | 3.0   | 0.015-   | _       | _     | 3.0–7.8     |

|      | 1.3    | 6.3      |   |                |      | 0.67    |     | 1.2    |   |       |                      |
|------|--------|----------|---|----------------|------|---------|-----|--------|---|-------|----------------------|
| #185 | 0.073- | 0.0034-  |   | ≤0.28          | 0.23 | 0.60-   | 2.8 | 0.003- |   | 0.84- | 2.4–7.7              |
| #103 | 2.8    | 9.1      | _ | ≥0.28          | 0.23 | 1.1     | 2.0 | 0.63   | _ | 1.6   | 2.4-7.7              |
| #186 | 0.039- | 0.012-   |   | ≤0.85          | 0.23 | 1.2–1.3 | 3.0 | 0.005- |   | 1.6-  | 3.6–5.8              |
| #100 | 2.0    | 7.9      | _ | ≥0.63          | 0.23 | 1.2–1.3 | 3.0 | 1.2    | _ | 3.3   | 3.0-3.8              |
| #202 | 0.19-  | 0.0034-  |   | ≤1.5           | 0.23 | 0.60-   | 3.0 | 0.01-  |   |       | 3.1-8.0              |
| #202 | 6.7    | 3.0      | _ | ≥1.5           | 0.23 | 1.1     | 3.0 | 1.3    | _ | _     | 3.1-6.0              |
| #228 | 0.042- | 0.011 -  | _ | ≤0.20          | 0.23 | 0.60-   | 3.0 | 0.004- |   | _     | 3.1–4.4              |
| #220 | 2.8    | 0.82     |   | <u>\$</u> 0.20 | 0.23 | 1.1     | 3.0 | 1.1    | _ |       | 3.1-4.4              |
| #256 | 0.048- | 0.0021 - |   | ≤0.60          | 0.23 | 0.60-   | 3.0 | 0.005- |   | 0.92- | 3.2–4.8              |
| #230 | 2.8    | 0.50     | _ | ≥0.00          | 0.23 | 1.1     | 3.0 | 0.68   |   | 1.8   | 3.2 <del>-4</del> .6 |

In the following, the influence of each uncertainty source on the total uncertainty budget will be described in greater detail.

#### 4.3.1 Noise

For most Brewers, the irradiance uncertainty produced by noise was most dominant (second or third most influential source of uncertainty) for wavelengths below 300 nm. At larger wavelengths, above 310 nm, this source loses influence, resulting in an uncertainty of less than 0.6 % in the UV irradiance measured, regardless of the intensity of the incoming radiation. Brewer #117 was an exception, with a SNR of 0.1, noise led to irradiance uncertainties of up to 0.9 %. In this case, noise was the third most dominant source for wavelengths larger than 330 nm (see Fig. 5).

# 4.3.2 Dark signal

The irradiance uncertainty caused by dark signal correction is important solely at wavelengths smaller than 295 nm. For larger wavelengths, its impact can be disregarded as dark signal correction leads to irradiance uncertainties of less than 0.06 % in double monochromator Brewers. It is interesting to note that single Brewer spectrophotometers showed larger dark signal contributions. For example, for wavelengths above 310 nm, dark signal caused irradiance uncertainties of 0.3 %. These two Brewers (Brewers #117 and #151) recorded during the campaign larger dark signals than double Brewers. As a result, the standard deviation obtained is greater as well (see Section 3.1.1), resulting in a larger contribution.

# 4.3.3 Stray light

Stray light was only considered for the uncertainty evaluation of single monochromator Brewers, i.e. Brewers #117 and #151. The sensitivity analysis shows that the irradiance uncertainty produced by stray light correction increases rapidly as

wavelength decreases. Furthermore, it also increases with SZA as shown in Fig. 5. For single Brewers and at wavelengths below 300 nm, stray light is the dominant source, accounting for more than 95 % of the total uncertainty budget.

# 4.3.4 Dead time





Dead time contribution increases with the number of UV photons recorded, i.e. as SZA declines and wavelength rises. Most of the Brewers used in this study have an uncertainty of 1 ns in their dead time, which results in a maximum uncertainty in the irradiance measured of 0.2 % at 68° SZA and 0.8 % at 33° SZA. Therefore, dead time is not a dominant uncertainty source and only becomes significant at small SZAs (fourth or fifth most influential source). However, this is not true for larger dead time uncertainties, as is the case of Brewer #202. Since its dead time uncertainty is 3 ns, the maximum uncertainty produced in the irradiance measured is 0.8 % at 57° and 1.5 % at 33°. This larger uncertainty is likely caused by the replacement and voltage adjustment of the standard lamp during the RBCC-E campaign (WMO, 2023). Thus, for this Brewer the dead time correction is the second most influential uncertainty source for wavelengths larger than 320 nm.

Finally, the irradiance uncertainties previously estimated can be compared to the one reported by Fountoulakis et al. (2016b). Their study shows that if the DT ranges from 15 to 45 s and has an error of 2 ns, it leads to irradiance uncertainties of 0.12–0.13, 0.25–0.28, and 0.69–1.13 % for signals of 1, 2, and 5 million counts s<sup>-1</sup>, respectively. These values are similar to the ones found for all Brewers, except Brewer #202, which has an uncertainty larger than 2 ns. For these Brewers, the irradiance uncertainty is less than 0.15, 0.35, and 0.9 % for signals of 1, 2, and 5 million counts s<sup>-1</sup>, respectively.

#### 585 4.3.5 Distance adjustment

An uncertainty of 0.58 mm when placing the reference lamp at 500 mm results in a 0.23 % uncertainty in irradiance. On the other hand, Brewers #150 and #158 display slightly different results, as their reference lamps were placed at 413 mm and 497.7 mm with uncertainties of 0.58 mm and 0.59 mm, respectively (see Sect. 3.1.2). For these Brewers, the distance adjustment leads to uncertainties of 0.24 % (#150) and 0.28 % (#158). According to Webb et al. (1998), if the nominal distance is d and its uncertainty  $u_d$ , the percentage uncertainty can be calculated using the inverse square law  $(1/r^2$ , where r is the distance between lamp and instrument) as  $[(d + u_d)^2 - d^2] * 100 / d^2$ . Therefore, the previous results agree with the formula proposed by Webb et al. (1998). It should be noted that all uncertainty sources involved in the calibration of the instrument (distance, uncertainty of the reference lamp, and radiometric stability) have no angular dependency. As they only affect the responsivity of the instrument, they have the same influence on all UV measurements as shown in Eq. (8). Furthermore, the uncertainty produced by the distance adjustment has no spectral dependency. Since the reference lamp can be regarded as a point source, the UV irradiance follows the inverse square law. Therefore, a change in distance has the same effect on all wavelengths measured.

# 4.3.6 Uncertainty of the reference lamp

The uncertainty of the 1000 W lamps used during the calibration is the second most dominant uncertainty source in the UV-A region. The irradiance uncertainty depends on the lamp used and the wavelength measured, ranging from 0.6–1.4 % at 290 nm to 0.4–1.2 % at 360 nm. The Brewer spectrophotometers least affected by this uncertainty source had been calibrated using two reference lamps during the intercomparison campaign. Therefore, the overall uncertainty of a Brewer spectrophotometer can be reduced by calibrating the instrument with more than one reference lamp. This agrees with the recommendations of Webb et al. (1998). They suggest calibrating the instruments using three reference lamps.

# 605 4.3.7 Radiometric stability




Radiometric instability is the dominant uncertainty source for all Brewer spectrophotometers, for wavelengths larger than 300 nm (see Figs. 5 and 6). For most Brewers, this uncertainty source leads to irradiance uncertainties of 3 %. This was expected, since the UV irradiance is inversely proportional to the responsivity (as shown in Eq. (8)), and a 3 % uncertainty in the responsivity was assumed for most Brewers, in agreement with the findings of Garane et al. (2006) and Lakkala et al. (2008) (see Section 3.1.2.). On the other hand, Brewers #150 and #185 had their instability characterised using their calibration records and reported irradiance uncertainties of up to 3.6 % and 2.5 %, respectively. Brewer #150 is calibrated yearly using 1000 W lamps, while Brewer #185 is calibrated approximately every 2–3 months using 200 W lamps and yearly with 1000 W lamps. Therefore, calibrating frequently is recommended to reduce the instrument's combined standard uncertainty.

#### 615 4.3.8 Wavelength shift

Wavelength shifts are responsible for the rapid fluctuations of the relative uncertainties of Brewers #151 and #158 (see Figure 2). The spikes were larger for Brewer #151 as the wavelength shifts of this instrument, for wavelengths above 350 nm, were 10 times larger than the ones of the other Brewers. For Brewer #151, a wavelength shift of 0.12 nm at 355 nm resulted in a relative irradiance uncertainty of 4 %, becoming the dominant uncertainty source at this region. This is interesting as it shows the influence wavelength shifts can have on the UV-A irradiance measured. On the other hand, Brewer #158 has shifts of 0.05 nm, resulting in a 1 % uncertainty in the irradiance for wavelengths between 310 and 360 nm. Furthermore, these large wavelength shifts indicate that the dispersion function of these instruments might be outdated. Therefore, special attention should be paid to the wavelength scale of the instrument by performing frequent and accurate wavelength calibrations.

For the remaining Brewers, the contribution of wavelength shift is negligible for wavelengths above 300 nm (less than 0.3 %). At shorter wavelengths, shifts of 0.03 nm can produce up to 20 % irradiance uncertainty. Nevertheless, they are not a dominant uncertainty source regardless of the wavelength and SZA measured (see Figs. 5 and 6).

# 4.3.9 Temperature correction

Although temperature correction has an important effect on the UV irradiance measured, its uncertainty has no significant impact on the overall uncertainty of Brewer #150. In fact, it leads to an irradiance uncertainty of less than 0.2 % for SZAs below 75°. However, since the dependency with temperature is specific for each instrument (Fountoulakis et al., 2017), different results may be found for other Brewers.

#### 4.3.10 Cosine correction




As mentioned earlier, this uncertainty source could only be studied for Brewers #150, #158, #185, #186, and #256, as they are the only ones with a characterised angular response. Figure 7 shows that the uncertainty in cosine correction has a great impact on the uncertainty budget of most Brewers, leading to an average irradiance uncertainty that ranges from 0.4 % (#150) to 1.9 % (#186) at 33° SZA. Brewer spectrophotometers #158, #185, and #256 present an intermediate situation, with uncertainties of around 1.4 % in the irradiance measured. These differences are likely due to the entrance optics. Brewers #185, #186, #256, and #158 have a flat diffuser, while Brewer #150 has a shaped diffuser. Furthermore, the irradiance uncertainty caused by cosine correction increases gradually with SZA for all Brewers except #150. Consequently, this uncertainty source is responsible for the increase of the relative combined uncertainty standard with SZA observed in Fig. 2a.

Figure 7. Relative standard relative uncertainty on 13 September 2023 caused by the cosine correction implemented. (a) Spectral dependency at 12:30 UTC. (b) SZA dependency at 350 nm.

The uncertainties shown in Fig. 7 are mostly produced by the uncertainties of the diffuse  $(f_d)$  and direct  $(f_r)$  cosine errors. In fact, these factors account for more than 98 % of the total irradiance uncertainty caused by cosine correction. As for the

- direct to global irradiance ratio, its impact on the uncertainty budget is negligible as only cloud-free conditions have been considered in the analysis. This would likely change if overcast or mixed sky conditions were to be included in the uncertainty evaluation. Therefore, under cloud-free conditions, the main sources of uncertainty in the cosine correction are the errors committed in the angular characterisation and in the assumption of isotropic sky radiance to calculate the diffuse cosine error.
- For Brewer spectrophotometers #158, #185, #186, and #256, the correction of cosine error is the second most important source of uncertainty for wavelengths larger than 300 nm, regardless of the SZA. For Brewer #150, thanks to its improved angular response, this source has less impact, being the third most influential uncertainty source for SZAs larger than 50°.

#### 5 Applications of Brewer uncertainty evaluation

- This study provides an accurate quantification of measurement uncertainties in Brewer spectrophotometer UV data, identifying the main sources of uncertainty and their relative contributions to guide instrumental optimisations. These aspects are of great interest for different studies and fields of work.
  - One of the key applications of accurately determining the uncertainties in spectral UV measurements is the computation of effective irradiance for various biological effects, such as erythema, vitamin D synthesis, melanoma risk, and DNA damage, through the integration of the spectral irradiance weighted by different action spectra (Webb et al., 2011).
- The findings benefit regulatory applications, supporting evidence-based UV exposure limits for outdoors workers (Vecchia et al., 2007) and improving standards for sun protection products (Young et al., 2017). The proposed methodology also allows sensitivity analysis to help identify paths for improving instrumentation, measurement procedures, and calibration protocols, which are essential for ensuring the traceability of UV spectroradiometer measurements to international standards (Gröbner et al., 2006). Reliable measurements in the 300–400 nm wavelength range with a relative uncertainty below 4 % are crucial for radiometric networks and studies comparing data from different stations. Ensuring this quality level requires periodic and regular calibrations using lamps traceable to international standards. For example, the QASUME (Quality Assurance of Solar Ultraviolet Spectral Irradiance Measurements) project has established a European reference standard for UV solar radiation measurements, achieving a global UV irradiance uncertainty of approximately ±4 % in the 300–400 nm range (Gröbner and Sperfeld, 2005) and a direct solar irradiance uncertainty of about 0.7 % (Gröbner et al., 2023). More advanced developments, such as QASUMEII, have further improved accuracy, with a combined uncertainty for global UV measurements of 1.01 % between 310 and 400 nm and 3.67 % at 300 nm (Hülsen et al., 2016).
  - Furthermore, the uncertainty framework significantly strengthens the validation of satellite-based UV products from instruments such as OMI, TROPOMI, and TEMPO (Klotz et al., 2024; Tanskanen et al., 2007), where ground-based measurements with an uncertainty of less than 5 % are crucial for calibration.

In summary, precise quantification of uncertainty in spectral UV measurements benefits a broad range of scientific, regulatory, and public health applications, reinforcing the need for rigorous uncertainty assessment in Brewer spectrophotometer measurements.

# **6 Conclusions**






The uncertainties of the UV spectra measured by eight double and two single monochromator Brewer spectrometers have been estimated using a Monte Carlo method. The UV scans studied were performed during the 18th RBCC-E intercomparison campaign at the El Arenosillo (Huelva, Spain).

Using the information provided by participating operators and EuBrewNet, the uncertainty sources of the ten Brewers were characterised. This was difficult since the available data for many uncertainty sources was either limited (such as radiometric stability, stray light, and noise) or unavailable (cosine error and temperature dependence). Therefore, further work is needed to characterise the Brewer network thoroughly. Furthermore, this study also shows the necessity of establishing coherent QC procedures. The results obtained in this work may vary from the QC performed by other Brewer operators as their instrument characterisation and processing algorithms can be different.

Once characterised, the combined standard uncertainty (absolute and relative values) was derived and a sensitivity analysis was performed to identify the most influential uncertainty sources.

The absolute combined standard uncertainty of single and double monochromator Brewers increases with increasing wavelength and decreasing SZA. For single Brewers, the absolute values are three times higher than those of double Brewers due to stray light. Small differences between double Brewers are observed, due to the influence of the correction of (a) dark signal and noise below 300 nm and (b) dead time and cosine error at larger wavelengths. Regarding the relative values (the absolute combined standard uncertainty divided by the UV irradiance measured) of all Brewers (single and double), it is instrument specific and varies between 2.5 % and 5 % for wavelengths larger than 300 nm. For half of the Brewers studied, the relative uncertainty shows no spectral nor angular dependency. This behaviour is linked to the dominant uncertainty sources. If radiometric stability is the dominant source, the relative combined uncertainty shows no spectral nor SZA dependency, as the stability doesn't have either of these dependencies. On the other hand, if the cosine error of the instrument is significant, then its correction leads to a relative irradiance uncertainty that depends on both wavelength and SZA. Furthermore, spikes in the relative combined standard uncertainty are expected if the wavelength shift is large enough. A shift of 0.1 nm can lead to uncertainties of 5 % in the UV-A region.

For the ten Brewer spectrophotometers analysed in this study, the average combined standard uncertainty in erythemal spectral irradiance ranges between 2.7 % and 3.9 %, with maximum values varying from 17 % for a single Brewer to 3.4 % for a double Brewer for wavelengths above 310 nm. This variability indicates the need of characterising each Brewer spectrophotometer individually rather than relying on generic values, which may not fully exploit the precision these instruments can achieve (Gröbner et al., 2006). When integrating erythemal spectral irradiance to compute the UV Index

(UVI), the resulting uncertainty ranges from 2.7 % to 6.2 %. The UVI, along with cumulative erythemal irradiance doses, represents a fundamental metric for informing the public about the potential adverse effects of UV radiation (Lucas et al., 2019).

The sensitivity analysis performed shows that the source of uncertainties in the Brewer signal (noise, dark signal, stray light, and dead time) are important for wavelengths below 300 nm and large SZAs. However, they can also become significant above 310 nm and for SZAs below 50° if the uncertainties of dead time and SNR are larger than 3 ns and 10 %, respectively. For wavelengths above 300 nm, cosine correction (when implemented), radiometric stability, and the uncertainty of the reference lamp are usually the most dominant sources, regardless of the SZA. Radiometric stability is the most influential out of these three uncertainty sources, causing an irradiance uncertainty of 3 %.

Based on the findings of this sensitivity analysis, to reduce the overall uncertainty of a Brewer spectrophotometer, it is recommended to (a) monitor the instrument' stability by calibrating it more than once a year, (b) calibrate the reference lamps periodically to ensure up-to-date calibration certificates, (c) replace the traditional entrance optics to improve the angular response, (d) monitor the dead time to ensure uncertainties of less than 2 ns, (e) monitor wavelength shifts and reduce them below 0.05 nm through frequent wavelength calibrations and accurate determinations of the instrument wavelength scale, and (f) calibrate the instrument using two or more reference lamps. Although replacing the entrance optics will modify the responsivity of the instrument, this change will not affect the calculation of UV trends or long-term monitoring as long as the data is re-evaluated and its QC revisited (Fountoulakis et al., 2016a).






The relative combined standard uncertainties of the Brewers used in this study can be compared with the ones obtained in previous studies. Garane et al. (2006) determined a combined standard uncertainty of 5.3 % at 320 nm for a single Brewer (MKII version). This value is slightly larger than the one obtained in our work at 320 nm (3.1–3.3 %). This is likely produced by cosine correction. While Garane et al. (2006) included this uncertainty source in their evaluation, none of the single Brewers participating in the RBCC-E campaign had their cosine error characterised. Regarding the double Brewers studied, their UV irradiance uncertainty ranges between 2.5 % and 5 % for wavelengths larger than 300 nm. These values are similar to the uncertainty found by Garane et al. (2006). They reported a relative uncertainty of 4.8 % for their double Brewer. Furthermore, the uncertainty of the double Brewers studied is also comparable to the European reference units, QASUME I and QASUME II. Hülsen et al. (2016) found relative uncertainties of 3.85 % and 3.67 % at 300 nm for QASUME and QASUME III, respectively. Moreover, the irradiance uncertainties determined in this work are similar to the ones described in other publications (Bernhard & Seckmeyer, 1999; Fountoulakis et al., 2020). Therefore, the relative combined standard uncertainties determined in this study are comparable to those of other UV spectroradiometers. This also applies to the uncertainties of erythemal irradiance and UV index, as the values estimated are also similar to the ones found for other instruments (Bernhard and Seckmeyer, 1999; Cordero et al., 2007).

Finally, it should be noted that further work is needed to ensure that the uncertainty of all UV scans measured by Brewer spectrophotometers is evaluated. The Monte Carlo method used in this study is easy to implement, but it requires a large number of trials to provide reliable results. Considering the number of uncertainty sources in Brewer measurement

procedure, this results in a heavy calculation cost. On a standard laptop, it took around 8 hours per UV scan measured to calculate the combined standard uncertainty and the sensitivity analysis (10<sup>6</sup> iterations). Although this execution time could be reduced by optimising the code or using a computer with better performance, the MCM can be impractical to evaluate the uncertainty of Brewers long UV records.


Data and code availability. The data used for this study is available at EuBrewNet (<a href="http://eubrewnet.aemet.es/">http://eubrewnet.aemet.es/</a>) and the code used was based on the algorithm available at Zenodo (González et al., 2024a). Aerosol optical depth measured at INTA/El Arenosillo station was used for the cosine correction and can be downloaded from AERONET (Holben et al., 1998).

Auth Valid Resc

Author contributions. Conceptualization, C.G., J.M.V, and A.S.; Methodology, C.G. and A.S.; Software, C.G. and A.S.; Validation, C.G., J.M.V., A.R., J.L.S., J.M.S.A., and A.S.; Formal analysis, C.G. and J.L.S.; Investigation, C.G. and J.M.V.; Resources, J.M.V., A.R., J.L.S., J.M.S.A., R.K., A.R.D.S., P.B., A.D., N.J., and G.G.; Data Curation, C.G., J.M.V., A.R., J.L.S., and A.S.; Writing – Original Draft, C.G.; Writing – Review & Editing, J.M.V., A.R., J.L.S., J.M.S.A., R.K., A.R.D.S, P.B., A.D., N.J., G.G., and A.S.; Visualization, C.G.; Supervision, J.M.V., A.R., and A.S.; Funding acquisition, J.M.V.


Competing interests. The authors declare that they have no conflict of interest.

Acknowledgements. The authors thank the European Brewer Network (<a href="http://eubrewnet.aemet.es/">http://eubrewnet.aemet.es/</a>) for providing access to the data. Furthermore, authors are grateful to Vladimir Savastiouk who provided most of the calibration records of the Brewers as well as insight on how to include the stray light in the uncertainty evaluation. Gregor Hülsen is also to be thanked for providing the slit functions of the Brewers. Julian Gröbner is to be acknowledged for its insight on the precision of the micrometre step and the cosine correction methodology. Finally, we also thank Victoria E. Cachorro Revilla and Margarita Yela González for their effort in establishing and maintaining AERONET El Arenosillo/Huelva site.

Financial support. This work is part of the PID2023-149390OB-C22 and PID2023-149390OB-C21 R+D+I grants funded by MCIN/AEI/10.13039/501100011033/ and "ERDF A Way of Doing Europe".

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
