# Peer review of "Evaluation of the uncertainty of the spectral UV irradiance measured by double- and single-monochromator Brewer spectrophotometers"

_EGUsphere, 2025_

## Author Comment (AC1)

**Response to Anonymous Reviewer #1**

Authors' response to Reviewer #1 comments on "Evaluation of the uncertainty of the spectral UV irradiance measured by double- and single-monochromator Brewer spectroradiometers". The authors thank the Reviewer for the careful and constructive examination of the manuscript and reply to all comments below.

The answer is structured as follows: (1) comments from Reviewer #1, (2) authors' response and (3) authors' change in the manuscript.

(1) General comments

The paper presents a methodology to evaluate uncertainties in spectral UV irradiance measurements by Brewer spectroradiometers. The methodology considers most of the possible sources of uncertainty and has been applied on several instruments. In this respect, the study is innovative and useful for the uncertainty evaluation of the global Brewer network. However, it is alarming that the evaluation for one spectrum takes so long (8 hours), making the method impractical for evaluating large numbers of spectra.

The presentation is generally clear, but there are several places where clarifications are needed. See specific comments below.

(2) It took 8 hours to process both the spectrum and its sensitivity analysis. However, the time taken could be reduced by using a computer with a better performance and by optimizing the code. The latter can be achieved by using parallel programming or by using genetic algorithms such as particle swarm optimization.

(3) The previous information has been added to the text as follows: "Although this execution time could be reduced by optimising the code or using a computer with better performance, the MCM can be impractical to evaluate the uncertainty of Brewers long UV records."

(1) The language can certainly be improved. There are many typographical, syntax and grammar errors that can easily be corrected by careful reading. In my Technical Comments section below I have listed a few, but there are many more.

(2, 3) The manuscript has been carefully proofread to correct any typographical or grammatical errors.

(1) Therefore, my recommendation is that the paper would be suitable for publication in ACP, after addressing by comments below.

(1) Specific comments:

(1) 27: The difference between single and double monochromators is mainly due to stray light and not due to dark signal. I suggest deleting "and dark count".

(2, 3) The term "dark count" has been deleted and the phrase has been corrected to "the differences between single- and double-monochromator Brewers increase, due to stray light".

(1) 29-30: How the "irradiance of the calibration lamp" is involved in the uncertainty? I think you mean the uncertainty associated with the reference lamp calibration, therefore I suggest replacing "irradiance" with "uncertainty". The same applies to lines 418 and 618.

(2) It is the uncertainty of the irradiance emitted by the reference lamp. As the reviewer states, it is related to the calibration of the reference lamp.

(3) For greater clarity, the term "irradiance of the reference lamp" has been replaced with "uncertainty of the reference lamp" throughout the text.

(1) 153-155: I am not sure what you mean here: First you say that some uncertainties are considered in the analysis and then that you prefer to ignore them. Please clarify.

(2) Lines 153-155 refer to those uncertainty sources that have been included in the study, but their characterisation is not as thorough as the ones performed by other authors. For instance, the Brewer noise was determined by studying the signal-to-noise-ratio (SNR) from lamp measurements, as Bernhard and Seckmeyer (1999) or Cordero et al. (2012) did for the uncertainty evaluation of their instruments. In the RBCC-E campaign, Brewers were calibrated by performing 4 lamp scans. Therefore, the SNR was derived from a limited number of UV scans. However, we believe that, rather than ignoring these uncertainty sources, it is preferable to include them in the study, even if their characterisation is limited.

(3) To clarify this issue, the indicated lines have been rewritten as "In this study, the uncertainty sources have been characterised following the methodologies of Bernhard and Seckmeyer (1999), González et al. (2023, 2024), and Savastiouk et al. (2023). It should be noted that some of the uncertainties determined in this study (such as those related to noise, stray light, or radiometric stability) have their own uncertainty, as the data used for their estimation does not allow for appropriate statistics. Nevertheless, in all these cases, uncertainty values have been given and included in the Monte Carlo simulation".

(1) 157: Replace "counts" with "signal". It is better to use "signal" when you are referring generally to what is measured. This applies, for example, also to lines 159 and 160 and elsewhere.

(2, 3) Following the reviewer's suggestion, the term "counts" have been replaced with "signal" in lines 157, 159, and 160. To ensure consistency throughout the manuscript, the term "signal" has replaced "counts" whenever the text was referring generally to what is measured.

(1) 157: What do you mean by unprocessed? To what does this differ from (1)? My understanding after reading the next sections is that this category refers to uncertainties related to absolute irradiance, specifically to wavelength shifts, angular response and temperature dependence. All three affect the absolute irradiance rather than the raw (or unprocessed) irradiance.

(2) It is referring to the uncorrected UV irradiance. Even if the measured signal is processed, the derived irradiance is affected by other uncertainty sources (wavelength shifts, angular response, and temperature dependence) and needs correction.

(3) To clarify this issue, the term "raw irradiance" has been replaced with "uncorrected absolute irradiance" throughout the manuscript.

(1) 202: Does "40" refers to all dark signal measurements (which I think is too low) or to measurements at each temperature? Please specify.

(2) The 40 measurements mentioned in the text were the dark signal measurements performed only during the UV days (11–14 September). However, for the uncertainty estimation, all dark signal measurements performed during the campaign (from 5 to 15 September) were considered. This results in more than 100 dark signal measurements for all Brewers studied. Some Brewers

could perform less dark signal measurements than others due to calibration and necessary maintenance.

(3) The phrase has been corrected to "The number of available measurements depended on the instrument, but, in total, more than 100 dark signal measurements were recorded by each Brewer during the intercomparison campaign".

(1) 242, 245, 477: To which period do these drifts refer?

(2) The term "drift" was used to describe the uncertainty of the radiometric stability. It was determined by calculating the standard deviation from consecutive calibrations, as recommended by Bernhard and Seckmeyer (1999). To do so, all available calibration records from every Brewer were used. Unfortunately, not all Brewers studied had enough calibration files as they had undergone several modifications or were not calibrated frequently enough. Only two Brewers had enough calibration records to derive a standard deviation: Brewers #150 and #185. For Brewer #150, the yearly calibration files from 2005 to 2023 were used. On the other hand, to derive the uncertainty of Brewer #185, the monthly calibration records from 2021 to 2024 were used. No data from prior years have been considered as the instruments had different entrance optics.

(3) To avoid confusion, the term "drift" has been replaced with "uncertainty". Furthermore, the following information has been added: "For Brewer #150, the radiometric uncertainty was derived using the yearly calibration files from 2005 to 2023, while for Brewer #185 the uncertainty was calculated using the monthly calibration files recorded from 2021 to 2024. As mentioned earlier, no data from prior years could be used as the entrance optics of Brewers #150 and #185 were replaced in 2005 and 2021, respectively".

(1) 251: Since the spectral range is defined from short to long wavelengths, the irradiance is increasing rather than declining. You could cope with it more easily by saying "marked variability".

(2, 3) The previous information has been replaced by the term "marked variability".

(1) 269: It appears that not all uncertainties were considered for all instruments. You might consider summarizing in a table the types of uncertainties considered in each instrument.

(2, 3) Following the reviewer's suggestion, the following table has been added at the beginning of Section 3.1:

Table 2. Summary of the uncertainty sources considered for each Brewer under study. Red squares (–) represent the uncertainty sources not included in the evaluation, while green squares (×) indicate those uncertainty sources considered.

| Uncertainty sources considered | Brewer ID | | | | | | | | | |
|---|---|---|---|---|---|---|---|---|---|---|
| | #117 | #150 | #151 | #158 | #172 | #185 | #186 | #202 | #228 | #256 |
| Noise | × | × | × | × | × | × | × | × | × | × |
| Dark signal | × | × | × | × | × | × | × | × | × | × |
| Stray light | × | – | × | – | – | – | – | – | – | – |
| Dead time | × | × | × | × | × | × | × | × | × | × |
| Distance adjustment | × | × | × | × | × | × | × | × | × | × |
| Uncertainty of the reference lamp | × | × | × | × | × | × | × | × | × | × |
| Radiometric stability | × | × | × | × | × | × | × | × | × | × |
| Wavelength shift | × | × | × | × | × | × | × | × | × | × |
| Temperature correction | – | × | – | – | – | – | – | – | – | – |
| Cosine correction | – | × | – | × | – | × | × | – | – | × |

(1) 333-335: Why were spectra corrected for shifts only if they were measured at SZAs<90°?

(2) At larger SZAs, the surface UV irradiance decreases considerably and, as a result, the signal recorded by the instrument is close to its detection threshold. Furthermore, since the "El Arenosillo" Observatory is at sea level, the instability of the atmosphere increases for these larger values of the SZA due to sea turbulence. Therefore, including the UV scans measured at large SZAs would result in unreliable uncertainty estimations.

(3) The previous information has been added to the text as follows: "Larger SZA values have not been considered as the UV irradiance recorded in these conditions is small, close to the detection threshold of the Brewer spectrometers. Furthermore, since the "El Arenosillo" Observatory is at sea level, at large SZAs the instability of the atmosphere increases due to sea turbulence.

(1) I think the second sentence complicates the discussion. You can simply say: "Only spectra recorded at SZA's smaller than 90° were used in this study."

(2, 3) The second sentence has been deleted and replaced with "Only spectra recorded at SZAs smaller than 90° were used in this study".

(1) 359: What about cloudy conditions? Other sources of uncertainty are also involved. Maybe you could discuss briefly what happens under cloudy conditions.

(2) Under cloudy conditions, the methodology for calculating the cosine correction and noise of the spectroradiometer changes. The cosine correction depends on the SZA, the AOD, and the cloudiness. Therefore, cloud cover must be an input introduced in libRadtran when calculating the direct-to-global ratio. On the other hand, noise was derived from stable measurements recorded using a reference lamp. However, clouds have a strong effect on surface UV irradiance and can lead to short-time fluctuations. Under these circumstances, noise needs to be thoroughly characterised by using, for example, a group of near outdoors measurements. Nevertheless, this is difficult for a Brewer spectroradiometer since it takes approximately 5 minutes to record a complete UV spectrum.

(3) The previous information has been added to the text as follows: "Under cloudy conditions, the methodology used for calculating the cosine correction and noise must be adapted. As the cosine correction depends on the cloudiness, the cloud cover must be considered when modelling the direct-to-global ratio. Furthermore, clouds strongly affect the surface UV irradiance and can lead to short-time variations. As a result, noise needs to be thoroughly characterised. For example, by studying groups of near outdoors measurements. For Brewer spectroradiometers, this can be difficult as the instrument does not have enough temporal resolution to detect fast fluctuations of solar UV irradiance".

(1) 371: Good agreement cannot be assessed by this figure. If ratios were shown against one instrument (e.g., the QASUME or the average / median of all instruments) the level of agreement would have been more evident.

(2) Indeed, the agreement between instruments should be assessed using a reference instrument. The following figure has been obtained by studying the ratio from each Brewer to the QASUME reference (2023-09-13 at 14:00 UTC):

[Figure]

It can be seen that the agreement between the instruments is within $\pm 10$ % from unity for wavelengths above 310 nm. Nevertheless, the analysis of the instruments' performance (QA) is beyond the scope of this study (QC).

(3) The phrase "Furthermore, there is good agreement between all the Brewers between 310 and 360 nm" has been deleted from the manuscript.

(1) 378: I am not sure if Figure 2 is necessary. It is expected that the standard uncertainty will increase with wavelength and solar elevation since the irradiance increases. In contrast Figure 3 is more meaningful to discuss.

(2, 3) Following the reviewer's comment, Figure 2 has been deleted from the manuscript and the phrases "As an example, Fig. 2 shows the absolute combined standard uncertainties of the UV irradiances measured on 13 September 2023 at 14:00 UTC (40° SZA). The scale presented is logarithmic to highlight the differences between the Brewers at short wavelengths" have also been removed.

(1) 393: Better say, at the end of the sentence: "... and some Brewers showed almost no SZA dependency, as shown later in Figure 4".

(2, 3) Following the reviewer's suggestion, the phrase has been corrected to "For most Brewers, the relative uncertainty values ranged from 2.5 % to 4 % for wavelengths between 300 and 360 nm and some Brewers showed almost no SZA dependency, as shown later in Figure 4".

(1) 406: In fact, half of the 10 Brewers considered show some SZA dependency and half are not; so, I wouldn't say "most Brewers".

(2, 3) The phrase has been corrected to "Figure 4 shows that the relative combined standard uncertainty of half of the Brewers has no angular dependency". This affirmation has also been corrected in the Conclusions section.

(1) 416: In this section, a table summarizing the ranges of uncertainties due to different factors together with the combined uncertainty would be useful.

(2, 3) The following information has been added to the sensitivity analysis section:

"As a summary, Table 3 shows the relative individual and combined standard uncertainties for each Brewer under study at SZAs below 80° and wavelengths larger than 300 nm. Larger SZAs and shorter wavelengths have not been included in this table as the relative uncertainties increase greatly since the UV irradiance measured approaches zero (see Fig. 1).

Table 3. Range of the uncertainties produced by each uncertainty source individually and the combined standard uncertainty for SZAs below 80° and wavelengths larger than 302 nm for each of the Brewers studied.

| ID | Individual uncertainty (%) | | | | | | | | | | Combined uncertainty (%) |
| --- | --- | --- | --- | --- | --- | --- | --- | --- | --- | --- | --- |
| | Noise | Dark signal | Stray Light | Dead time | Dist. Adjust. | Unc. Lamp | Stab. | λ shift | Temp. Corr. | Cos. Corr. | |
| #117 | 0.070–5.3 | 0.060–21 | ≤2.1 | ≤0.17 | 0.23 | 1.2–1.3 | 3.0 | 0.004–1.6 | – | – | 3.2–22 |
| #150 | 0.029–3.1 | 0.0045–2.3 | – | ≤0.84 | 0.24 | 0.61–0.75 | 3.6 | 0.004–0.82 | 0.0091–0.037 | 0.43–0.51 | 2.9–5.1 |
| #151 | 0.080–2.9 | 0.043–20 | ≤1.1 | ≤0.58 | 0.23 | 0.43–0.61 | 3.0 | 0.014–3.8 | – | – | 3.1–20 |
| #158 | 0.048–2.6 | 0.0013–2.1 | – | ≤0.66 | 0.28 | 0.85–1.4 | 3.0 | 0.014–2.4 | – | 0.80–1.6 | 3.3–5.5 |
| #172 | 0.026–1.3 | 0.012–6.9 | – | ≤0.44 | 0.23 | 0.39–0.67 | 3.0 | 0.015–1.2 | – | – | 3.0–7.8 |
| #185 | 0.073–2.8 | 0.0034–6.4 | – | ≤0.28 | 0.23 | 0.60–1.1 | 2.8 | 0.003–0.63 | – | 0.84–1.6 | 2.4–7.7 |
| #186 | 0.039–2.0 | 0.010–2.6 | – | ≤0.85 | 0.23 | 1.2–1.3 | 3.0 | 0.005–1.2 | – | 1.6–3.3 | 3.6–5.8 |
| #202 | 0.19–6.7 | 0.0022–2.9 | – | ≤1.5 | 0.23 | 0.60–1.1 | 3.0 | 0.01–1.3 | – | – | 3.1–8.0 |
| #228 | 0.042–2.8 | 5.7e-4–0.94 | – | ≤0.20 | 0.23 | 0.60–1.1 | 3.0 | 0.004–1.1 | – | – | 3.1–4.4 |
| #256 | 0.048–2.8 | 6.6e-4–1.3 | – | ≤0.60 | 0.23 | 0.60–1.1 | 3.0 | 0.005–0.68 | – | 0.92–1.8 | 3.2–4.8 |

In the following, the influence of each uncertainty source on the total uncertainty budget will be described in greater detail."

(1) 417-418: The relative contribution to the combined uncertainty is based (or at least it is shown in Figure 5) by the two example spectra. I think it would be more representative to show average contributions from a larger number of spectra recorded for several days in a narrow range of SZAs.

(2, 3) Following the reviewer's comment, Figures 5 and 6 have been updated to show the average contributions of the uncertainty sources over a narrow range of SZAs. This has resulted in the following figures and captions:

[Figure]

Figure 5. Relative contribution of the uncertainty sources of a single monochromator Brewer (#117) to the combined standard uncertainty of the UV spectrum measured at three wavelengths (293. 320, and 360 nm) and two SZAs, (a) 33° and (b) 63°. Each contribution was calculated from the average over a $\pm 1°$ SZA band.

[Figure]

Figure 6. Relative contribution of the uncertainty sources of a double monochromator Brewer (#185) to the combined standard uncertainty of the UV spectrum measured at three wavelengths (293. 320, and 360 nm) and two SZAs, (a) 33° and (b) 63°. Each contribution was calculated from the average over a $\pm 1°$ SZA band.

(1) 424: You might consider including a table with individual and total uncertainties for each of the Brewers considered.

(2) A table with the individual and the combined standard uncertainties has been added to the sensitivity analysis section, as asked earlier in Line 416.

(1) 452: How do your results on Dead Time compare with those derived by Fountoulakis et al., 2016? https://amt.copernicus.org/articles/9/1799/2016/

(2, 3) Following the reviewer's comment, the following information has been added to Section 4.2.4: "Finally, the dead time contribution can be compared to the one reported by Fontoulakis et al. (2016). Their study shows that if the DT ranges from 15 to 45 s and has an error of 2 ns, it leads to irradiance uncertainties of 0.12–0.13, 0.25–0.28, and 0.69–1.13 % for signals of 1, 2, and 5 million counts s$^{-1}$, respectively. These values are similar to the ones found for all Brewers, except Brewer #202, which has an uncertainty larger than 2 ns. For these Brewers, the irradiance uncertainty is less than 0.15, 0.35, and 0.9 % for signals of 1, 2, and 5 million counts s$^{-1}$, respectively".

(1) 506: Caption of Fig. 7: This figure shows the relative standard uncertainty (not the combined). Please correct the axis title and caption.

(2, 3) The axis title and caption of Figure 7 have been corrected so they include the term "relative standard uncertainty" instead of "combined standard uncertainty".

(1) 517-525: Although the contents of this paragraph are correct, they are not very relevant to the topic of the paper. The authors have already published a work on the TCO uncertainties using the same methodology.

(2, 3) The mentioned paragraph and the references therein have been removed from the manuscript.

(1) 556-562: I am afraid that the uncertainty in quantifying the effects of UV radiation on materials is much higher than the uncertainty in UV measurements.

(2, 3) Following the reviewer's comment, the following phrase has been added at the end of the indicated paragraph "Nevertheless, it should be noted that the uncertainty in quantifying the effect of UV radiation on materials can be much higher than the uncertainty in UV measurements itself".

(1) 581-586: Note that photolysis frequencies are estimated from actinic flux measurements which are not measured by Brewer spectroradiometers. However, generally speaking, the estimation of uncertainties of relevant instruments can benefit from this study.

(2, 3) The following information has been added at the end of the indicated paragraph "Although ozone photolysis rates cannot be derived from Brewer spectroradiometers measurements, the methodology used in this study is general and could be of use to other instruments that do measure actinic flux".

(1) 591: In the conclusions section, I suggest discussing the uncertainties against those reported in previous studies. Does this study show significantly different results from the uncertainties in spectral UV irradiance usually quoted in the literature?

(2, 3) Following the referee's suggestion, the uncertainty determined for both single and double Brewers have been compared with those reported in previous studies. The following discussion

has been added to the manuscript: "The relative combined standard uncertainties of the Brewers used in this study can be compared with the ones obtained in previous studies. Garane et al. (2006) determined a combined standard uncertainty of 5.3 % at 320 nm for a single Brewer (MKII version). This value is slightly larger than the one obtained in our work at 320 nm (3.1–3.3 %). This is likely produced by cosine correction. While Garane et al. (2006) included this uncertainty source in their evaluation, none of the single Brewers participating in the RBCC-E campaign had their cosine error characterised. Regarding the double Brewers studied, their UV irradiance uncertainty ranges between 2.5 and 5 % for wavelengths larger than 300 nm. These values are similar to the uncertainty found by Garane et al. (2006). They reported a relative uncertainty of 4.8 % for their double Brewer. Furthermore, the uncertainty of the double Brewers studied is also comparable to the European reference units, QASUME I and QASUME II. Hülsen et al. (2016) found relative uncertainties of 3.85 % and 3.67 % at 300 nm for QASUME and QASUME II, respectively. Moreover, the irradiance uncertainties determined in our work are similar to the ones described in other publications (Bernhard & Seckmeyer, 1999; Fountoulakis et al., 2020). Therefore, the relative combined standard uncertainties determined in this study are comparable to those of other UV spectroradiometers".

(1) 623: Apart from the need to monitor the wavelength shifts, it is essential to reduce them through accurate determination of the instrument's wavelength scale and frequent wavelength calibrations.

(2, 3) The paragraph has been corrected and the following information has been added: "(d) monitor wavelength shifts and reduce them below 0.05 nm through frequent wavelength calibrations and accurate determination of the instrument wavelength scale".

(1) Technical comments:

(1) 30: Rephrase to "measured UV irradiance decreases, the dark signal"

(2, 3) The phrase has been changed to "As the measured UV irradiance decreases, the dark signal, stray light, and noise become dominant".

(1) 99: Replace "using" with "by"

(2, 3) The term "using" has been replaced with "by".

(1) 177: Please rephrase to: "These values are deemed reliable as they were derived by analyzing data from over 20 single Brewers."

(2, 3) The phrase has been modified according to the referee's suggestions.

(1) 191: replace "was" with "is"

(2, 3) The term "was" has been replaced with "is".

(1) 289: Insert "by" before "integrating"

(2, 3) The term "by" has been inserted as indicated.

(1) 294: Replace "diffuser error" with "diffuse error"

(2, 3) The typo has been corrected.

(1) 295: "derived for inhomogeneous sky radiance distribution"

(2, 3) The phrase has been corrected to "derived for inhomogeneous sky radiance distribution".

377: Replace "displaying" with "display"

(2, 3) The term "displaying" has been replaced with "in a similar way than", for greater clarity.

(1) 445: Delete "those of"

(2, 3) The term "those of" has been deleted.

456: Replace "as SZAs decrease" with "at small SZAs"

(2, 3) The term "as SZAs decrease" has been replaced with "at small SZAs".

(1) 603: Replace "depended on the wavelength and SZA, increasing as wavelength rose and SZA declined." with "increases with increasing wavelength and decreasing SZA".

(2, 3) The phrase has been corrected following the referee's suggestion.

(1) 604: Replace "tripled" with "are triple"

(2, 3) The term "tripled" has been replaced with "are three times higher than", for greater clarity.

(1) 624: Replace "committed in" with "associated with"

(2, 3) The term "committed in" has been replaced by "associated with".

---

## Author Comment (AC2)

**Response to Anonymous Reviewer #2**

Authors' response to Reviewer #2 comments on "Evaluation of the uncertainty of the spectral UV irradiance measured by double- and single-monochromator Brewer spectroradiometers".

The answer is structured as follows: (1) comments from Reviewer #2, (2) authors' response and (3) authors' change in the manuscript.

(1) General comments:

(1) The manuscript describes the evaluation of uncertainties of spectral UV irradiance measured by two types of Brewer spectroradiometers. The evaluation is done using measurements from an intercomparison campaign. I think the study and the results are very important for the scientific community, as they are among the first to use the MCM technique to account for the propagation of uncertainties of Brewers. The sensitivity test is used to distinguish the impact of each uncertainty component, which is also a very interesting result and can help the scientific community to improve the quality of measurements.

(2) The authors thank the Reviewer for their careful and constructive examination of the manuscript and reply to all their comments below.

(1) However, I think some issues need to be clarified before the manuscript can be published: There are already WMO-GAW guidelines for quality control (QC) and quality assurance (QA) of UV measurements, which are WMO-GAW Reports No. 146 (Webb et al., 2003, Quality assurance in monitoring Solar UV radiation: the state of the art) and No. 126 (Webb et al., 1998, Guidelines for site quality control of UV monitoring). I think you should discuss more in the Introduction how your study reflects the guidelines of these reports and refer to the "deductive" and "inductive" methods for QA introduced in Webb et al., 2003. You should also refer more closely to Webb et al., 1998. In addition, I think you should consider showing and discussing your uncertainty estimate and the results of the intercomparison campaign. You have the ideal setup of having the QASUME intercomparison results from the El Arenosillo campaign. Thus, you could compare the results of your "deductive" method with the results of the "inductive" method.

(2) The results of the intercomparison campaign (UV index, wavelength shift, daily global irradiance ratio, and its daily variation) are published in the report elaborated by PMOD/WRC (Hülsen, 2023). Therefore, it would be redundant to include them in this study. Nevertheless, the authors agree that it is interesting to include the uncertainty evaluation in the comparison to the QASUME. As a result, an uncertainty has been derived for the Brewer/QASUME ratio by combining the irradiance uncertainty of the QASUME (Hülsen et al., 2016) and each of the Brewers studied. Moreover, references to WMO-GAW reports No.146 and No. 126 have been added throughout the manuscript (especially in the introduction and methodology sections).

(3) Information regarding the QA methods proposed by Webb et al. (2003) has been added in the introduction section as: "On the other hand, QA can be performed using two methods (Webb et al., 2003). In the first one (inductive), the instrument's performance is assessed through intercomparison campaigns. As for the second (deductive) method, the user deduces the instrument's quality through a meticulous description of the calibration process as well as the instrumental characteristics, such as its linearity and angular response. For QA purposes, the general principles established by Webb et al. (1998) should be followed, expanded, and refined, so the user can report reliable uncertainties for any measurement, not limiting the analysis for a typical measurement at the station (Webb et al., 2003)". Moreover, the QA used in the RBCC-E

campaign has been described to reflect the guidelines of Webb et al. (2003). In this way, the following information has been added in line 60: "The QA performed for the instruments used in this work corresponds to the inductive method described by Webb et al., (2003). It is carried out during the campaigns performed by the Regional Brewer Calibration Center–Europe (RBCC-E) where Brewer spectrophotometers are compared to the European reference spectroradiometer, the QASUME unit (e.g. Gröbner et al., 2010; Lakkala et al., 2008). These intercomparison campaigns meet the main requirements laid out by Webb et al. (2003), i.e. transparency and objective comparison algorithms (see the campaign reports at the PMOD/WRC website, https://www.pmodwrc.ch/en/world-radiation-center-2/wcc-uv/qasume-site-audits/, the report of the 18th intercomparison campaign, Hülsen, 2023, and an overview of the EuBrewNet's algorithms, López-Solano, 2024)".

As for the QC guidelines, in line 82 (introduction), a comment has been added to reflect the work of Webb et al. (1998): "All the necessary uncertainty sources considered by Webb et al. (1998) have been included in the uncertainty evaluation presented in this work, plus some highly recommended and additional sources (such as stray light, alignment, or wavelength accuracy)". Furthermore, this study has also been mentioned in the methodology section to reflect their findings regarding radiometric stability and the current of the reference lamp. In this way, in line 221 the following information has been added: "Based on the findings of Webb et al. (1994), the standard practice is to assume that a 1 % change in the current of the reference lamp leads to a 10 % change in the spectral irradiance measured by the instrument (e.g. Bernhard and Seckmeyer, 1999; Webb et al., 1998)". Moreover, the results obtained in former Section 4.2.5 (distance alignment) have been compared with the ones determined by Webb et al. (1998): "According to Webb et al. (1998), if the nominal distance is d and its uncertainty $u_d$, the percentage uncertainty can be calculated using the inverse square law ($1/r^2$, where $r$ is the distance between lamp and instrument) as $[(d + u_d)^2 - d^2] * 100 / d^2$. Therefore, the previous results agree with the formula proposed by Webb et al. (1998)". Additionally, the calibration guidelines established by Webb et al. (1998) have also been mentioned in the sensitivity analysis in line 473 as: "This agrees with the recommendations of Webb et al. (1998). They suggest calibrating the instruments using three reference lamps".

Finally, a section regarding the uncertainty ratio between the QASUME and the instruments has been added to the Results section. A new figure and table have been added to the manuscript as follows:

"The corrections applied to the measured irradiance (described in Section 3.2) are recommended by numerous studies to improve the quality of the measurements (e.g. Fountoulakis et al., 2016b; Garane et al., 2006; Kerr, 2010; Lakkala et al., 2008, 2018). This was also verified during the 18th RBCC-E campaign, as the results show that including the cosine correction improves considerably the comparison to the QASUME (Hülsen, 2023). Although the campaign report shows the ratio of each participating Brewer to the QASUME (see Hülsen (2023)), it is interesting to represent the ratio of all studied Brewers together. In this way, Fig. 4 displays the global irradiance ratio to the QASUME obtained from dividing the irradiances shown in Fig. 1 to the irradiance recorded by the QASUME unit.

[Figure]

**Figure 4. Global irradiance ratio to the QASUME recorded on 13 September at 14:00 UTC. (a) First group (double Brewers with cosine correction). (b) Second group (two single and three double Brewers with no cosine correction implemented).**

Figure 4 shows the effectiveness of the cosine correction, since Brewers with such correction implemented (Fig. 4a) report irradiances more similar to the one measured by the QASUME. Nevertheless, the agreement between all Brewer spectrophotometers and the QASUME is within 10 % for wavelengths above 310 nm.

Furthermore, the irradiance uncertainty found for each Brewer in the previous section can be used to derive the uncertainty of their ratio to the QASUME. Table 3 summarises the combined standard uncertainty of the average Brewer/QASUME ratio measured on 13 September at three different wavelengths. These uncertainties were computed by combining the irradiance uncertainty of each Brewer and the one from the QASUME, provided by Hülsen et al. (2016).

**Table 3. Number of simultaneous scans, mean ratio to the QASUME and its combined standard uncertainty (both absolute and relative) determined between 310 and 360 nm on 13 September.**

| Brewer ID | N | Ratio to the QASUME (310–360 nm) | | |
|:---:|:---:|:---:|:---:|:---:|
| | | Mean value | Combined standard uncertainty | Relative standard uncertainty (%) |
| #117 | 19 | 0.927 | 0.034 | 3.7 |
| #150 | 20 | 1.035 | 0.035 | 3.4 |
| #151 | 24 | 0.914 | 0.033 | 3.6 |
| #158 | 17 | 0.972 | 0.036 | 3.7 |
| #172 | 19 | 0.947 | 0.033 | 3.5 |
| #185 | 18 | 0.978 | 0.030 | 3.1 |
| #186 | 15 | 1.003 | 0.043 | 4.3 |
| #202 | 19 | 0.928 | 0.033 | 3.6 |
| #228 | 19 | 0.937 | 0.033 | 3.5 |
| #256 | 19 | 1.003 | 0.037 | 3.7 |

Table 3 shows that only those Brewer spectrophotometers with a cosine correction implemented (#150, #158, #185, #186, and #256) include the ideal value of the ratio (unity) within their uncertainty interval. The remaining Brewers underestimate the UV irradiance and deviate from unity. This is likely caused by the cosine and temperature errors of the instruments, which couldn't be corrected (there was no available information regarding their characterisation). Therefore, to improve the performance of these uncorrected Brewers these two sources must be characterised and corrected".

(1) Another concern is the uncertainty associated with cosine correction. I don't understand why you get such higher uncertainties in measurements that are cosine corrected compared to those

that are not. Shouldn't it be the other way around? If the cosine correction is in the range of 5 to 10%, then the measurements missing that correction should be at least that far off and have higher uncertainties? And, as mentioned in the specific comments, please include the information of which part of the cosine correction (angular response characterization, dir/diff ratio determination...?) has the largest uncertainty. Maybe I missed something: please explain this more clearly.

(2) It is important to differentiate between an error and the uncertainty associated with its correction. If the spectral irradiance measured is not corrected for the instrument's cosine error, it will have a larger error (see the Figure 4 in the response to the previous comment). In fact, the comparison campaign showed that correcting the Brewer cosine error improved the global irradiance ratio to the QASUME (Hülsen, 2023). On the other hand, the calculation of the cosine correction factor has an uncertainty, which needs to be taken into account in the uncertainty analysis. Nevertheless, this uncertainty can only be determined if the cosine error is characterised. That is why the Brewers with no cosine correction have smaller uncertainties, because this uncertainty source could not be considered in their evaluation. As a result, the uncertainty evaluation of these Brewers is limited, as the cosine error is not included in the analysis.

Regarding the contributions to the cosine correction (angular characterisation, direct to global ratio, etc.), it wasn't considered in the original manuscript. To analyse the influence of every part of the cosine correction, the sensitivity analysis has been broadened to evaluate the uncertainty produced by these factors on their own.

(3) To clarify this issue, the definition of uncertainty and error used has been included in the methodology section as follows: "In the following, the term "error" will denote the imperfections in a measurement result, while the term "uncertainty" will be used to reflect the existing doubt regarding the value of the measured spectral UV irradiance". Furthermore, the limitation of the uncertainty evaluation of Brewers with no cosine correction has been addressed in the results section (line 363) by adding the following information "The uncertainty evaluation of the Brewer spectrophotometers in this second group is limited as it is missing one of the key uncertainty sources in solar radiometry, cosine correction. As a result, the uncertainties determined are likely an underestimation. Nevertheless, these estimations represent the uncertainty of the spectral irradiance reported by most of the participating Brewer spectrophotometers". Finally, the contributions of the cosine correction have been added in former Section 4.2.10 (sensitivity analysis of cosine correction) as "The uncertainties shown in Fig. 7 are mostly produced by the uncertainties of the of the diffuse ($f_\mathrm{d}$) and direct ($f_\mathrm{r}$) cosine errors. In fact, these factors account for more than 98 % of the total irradiance uncertainty caused by cosine correction. As for the direct to global irradiance ratio, its impact on the uncertainty budget is negligible as only cloud-free conditions have been considered in the analysis. This would likely change if overcast or mixed sky conditions were to be included in the uncertainty evaluation. Therefore, under cloud-free conditions, the main sources of uncertainty in the cosine correction are the errors committed in the angular characterisation and in the assumption of isotropic sky radiance to calculate the diffuse cosine error".

(1) Then comes the transfer of the irradiance scale from an accredited lab to the instrument itself (which is basically the spectral response determination). There are uncertainties in the transfer, which you've described well, but it's unclear how you've accounted for them in your analysis. I wonder if when you do the lamp calibration (transfer of the irradiance scale), you also have to

take into account the uncertainties in the Brewer irradiance measurements, don't you (in addition to the distance from the lamp: levelling, dead time, noise, etc....)?

(2) During the calibration, the Brewer spectroradiometer records the signal measured under the lamp four times. As the reviewer states, this signal is affected by several uncertainty sources such as dead time, noise, dark signal, and stray light (especially if the Brewer is a single-monochromator spectrometer). When determining the response of the instrument, the corrected signal (for dead time, dark signal, etc.) is used. This is not mentioned in Section 3.1.2, as it is focused on the sources of uncertainty produced during the calibration (distance alignment, uncertainty of the reference lamp, etc.), while the signal uncertainty sources are described earlier, in Section 3.1.1.

(3) To clarify this issue, the following information has been added to Section 3.1.2 at line 219: "Then, the signal under the lamp is recorded several times and corrected for dark counts, dead time, and stray light (see Section 3.1.1.). The responsivity of the instrument is derived by dividing the corrected signal by the irradiance of the reference lamp. However, this responsivity is also affected by other sources of uncertainty produced during the radiometric calibration such as the distance adjustment between the lamp and the diffuser, the radiometric stability, and the uncertainty of the spectral irradiance emitted by the reference lamp".

(1) And about stability, changes in response over time: I think you should discuss this as an additional source of uncertainty for long-term monitoring. Not for newly calibrated instruments. And add a time specification. That you assume for example a 1-3% drift in response per year. The drift is really instrument-dependent and depends on modifications/maintenance/upgrades of each instrument. As stated in Webb et al. 1998, "Each time the calibration is checked or changed, the data gathered between the two calibration times has an uncertainty which depends on the difference between the two calibrations, and it is these uncertainties which should be used for this calculation".

(2) The term "drift" was used to describe the uncertainty of the radiometric stability. It was determined by calculating the standard deviation of the differences between consecutive calibrations, as recommended by Bernhard and Seckmeyer (1999). Webb et al. (1998) recommend a similar approach. They characterise the stability by deriving the root mean square (rms) uncertainty over a significant period of time. Therefore, short- and long-term radiometric stability can be characterised as long as Brewer spectroradiometers are calibrated frequently enough to ensure a reliable standard deviation (or rms uncertainty). However, most of the Brewers used in this study are usually calibrated once a year (or even once every two years). Consequently, no short-term monitoring uncertainty could be derived and the radiometric stability was determined using all available calibration records. Moreover, this uncertainty depends on the instrument as well as on its modifications and maintenance. This further complicates the determination of the radiometric stability for each Brewer as most of them have been operating for several years and have undergone many modifications throughout this period. Therefore, to include this uncertainty source in the uncertainty evaluation, a 3 % uncertainty was assumed for most Brewers, in agreement with the findings of Garane et al. (2006) and Lakkala et al. (2008).

On the other hand, the newly calibrated instrument used in this study had been calibrated five times between its deployment and the RBCC-E campaign (1.5 years). During its first year in operation, the response of this Brewer decreased significantly, showing a deviation larger than 10 %. Then, it gradually stabilised. Therefore, assuming a 3 % uncertainty for this Brewer as well as the older ones, might not be such a large overestimation of its stability.

(3) To avoid confusion, the term "drift" has been replaced with "uncertainty". Furthermore, lines 236–238 have been modified to add the previous information as: "To characterise the radiometric stability of every Brewer, several studies recommend studying the difference between consecutive calibration factors over a significant period (e.g. a year) (Bernhard and Seckmeyer, 1999; Webb et al., 1998). These methods require that the instrument is calibrated frequently to derive reliable statistics". Furthermore, in line 244 a clarification regarding the assumption of a 3 % uncertainty for the newly calibrated Brewer has been added: "It should be noted that this value, derived from long-term monitoring, might not be a large overestimation for the newly calibrated Brewer (#256), as this instrument showed large instabilities during its first year of operation".

(1) Specific comments:

(1) line 157: Please specify in which section you describe which of the three uncertainty sources. Check that you make clear in text what do you mean by the three different uncertainty sources.

(2, 3) The sections for each of the uncertainty sources have been added to the manuscript. Furthermore, at the beginning of each section, the uncertainty sources have been listed and then described in detail. Moreover, following the suggestion made by Reviewer #1, the following table summarising the uncertainty sources considered has also been added to the text.

**Table 2. Summary of the uncertainty sources considered for each Brewer under study. Red squares (−) represent the uncertainty sources not included in the evaluation, while green squares (×) indicate those uncertainty sources considered.**

| Uncertainty sources considered | Brewer ID | | | | | | | | | |
|---|---|---|---|---|---|---|---|---|---|---|
| | #117 | #150 | #151 | #158 | #172 | #185 | #186 | #202 | #228 | #256 |
| Noise | × | × | × | × | × | × | × | × | × | × |
| Dark signal | × | × | × | × | × | × | × | × | × | × |
| Stray light | × | − | × | − | − | − | − | − | − | − |
| Dead time | × | × | × | × | × | × | × | × | × | × |
| Distance adjustment | × | × | × | × | × | × | × | × | × | × |
| Uncertainty of the reference lamp | × | × | × | × | × | × | × | × | × | × |
| Radiometric stability | × | × | × | × | × | × | × | × | × | × |
| Wavelength shift | × | × | × | × | × | × | × | × | × | × |
| Temperature correction | − | × | − | − | − | − | − | − | − | − |
| Cosine correction | − | × | − | × | − | × | × | − | − | × |

(1) lines 168-169: "However, deriving the uncertainty of this method for the Brewers under study is difficult, as it would use the information from only five wavelengths (from 290 to 292 nm)." -> I don't see your point. Uncertainty related to this approach could also be determined, e.g., compared to the QASUME. I think this sentence should be rethought.

(2) As stated by the referee, the comparison against the QASUME can be used to derive an uncertainty. At first, this approach was discarded as it would also consider, besides the stray light effects, the temperature and the cosine error of the instrument (single Brewers are not corrected for these two error sources). Nevertheless, the effect of these two sources is expected to be small below 292 nm. As a result, the uncertainty determined in this way is not a huge overestimation of the uncertainty produced by stray light correction. Furthermore, since the stray light was estimated during the RBCC-E campaign using the five-wavelength method, the uncertainty should be estimated using the suggested approach (comparison to the QASUME) instead of the one proposed by Savastiouk et al. (2023).

(3) Following the comments made by Reviewers #2 and #3, the methodology of Savastiouk et al. (2023) has been replaced with the one performed in the 18th RBCC-E intercomparison campaign. Therefore, the description in lines 168–184 has been changed to "The uncertainty of this method was estimated by comparing the corrected irradiance to the QASUME from 290 to 292 nm. This estimation also includes the effects of temperature and cosine errors since the single Brewers under study are not corrected for these two sources of error. Nevertheless, since the effect of these two sources is expected to be small below 292 nm, the uncertainty determined might be only a slight overestimation. Furthermore, the standard deviation from the measurements of the five wavelengths (from 290 to 292 nm) was also derived and combined with the uncertainty obtained from the QASUME comparison".

(1) line 229: "Brewer #150, on the other hand, has an additional source of uncertainty since the position of its diffuser's reference plane needs to be determined as well (González et al., 2023)"-> I don't understand the sentence, as the previous sentence describes as distance 500+-0.6mm, which is less than the "standard" one of 500+-1mm explained earlier.

(2) The indicated phrase can be confusing as it is comparing uncertainty and precision errors. The ruler used to adjust the distance has a precision of 1 mm, which translates to an uncertainty of 0.58 mm according to the GUM (Guide to the expression of Uncertainty in Measurement). This guide recommends that If the only available information are the lower $x_-$ and upper $x_+$ limits of a variable $x$, the uncertainty should be calculated as:

$$u^2(x) = (x_+ - x_-)^2/12 \,.$$

In our case, the lower and upper limits are 1 mm, which results in an uncertainty of 0.58 mm, according to the previous equation. However, Brewer #150 has an additional source of uncertainty as a result of the experimental determination of its diffuser plane, 0.15 mm (González et al., 2023). Combining both uncertainties (0.6 mm and 0.15 mm) leads to an uncertainty of 0.59 mm, slightly larger. As a result, the irradiance uncertainty produced by the distance adjustment is slightly larger for Brewer #150 (0.24 %) than for the other instruments (0.23 %).

(3) The previous information has been added to the text as follows: "According to the GUM, this precision error translates in an uncertainty of 0.58 mm (BIPM et al., 2008a)". As for the uncertainty of Brewer #150, it has been described as "Brewer #150, on the other hand, has an additional source of uncertainty since the position of its diffuser's reference plane needs to be determined as well (González et al., 2023), resulting in an uncertainty of 0.59 mm".

(1) line 230:"Regarding the uncertainties of the irradiances of the reference lamps, there is no need to determine them since all lamps used during the campaign had been previously calibrated in different standard laboratories." -> I suggest skipping this sentence or rephrasing it.

(2, 3) Following the reviewer's comment, the phrase has been deleted from the text.

(1) line 255: Please add a short sentence describing the idea of SHICRIVM wavelength shift detection.

(2, 3) Noted. The following information regarding SHICrivm has been added to the manuscript: "This software estimates the wavelength shift by comparing the structure of the spectrum measured by the ground-based instrument with the extraterrestrial spectrum. The latter is simulated using the SUSIM extraterrestrial spectrum (Slaper et al., 1995)".

(1) line 260: "Furthermore, there is a second contribution to the wavelength misalignment, the precision of the micrometre, i.e. the system setting the wavelengths measured by a Brewer spectroradiometer". -> yes, it contributes to the wavelength misalignment, but the impact is seen in the wavelength shift recorded by SHICRIVM. Please rephrase.

(2, 3) Following the reviewer's comment, line 260 has been rewritten as "the shifts determined by SHICrivm include the wavelength misalignment produced by the precision of the micrometre, i.e. the system setting the wavelengths measured by a Brewer spectrophotometer. This precision is approximately 8 pm (Gröbner et al., 1998)".

(1) Figure 6: I think you should also include the uncertainty related to stray light, even if the double monochromators are known to have less stray light than single ones.

(2) To our knowledge, no study of stray light in double Brewer exists. Savastiouk et al. (2020) tried to estimate it by looking at the sulphur dioxide retrievals at large ozone slant column densities. Their test suggests that the correction factor for stray light in MkIII Brewers is an order of magnitude lower than for single Brewers. Therefore, the level of stray light is very low and its contribution is likely lower than dark counts or noise at short wavelengths.

(1) Section 3.2 UV model -> Is it really a model? Couldn't the Chapter be called "UV processing algorithm"?

(2, 3) Following the reviewer's suggestion, the title of the chapter has been changed to "UV processing algorithm".

(1) line 465: What about the wavelength dependency of the distance-related uncertainty? The irradiance of the calibration lamp is wavelength-dependent, isn't it? So the effect of the distance measurement error is wavelength-dependent, too?

(2) The distance alignment and the irradiance of the reference lamp are two different uncertainty sources. In Line 465 only the effect of the distance adjustment is shown and no wavelength dependency is observed. On the other hand, if the uncertainty of the reference lamp is the only error source considered, it does lead to wavelength-dependent irradiance uncertainties, as described in Section 4.2.6. The distance alignment does not produce wavelength-dependent uncertainties as the distance between the lamp and the diffuser is sufficiently large and the lamp can be regarded as a point source. As a result, the UV irradiance decreases with distance following the inverse square law:

$E'(\lambda) = E(\lambda)/d^2$.

(3) The previous information has been added to the text as follows: "Furthermore, the uncertainty produced by the distance adjustment has no spectral dependency. Since the reference lamp can be regarded as a point source, the UV irradiance follows the inverse square. Therefore, a change in distance has the same effect on all wavelengths measured".

(1) Line 475 and 4.2.7. Radiometric stability: I think you should report the uncertainties at the moment of a fresh calibration. If I understand right, you assume some drift in the spectral responsivity of the Brewer, which is based on measurements over several years. The instruments drift at a different speed, and the spectral response is strongly dependent on instrumental modifications.

(2) To derive the uncertainty of the radiometric stability, it is recommended to study the difference between consecutive calibrations over a significant period (e.g. a year) (Webb et al., 1998; Bernhard and Seckmeyer, 1999). Although the Brewers are calibrated during the intercomparison campaign, this calibration file alone is not enough to derive a reliable uncertainty estimate. As a result, we checked the calibration records of all Brewers used in this study. Nevertheless, since the instruments are calibrated once every 1–2 years and they have undergone several modifications, it was impossible to calculate a reliable standard deviation or rms uncertainty, as recommended by Bernhard and Seckmeyer (1999) and Webb et al. (1998), respectively. Therefore, to include this source in the uncertainty analysis, an uncertainty was estimated from the literature (a 3 % uncertainty). To our knowledge, there are no studies that have reported radiometric uncertainties at the moment of a fresh calibration for Brewer spectrophotometers. In the sensitivity analysis, it was observed that radiometric stability caused irradiance uncertainties of 3 % (no other uncertainty sources were considered for this analysis). This was expected as the UV irradiance is inversely proportional to the responsivity (as shown in Eq. (8)), and a 3 % uncertainty was assumed for the latter.

(3) To clarify this issue, the methodology and the results sections have been modified. In this way, the calculation of the radiometric uncertainty in the methodology section has been written in greater detail, indicating the data period used and highlighting that the instrument modifications greatly affect the responsivity of the instrument. Besides the information shown in the response to the general comments, the following information has been added to the text: "For Brewer #150, the radiometric uncertainty was derived using the yearly calibration files from 2005 to 2023, while for Brewer #185 the uncertainty was calculated using the monthly calibration files recorded from 2021 to 2024. As mentioned earlier, no data from prior years could be used as the entrance optics of Brewers #150 and #185 were replaced in 2005 and 2021, respectively". On the other hand, lines 476–477 from Section 4.2.7 have been rewritten for greater clarity as: "For most Brewers, this uncertainty source leads to irradiance uncertainties of 3 %. This was expected, since the UV irradiance is inversely proportional to the responsivity (as shown in Eq. (8)), and a 3 % uncertainty in the responsivity was assumed for most Brewers, in agreement with the findings of Garane et al. (2006) and Lakkala et al. (2008) (see Section 3.1.2.). On the other hand, Brewers #150 and #185 had their instability characterised using their calibration records and reported irradiance uncertainties of up to 3.6 % and 2.5 %, respectively".

(1) Section 4.2.10 Cosine correction: What is the reason for the high uncertainty related to cosine correction? Is it due to the uncertainty in angular response characterization or the DIR/DIFF ratio or something else? Please include this information in the text.

(2) It is mostly produced by the errors committed in the angular response characterisation and the assumption that the sky radiance is isotropic (calculation of the diffuse cosine error). The DIR/GLO ratio has no impact in the uncertainty budget since only cloud-free conditions have been considered in this study. If cloudy conditions were to be included, this ratio would likely become a significant source of uncertainty.

(3) The previous information has been added to section 4.2.10 as follows: "The uncertainties shown in Fig. 7 are mostly produced by the uncertainties of the diffuse (fd) and direct (fr) cosine errors. In fact, these factors account for more than 98 % of the total irradiance uncertainty caused by cosine correction. As for the direct to global irradiance ratio, its impact on the uncertainty budget is negligible as only cloud-free conditions have been considered in the analysis. This would likely change if overcast or mixed sky conditions were to be included in the uncertainty

evaluation. Therefore, under cloud-free conditions, the main sources of uncertainty in the cosine correction are the errors committed in the angular characterisation and in the assumption of isotropic sky radiance to calculate the diffuse cosine error".

(1) line 520: But the Brewer ozone processing is not based on UV measurements you describe in this study. I don't see the point of this paragraph.

(2, 3) Following the reviewers' comments, the indicated paragraph has been deleted from the manuscript.

(1) line 534: Do you mean combined uncertainty?

(2, 3) Yes. Nevertheless, following the comment of Reviewer #3, this section has been summarised and the indicated line has been removed from the text.

(1) line 573: For global radiation?

(2, 3) Yes, for global UV radiation. The previous information has been added to the manuscript as follows: "More advanced developments, such as QASUMEII, have further improved accuracy, with a combined uncertainty for global UV measurements of 1.01 % between 310 and 400 nm and 3.67 % at 300 nm".

(1) line 582: I refer to my earlier comment that, at least for the Brewer, the ozone is not calculated using the UV irradiances which you describe in this manuscript.

(2, 3) Following the comments made by Reviewer #1 and #2, the paragraph regarding tropospheric ozone (former lines 582–586) has been removed from the manuscript.

(1) line 585: I don't see the connection between your work and the last sentence of this paragraph.

(2, 3) We agree with Reviewers #1 and #2 that tropospheric ozone is not calculated from Brewer UV measurements and that former lines 582–586 should be removed.

---

## Author Comment (AC3)

**Response to Anonymous Reviewer #3**

Authors' response to Reviewer #3 comments on "Evaluation of the uncertainty of the spectral UV irradiance measured by double- and single-monochromator Brewer spectroradiometers". The authors thank the Reviewer for the careful and constructive examination of the manuscript and reply to all their comments below.

The answer is structured as follows: (1) comments from Reviewer #3, (2) authors' response and (3) authors' change in the manuscript.

**(1) GENERAL COMMENTS**

(1) The study by González et al. builds upon previous work by the same authors (https://doi.org/10.1029/2023JD039500), extending the methodology applied to Brewer #150 to a broader set of Brewer spectrophotometers participating in an RBCC-E campaign. The authors discuss various factors influencing the uncertainty in spectral UV measurements and estimate the overall uncertainty using a Monte Carlo approach. A notable limitation, however, is that approximately half of the instruments were not fully characterized, for instance lacking angular response measurements. An additional contribution beyond the 2023 study is the inclusion of stray light effects, addressed here using a method originally developed for direct-sun observations and never demonstrated to work on global UV irradiance.

(2, 3) A statement has been added to the text to acknowledge the incomplete characterisation of several instruments, and, as a result, their limited uncertainty evaluation. Moreover, this limitation has become apparent when comparing these instruments with the QASUME unit (the comparison and its uncertainty has been added in Section 4, Results). As for the stray light, the methodology has been modified to incorporate the standard correction implemented in EuBrewNet (five-wavelength method). This method has also been used in previous uncertainty evaluation studies for both UV and AOD measurements (Arola and Koskela, 2004; Garane et al., 2006).

(1) The general topic is of high relevance to both the Brewer user community and the broader scientific community, particularly in the context of long-term UV trend detection related to ozone variability and climate change. Nevertheless, the manuscript contains several inaccuracies and errors concerning the description of the Brewer instrument. Furthermore, the methodology is presented in a confusing manner, frequently confounding systematic effects with the uncertainties introduced by their corrections. The improper distinction between systematic and random effects appears to result in absurd outcomes, for example, an estimated uncertainty in the UV Index of only 0.28–0.53%, which is significantly lower than the stated calibration uncertainty of the irradiance standard lamps used for Brewer calibration.

(2, 3) The inaccuracies and errors found in the description of the Brewer spectrophotometer have been corrected (Section 2 of the manuscript). The methodology section has been corrected as well, to specifically include the difference between errors and the uncertainties produced by their correction. As for the results obtained, the calculation of the UV index was revised, finding that it was wrongly implemented. Therefore, the UV index has been recalculated, now reporting uncertainties of 2.7–6.2 %. Furthermore, to ensure the credibility of the uncertainties estimated in this work, they have been compared to the ones found in previous studies. In this way, the combined standard uncertainty, UV index, and the uncertainties produced by the sources of error characterised in this study are similar to the ones reported in other Brewer spectrophotometers and UV spectroradiometers (Section 6 of the manuscript).

(1) I strongly recommend against publication of the manuscript in its current form, as it risks disseminating incorrect and potentially misleading information to the user community.

(2) Following the reviewer's comment, the manuscript has been corrected and improved. Moreover, the methodology has been revised and the uncertainty estimations recalculated, which has led to irradiance, UV index, and erythemal uncertainties similar to those found in previous studies.

**(1) SPECIFIC COMMENTS**

**(1) 1. Scope of the paper**

(1) The scope and applicability of the results remain unclear. For instance, it appears that the calculated uncertainties pertain to measurements performed during the RBCC-E campaign. However, a long-term drift in Brewer responsivity of 3 % is then mentioned: clarification is needed on whether this drift refers to a time span that includes or exceeds the duration of the campaign. Additionally, it is not clear whether the stray light correction method described in the manuscript was actually implemented during the campaign, or if the cosine correction was actually applied to the data described in the campaign report.

(2) The measurements (and their corrections) considered in this study were performed during the 18th RBCC-E campaign. This includes stray light for single Brewers as well as cosine and temperature correction for those double Brewers that had such sources characterised. The cosine correction applied improved considerably the comparison to the QASUME, as stated in the campaign report (Hülsen, 2023). However, there is one uncertainty source considered that was characterised using the calibration data set of Brewer spectrophotometers: radiometric stability. Several studies recommend characterising the instability of an instrument by studying the difference between consecutive calibrations (either by deriving the standard deviation or the rms uncertainty) (Webb et al., 1998; Bernhard and Seckmeyer, 1999). These methods require the instrument to be calibrated frequently so as to derive appropriate statistics. During the 18th RBCC-E campaign, Brewers were calibrated once using one or two 1000 W lamps. Therefore, no reliable standard deviation can be obtained from the calibrations taken during the campaign. Since most of the Brewers used in this study are usually calibrated once a year (or even once every two years), no short-term monitoring uncertainty could be derived either. As a result, all available calibration records were considered. Furthermore, the radiometric uncertainty depends on the instrument as well as on its modifications and maintenance. This further complicated the determination of the radiometric uncertainty for each Brewer as most of them have been operating for several years and have undergone many modifications throughout this time. In the end, only two Brewers had enough calibration records to derive this uncertainty: Brewer #185 and #150. For the remaining Brewers, a 3 % uncertainty was assumed, in agreement with the findings of Garane et al. (2006) and Lakkala et al. (2008).

(3) To clarify this issue, the time when measurements were performed as well as their corrections and uncertainty characterisation has been clarified in the methodology section. As a result, in Line 145 (beginning of the methodology section) the following information has been added: "The UV measurements used for the uncertainty evaluation were performed during the 18th RBCC-E intercomparison campaign (described in detail in Section 2.2.)". Moreover, an explanation regarding the corrections and their uncertainties have been added in Section 3.1 (characterisation of the uncertainty sources): "All these sources and their corresponding corrections have been applied to the Brewer UV measurements during the 18th RBCC-E campaign, whenever possible.

As will be described in the following, some of the participating Brewers were not fully characterised, lacking information regarding their temperature and cosine correction. The uncertainty sources included in the uncertainty evaluation of each Brewer are summarised in Table 2.

Table 2. Summary of the uncertainty sources considered for each Brewer under study. Red squares (−) represent the uncertainty sources not included in the evaluation, while green squares (×) indicate those uncertainty sources considered.

| Uncertainty sources considered | Brewer ID | | | | | | | | | |
|---|---|---|---|---|---|---|---|---|---|---|
| | #117 | #150 | #151 | #158 | #172 | #185 | #186 | #202 | #228 | #256 |
| Noise | × | × | × | × | × | × | × | × | × | × |
| Dark signal | × | × | × | × | × | × | × | × | × | × |
| Stray light | × | − | × | − | − | − | − | − | − | − |
| Dead time | × | × | × | × | × | × | × | × | × | × |
| Distance adjustment | × | × | × | × | × | × | × | × | × | × |
| Uncertainty of the reference lamp | × | × | × | × | × | × | × | × | × | × |
| Radiometric stability | × | × | × | × | × | × | × | × | × | × |
| Wavelength shift | × | × | × | × | × | × | × | × | × | × |
| Temperature correction | − | × | − | − | − | − | − | − | − | − |
| Cosine correction | − | × | − | × | − | × | × | − | − | × |

Furthermore, the data used to determine the stability uncertainty has been provided in line 244 (Section 3.1.2): "On the other hand, the uncertainty assumed (3 %) is also similar to the average uncertainty found for Brewers #150 and #185, of 2.9 % and 2.4 %, respectively. For Brewer #150, the radiometric uncertainty was derived using the yearly calibration files from 2005 to 2023, while for Brewer #185 the uncertainty was calculated using the monthly calibration files recorded from 2021 to 2024. As mentioned earlier, no data from prior years could be used as the entrance optics of Brewers #150 and #185 were replaced in 2005 and 2021, respectively".

**(1) 2. Inaccuracies and errors in the description of the Brewer spectrophotometer**

- **(1) Line 97: Global spectral UV radiation reaches the foreoptics after two prism reflections (UV-B prism and zenith prism), not one**

  (2, 3) The phrase has been corrected to: "global UV irradiance enters through the entrance optics, consisting of a Teflon diffuser covered by a quartz dome, and is redirected into the fore-optics using two prisms (UV-B and zenith prisms)".

- **(1) Lines 97–98: Filter Wheel #3 is typically set to the open position**

  (2) Filter Wheel #3 contains different filters for each measuring mode: (1) a UG-11/NiSO4 filter for ozone mode, (2) a BG-12 filter for NO2 mode of operation, and (3) a UG-11 filter for UV measurements (Kipp & Zonen, 2007). Maybe the reviewer is referring to Filter Wheel #1, which is set to the open position for UV measurements for both types (MkIII and MkIV) of Brewer spectrophotometers (Kipp & Zonen, 2007, 2018).

  (3) The following description of the three filter wheels has been added to the manuscript: "The first filter wheel has an open hole (open position) for UV measurements, a ground-quartz disk (pos. 1) for direct-Sun measurements and an opaque disk (pos. 2) for dark signal tests (Kipp & Zonen, 2007, 2018). The second filter wheel contains five neutral

density filters to adjust the intensity level of the incoming light. After passing the filter wheels, the light is focused onto the spectrometer". Then, after describing the spectrometer and the slit mask (see the response to the specific comments), Filter Wheel #3 has been described as: "In MkIV Brewers, the emerging light passes through a third filter wheel, which has several filters to block undesired radiation: (1) in the ozone mode, a UG-11/NiSO4 filter combination is used, (2) in the UV mode, the filter switches to a UG-11 filter, and (3) in the NO2 mode, a BG-12 filter is used to block UV radiation (Kipp & Zonen, 2007)".

- (1) Line 98: Filter wheels are mechanical components used to position optical elements, but are not optical elements themselves

(2, 3) The discussion regarding the filter wheels has been separated from the one regarding the optical elements. As a result, line 98 has been rewritten as: "The incoming radiation is then focused and collimated by the Iris diaphragm. Then, the intensity of the beam is adjusted before entering the spectrometer, using two filter wheels".

- (1) Line 105: MkIV Brewers use the third diffraction order for UV measurements. The diffraction grating has 1200 lines per *mm*, not 1800 lines per *nm* as stated

(2, 3) The diffraction order of MkIV Brewers has been corrected, as well as the units corresponding to the line density of the diffraction grating.

- (1) Line 108: The correct specification is 3600 lines per *mm*, not *nm*

(2, 3) The term "nm" has been replaced with "mm".

- (1) Line 108: According to the diffraction equation, identical diffraction angles for a given wavelength occur with 1200 lines/mm in third order and 3600 lines/mm in first order

(2, 3) The phrase "smaller diffracted angles than for double monochromator Brewers" has been removed from the manuscript.

- (1) Line 116: The Brewer slit function is better described as trapezoidal, not triangular (e.g., https://acp.copernicus.org/articles/14/1635/2014/)

(2, 3) The term "triangular" has been replaced with "trapezoidal".

- (1) Line 208: The statement "using the DT tests, this constant is frequently checked and updated when necessary" is misleading, especially after the claim that the dead time value is stored in the B-files. This could wrongly imply real-time updates. Furthermore, DT test results are highly sensitive to the intensity of the standard lamp (SL), making their use as an uncertainty estimate questionable. In practice, the dead time is more reliably verified through ozone direct-sun (DS) comparisons during calibration audits

(2) As the referee states, DT is not real-time updated, it is usually updated during calibration audits. As for its uncertainty estimation, both methodologies, the one using the standard lamp (SL) and the one using direct-sun (DS) measurements, have their limitations. The former may occasionally result in noisy results when the intensity of the SL is weak. On the other hand, the latter (DS measurements) can also lead to uncertain results when the Sun is covered (partially or fully) by clouds. In this study, the SL method was used and the results reported are similar to those obtained using solar DS

measurements. For example, the DT uncertainties estimated in this study and their resulting contribution are similar to the ones obtained by Fountoulakis et al. (2016b) (https://doi.org/10.5194/amt-9-1799-2016)

(3) To clarify this issue, the phrase "this constant is frequently checked and updated when necessary" has been modified to "this constant is frequently checked and updated during calibration audits". Furthermore, the following sentence has been added to Section 3.1.1 (error sources affecting Brewer signal): "The dead time uncertainties found using the previous methodology are similar to those reported by Fountoulakis et al. (2016b). They determined standard deviations of 1–2 ns for the Brewer dead time using direct-Sun measurements". Moreover, following the comments made by Reviewers #1 and #3, the following information has been added to former Section 4.2.4. (irradiance uncertainty produced by the dead time correction): "Finally, the irradiance uncertainties previously estimated can be compared to the one reported by Fountoulakis et al. (2016b). Their study shows that if the DT ranges from 15 to 45 s and has an error of 2 ns, it leads to irradiance uncertainties of 0.12–0.13, 0.25–0.28, and 0.69–1.13 % for signals of 1, 2, and 5 million counts s$^{-1}$, respectively. These values are similar to the ones found for all Brewers, except Brewer #202, which has an uncertainty larger than 2 ns. For these Brewers, the irradiance uncertainty is less than 0.15, 0.35, and 0.9 % for signals of 1, 2, and 5 million counts s$^{-1}$, respectively"

- (1) Line 263: Many modern Brewers are now equipped with heaters, thus minimizing internal temperature fluctuations

(2) Modern Brewers have heaters that are switched on when the internal temperature of the instrument drops below a specified value. The minimum temperature can be set at 10 °C or 20 °C (Kipp & Zonen, 2018). Therefore, these heaters are used to avoid operating at very low temperatures, but they do not control the internal temperature of the instrument. During the 18th RBCC-E campaign, hosted in the southwest of Spain in summer, the internal temperature of the Brewers fluctuated greatly, between 20 and 42 °C.

(3) The previous information has been added to the text as follows: "Brewer spectrophotometers are operated within a weather-proof housing and have electrical heaters to prevent operation at low temperatures. If the internal temperature of the instrument falls below 10 °C or 20 °C (Kipp & Zonen. 2018), these heaters are automatically switched on. Nevertheless, the Brewer internal temperature is not stabilised and can fluctuate throughout the day".

- (1) Lines 465–466: A more accurate phrasing would be: "The instrument is calibrated using the standard lamps with the input optics positioned at zenith."

(2) Lines 465–466 correspond to the irradiance uncertainty caused by the errors committed during the distance adjustment. They describe that the uncertainty sources affecting the responsivity (distance, uncertainty of reference lamp, and radiometric stability) have no angular dependency. Maybe the reviewer is suggesting rephrasing another part of the manuscript.

(3) To take into account the calibration procedure mentioned by the reviewer, the suggested phrase has been added in Section 2 (description of the instrument) as "As for their calibration, the instrument is calibrated using reference lamps (usually 1000 W

lamps) with the input optics positioned at zenith”. The term “standard” has been replaced with “reference” to differentiate between the lamps used for the radiometric calibration and the standard lamp that is inside the Brewer spectrophotometer.

**(1) 3. Treatment of stray light**

(1) The application of the algorithm proposed by Savastiouk et al. (2023) to spectral UV irradiance measurements raises several concerns. According to their paper, this method was specifically developed for direct-sun observations conducted at a fixed grating position and was not validated for use at other wavelengths or for measurements requiring grating movement, such as those involved in spectral UV irradiance scans. Additionally, spectral UV irradiance measurements are not performed simultaneously across wavelengths, further complicating the applicability of the method to this context.

The validity of applying this correction approach should be demonstrated, ideally through comparisons of corrected UV spectra from single-monochromator instruments with measurements from reference instruments with negligible stray light effects. Alternatively, the uncertainty introduced by the application of this stray light "model" should be explicitly quantified, in addition to the two uncertainty sources discussed in lines 181–182.

The manuscript states that “deriving the uncertainty of this [the 5-wavelength] method for the Brewers under study is difficult, as it would use the information from only five wavelengths.” However, it remains unclear what correction method, if any, was actually employed during the RBCC-E campaign. Moreover, the statement regarding the difficulty of the 5-wavelength approach is vague. The authors should clarify what specific challenges prevent its application and provide justification for the selected correction method.

(2) During the RBCC-E campaign, the stray light was corrected using the five-wavelength method, according to EuBrewNet's processing algorithm (https://eubrewnet.aemet.es/dokuwiki/doku.php?id=codes:uvaccess, last access: 29 May 2025). Then, the corrected spectra were compared against the reference unit QASUME (Hülsen, 2023). At first, this approach was discarded as it would also consider, besides the stray light effects, the temperature and the cosine error of the instrument (single Brewers are not corrected for these two error sources). Nevertheless, the effect of these two sources is expected to be small below 292 nm. As a result, the uncertainty determined in this way is not a huge overestimation of the uncertainty produced by stray light correction. Furthermore, since the stray light was estimated during the RBCC-E campaign using the five-wavelength method, the uncertainty should be estimated using the suggested approach (comparison to the QASUME) instead of the one proposed by Savastiouk et al. (2023).

(3) Following the reviewer's comment, the methodology implemented by Savastiouk et al. (2023) has been replaced with the one used in the RBCC-E campaign. As a result, the description in lines 168–184 has been changed to “The uncertainty of this method was estimated by comparing the corrected irradiance to the QASUME from 290 to 292 nm. This estimation also includes the effects of temperature and cosine errors since the single Brewers under study are not corrected for these two sources of error. Nevertheless, since the effect of these two sources is expected to be small below 292 nm, the uncertainty determined might be only a slight overestimation. Furthermore, the standard deviation from the measurements of the five wavelengths (from 290 to 292 nm) was also derived and combined with the uncertainty obtained from the QASUME comparison”. Furthermore, the results section has been updated to reflect the new stray light

correction methodology. As a result, all figures in sections 4.1 and 4.2 have been modified. Finally, the abstract and conclusion sections of the manuscript have also been revised and changed when necessary.

**(1) 4. Treatment of systematic error sources**

**(1) 4a. Ambiguity in use of terminology**

(1) According to the Guide to the Expression of Uncertainty in Measurement (GUM), all known systematic error sources must be corrected prior to uncertainty estimation, with the residual uncertainty from those corrections included in the total uncertainty budget. This methodological framework is not clearly articulated in the manuscript, particularly in the methodology section, and the terminology used throughout the manuscript often conflates systematic errors with uncertainty contributions.

For example, lines 26–28 state: "For wavelengths below 300 nm, the differences between single- and double-monochromator Brewers increase, due to stray light and dark counts. For example, at 295 nm, the relative uncertainties of single Brewers range between 11–14% while double Brewers have uncertainties of 4–7%." This wording implies that stray light is itself a source of uncertainty, rather than a systematic (albeit variable) error that should be corrected. A similar ambiguity is present in lines 449–450: "the contribution of stray light increases rapidly as wavelength decreases." It is unclear whether this refers to the increasing magnitude of the stray light error or to the uncertainty associated with correcting it. The same confusion is evident in the discussion of cosine correction.

The authors should explicitly distinguish between systematic effects (which must be corrected) and the uncertainties associated with those corrections.

(2, 3) Noted. All sections of the manuscript have been revised to differentiate clearly between the sources of error and the uncertainty caused by their correction. For example, lines 26–28 have been rephrased to "the differences between single- and double-monochromator Brewers increase, due to the uncertainty in stray light correction". On the other hand, lines 449–450 have been corrected to "The sensitivity analysis shows that the irradiance uncertainty produced by stray light correction increases rapidly as wavelength decreases". Furthermore, for greater clarity, the following description of the terms "error" (both systematic and random) and "uncertainty" has been added in the methodology section, at the beginning of section 3.1.: "Error sources are usually separated into random and systematic components. Random errors produce variations in repeated measurements and as such, are usually reduced by increasing the number of observations (BIPM et al., 2008a). On the other hand, systematic errors can be compensated by applying a correction factor to the irradiance measured. Even if all errors are appropriately characterised and corrected, there still remains a doubt, an uncertainty, about the accuracy of the reported result (BIPM et al., 2008b). In the following, the term "error" will denote the imperfections in a measurement result, while the term "uncertainty" will be used to reflect the existing doubt regarding the value of the measured spectral UV irradiance. In this way, it is important to differentiate between the source of error (for example cosine error) and the uncertainty associated with its correction".

**(1) 4b. Effectiveness of corrections**

(1) To justify the inclusion of systematic error corrections, the authors must demonstrate that these corrections lead to improved measurement accuracy. This could be done by presenting plots showing the relative (%) differences between corrected spectra and a reference standard (e.g.,

QASUME). Such comparisons would help assess the effectiveness of the applied stray light and cosine corrections. The residual discrepancies should then be discussed in the context of the total uncertainty budget.

(2) The corrections considered in this study (cosine error, temperature, stray light, etc.) have been proven to be effective by numerous studies (e.g. Fountoulakis et al., 2016b; Garane et al., 2006; Kerr, 2010; Lakkala et al., 2008, 2018). Moreover, the campaign report also shows that the cosine correction implemented improved the quality of the measurements (Hülsen, 2023). Nevertheless, following the suggestions made by all reviewers, a section has been added in the results of the manuscript to study the ratio of the instruments to the QASUME, the reference unit in Europe.

(3) The following information has been added in the results section:

"The corrections applied to the measured irradiance (described in section 3.2) are recommended by numerous studies to improve the quality of the measurements (e.g. Fountoulakis et al., 2016b; Garane et al., 2006; Kerr, 2010; Lakkala et al., 2008, 2018). This was also verified during the 18th RBCC-E campaign, as the results show that including the cosine correction improves considerably the comparison to the QASUME (Hülsen, 2023). Although the campaign report shows the ratio of each participating Brewer to the QASUME (see Hülsen (2023)), it is interesting to represent the ratio of all studied Brewers together. In this way, Fig. 4 displays the global irradiance ratio to the QASUME obtained from dividing the irradiances shown in Fig. 1 to the irradiance recorded by the QASUME unit.

[Figure]

Figure 4. Global irradiance ratio to the QASUME recorded on 13 September at 14:00 UTC. (a) First group (double Brewers with cosine correction). (b) Second group (two single and three double Brewers with no cosine correction implemented).

Figure 4 shows the effectiveness of the cosine correction, since Brewers with such correction implemented (Fig. 4a) report irradiances more similar to the one measured by the QASUME. Nevertheless, the agreement between all Brewer spectrophotometers and the QASUME is within 10 % for wavelengths above 310 nm.

Furthermore, the irradiance uncertainty found for each Brewer in the previous section can be used to derive the uncertainty of their ratio to the QASUME. Table 3 summarises the combined standard uncertainty of the average Brewer/QASUME ratio measured on 13 September at three different wavelengths. These uncertainties were computed by combining the irradiance uncertainty of each Brewer and the one from the QASUME, provided by Hülsen et al. (2016).

**Table 3. Number of simultaneous scans, mean ratio to the QASUME and its combined standard uncertainty (both absolute and relative) determined between 310 and 360 nm on 13 September.**

| Brewer ID | N | Ratio to the QASUME (310–360 nm) | | |
|---|---|---|---|---|
| | | Mean value | Combined standard uncertainty | Relative standard uncertainty (%) |
| #117 | 19 | 0.927 | 0.034 | 3.7 |
| #150 | 20 | 1.035 | 0.035 | 3.4 |
| #151 | 24 | 0.914 | 0.033 | 3.6 |
| #158 | 17 | 0.972 | 0.036 | 3.7 |
| #172 | 19 | 0.947 | 0.033 | 3.5 |
| #185 | 18 | 0.978 | 0.030 | 3.1 |
| #186 | 15 | 1.003 | 0.043 | 4.3 |
| #202 | 19 | 0.928 | 0.033 | 3.6 |
| #228 | 19 | 0.937 | 0.033 | 3.5 |
| #256 | 19 | 1.003 | 0.037 | 3.7 |

Table 3 shows that only those Brewer spectrophotometers with a cosine correction implemented (#150, #158, #185, #186, and #256) include the ideal value of the ratio (unity) within their uncertainty interval. The remaining Brewers underestimate the UV irradiance and deviate from unity. This is likely caused by the cosine and temperature errors of the instruments, which couldn't be corrected (there was no available information regarding their characterisation). Therefore, to improve the performance of these uncorrected Brewers these two sources must be characterised and corrected".

**(1) 4c. Missing information for some Brewers**

(1) Only five Brewer instruments were characterized for their angular response, yet cosine correction is an essential component of systematic error treatment. Given this limitation, the authors should consider restricting their analysis to only those instruments for which a full angular characterization was performed.

(2) It is well-known that cosine correction is an important source of uncertainty of the Brewer UV measurements. As a result, several studies have proposed different methodologies to characterise the angular response and then correct the cosine error (Antón et al., 2008; Bais et al., 2005; Gröbner et al., 1996; Lakkala et al., 2018). Even if all these methods exist, the RBCC-E intercomparison campaign held at the El Arenosillo station in 2023 showed that the irradiances of most Brewers were not corrected for cosine error. Since the characterisation of these instruments is incomplete, so is their uncertainty analysis. Nevertheless, the aim of this study is to evaluate the uncertainty of the spectral irradiance currently reported by most of the Brewers. That is why in the conclusions section we emphasise the difficulties found when trying to evaluate the uncertainty of the Brewers. If the analysis is restricted to the Brewers that are completely characterised, it wouldn't be representative of the current state of the Brewer network. Nevertheless, this limitation should be explicitly acknowledged in the manuscript.

(3) Following the reviewer's comment, the following information has been added at the beginning of the results section (line 363) "The uncertainty evaluation of the Brewer spectrophotometers in this second group is limited as it is missing one of the key uncertainty sources in solar radiometry, cosine correction. As a result, the uncertainties determined are likely an underestimation. Nevertheless, these estimations represent the uncertainty of the spectral irradiance reported by most of the participating Brewer spectrophotometers".

**(1) 4d. Uncertainty in cosine correction**

(1) The uncertainty associated with the direct-to-global ratio (DIR/GLO) has not been addressed in the manuscript. This is a significant omission, as this uncertainty can be non-negligible, particularly under overcast or mixed sky conditions. By excluding it, the analysis is effectively constrained to clear-sky scenarios. This limitation should be explicitly acknowledged and clearly stated in the discussion.

(2) Indeed, the cosine correction implemented in this study is valid under cloud-free conditions. Consequently, the results of the uncertainty analysis have been limited to clear-sky scenarios.

(3) The discussion section has been modified to: "The uncertainty evaluation was performed for all the UV scans measured during the campaign under cloud-free conditions". Furthermore, following the comments made by Reviewers #1 and #3, a remark has also been added regarding the appropriate methodology for cloudy conditions: "Under cloudy conditions, the methodology for calculating the cosine correction and noise must be adapted accordingly. As the cosine correction depends on the cloudiness, the cloud cover must be considered when modelling the direct-to-global ratio".

**(1) 4e. Uncertainty in wavelength alignment**

(1) Section 4.2.8 addresses wavelength alignment uncertainty, but the proposed treatment is incomplete. The wavelength shifts should first be minimized through the application of an improved dispersion equation, after which any residual shifts can be analyzed to assess temporal variability. Ignoring this step may overestimate the uncertainty or fail to correct for avoidable errors.

(2) Ideally, a wavelength recalibration should have been performed during the RBCC-E campaign. Nevertheless, the irradiances measured by these instruments (#151 and #158) are affected by the wavelength shifts reported in section 4.2.8. However, this limitation should be clearly acknowledged and a remark should be made on how to avoid these large errors.

(3) Following the reviewer's comment, the following information has been added in section 4.2.8: "These large wavelength shifts indicate that the dispersion function of these instruments might be outdated. Therefore, special attention should be paid to the wavelength scale of the instrument by performing frequent and accurate wavelength calibrations". Furthermore, in the conclusions section, this limitation has been stated as "Based on the findings of this sensitivity analysis, to reduce the overall uncertainty of a Brewer spectrophotometer, it is recommended to [...] (e) monitor wavelength shifts and reduce them below 0.05 nm through frequent wavelength calibrations and accurate determinations of the instrument wavelength scale".

**(1) 4f. Separation of systematic and random error sources**

(1) There appears to be a fundamental issue in how error sources are classified and treated. It seems that all sources of error were handled as random, without distinguishing between random and systematic effects. This approach leads to implausible outcomes, such as the claim that the UV Index can be determined with an uncertainty (0.28–0.53%) lower than that of the irradiance calibration lamps themselves (line 538). The authors should clearly explain how systematic and random error sources were identified and subsequently treated in the uncertainty analysis.

(2) The uncertainty analysis has been carried out following the recommendations of the Guide to the expression of Uncertainty in Measurements (GUM). This Guide recommends calculating the

uncertainty "*based on the concept that there is no inherent difference between an uncertainty component arising from a random effect and one arising from a correction for a systematic effect*" (BIPM et al., 2008b). Therefore, this methodology stands in contrast to older methods that recommended separating the sources into "systematic" and "random" and combining the uncertainties associated with each in their own way (BIPM et al., 2008a). This approach results in relative combined standard uncertainties that are similar to the ones found in previous studies (Bernhard and Seckmeyer, 1999; Garane et al., 2006; Hülsen et al., 2016; Fountoulakis et al., 2020). Therefore, the methodology implemented provides coherent uncertainty estimations. Nevertheless, the Monte Carlo simulation was wrongly adapted for the calculation of the UV index uncertainty. The propagation of the PDFs of the uncertainty sources was incorrect and, as a result, the determined uncertainty was way too small. Following the methodology of Cordero et al. (2007), the Monte Carlo simulation implemented for the uncertainty evaluation of the UV index has been revised and corrected.

(3) The information regarding systematic and random errors have been added in the methodology section as "Error sources are usually separated into random and systematic components. Random errors produce variations in repeated measurements and as such, are usually reduced by increasing the number of observations (BIPM et al., 2008a). On the other hand, systematic errors can be compensated by applying a correction factor to the irradiance measured". Then, in section 3.3 (Monte Carlo method) the combination of their uncertainty has been described as follows: "Following the GUM guidelines, the uncertainty arising from random and systematic errors have been treated identically (BIPM et al., 2008b)". As for the UV index, its uncertainty has been recalculated and the values reported in section 5 have been modified to "When integrating erythemal spectral irradiance to compute the UV Index (UVI), the resulting uncertainty ranges from 2.7 % to 6.2 %". Finally, in the conclusions section, a brief comparison of these values to the ones found in other studies has been added. In this way, the following information has been included in line 625 "Therefore, the relative combined standard uncertainties determined in this study are comparable to those of other UV spectroradiometers. This also applies to the uncertainties of erythemal irradiance and UV index, as the values estimated are also similar to the ones found for other instruments (Bernhard and Seckmeyer, 1999; Cordero et al., 2007)".

**(1) 5. Calibration and measurements**

(1) In standard practice, the uncertainty budget for UV irradiance measurements is assessed in two distinct phases: one for the calibration process and one for actual solar measurements (e.g., Bernhard and Seckmeyer, 1999). This distinction is important because calibration is itself a measurement procedure and shares several uncertainty sources with field measurements. Typically, the uncertainty associated with the calibration is propagated and added to the overall uncertainty of the spectral UV irradiance. Given that the authors employed a Monte Carlo method, it would be expected that the analysis reflects this two-phase approach, explicitly separating and then combining the calibration and measurement uncertainties. This methodological structure appears to be missing from the current work.

(2) There is no need to separate the uncertainty sources into two phases as the Monte Carlo method allows to vary all uncertainty sources at once. At each iteration of the Monte Carlo simulation, the signals measured outdoors and under the reference lamp are varied according to the PDFs of their uncertainty sources (noise, dead time, stray light, and dark signal). The irradiance of the reference lamp is also varied according to its uncertainty (distance adjustment and calibration certificate) and the responsivity is recalculated using the new values. Then, using the new values

of the outdoor signal and the responsivity, the corresponding UV irradiance is obtained. Finally, this irradiance is also varied according to the uncertainties produced by the wavelength misalignment, cosine, and temperature correction. A flow chart of this process can be found in González et al. (2024b). Although it is not necessary to separate the uncertainty analysis into two phases, it can be interesting to calculate the uncertainty of the calibration process by only considering the uncertainty sources described in sections 3.1.1 and 3.1.2 for the signal measured during the calibration and the responsivity, respectively.

(3) To make it clear that the calibration procedure also includes the uncertainty sources affecting field measurements, the following information has been added in section 3.1.2 "Then, the signal under the lamp is recorded several times and corrected for dark counts, dead time, and stray light (see Section 3.1.1.). The responsivity of the instrument is derived by dividing the corrected signal by the irradiance of the reference lamp. However, this responsivity is also affected by other sources of uncertainty produced during the radiometric calibration such as the distance adjustment between the lamp and the diffuser, the radiometric stability, and the uncertainty of the spectral irradiance emitted by the reference lamp". Furthermore, the irradiance uncertainty produced solely by the calibration procedure has been added in the sensitivity analysis (former section 4.2) as "Regarding the calibration of the instrument (uncertainty sources affecting the responsivity and the signal measured under the reference lamp), it leads to irradiance uncertainties that range from 2.3 % (Brewer #185) to 3.8 % (Brewer #150)".

**(1) 6. Bibliographic references**

(1) The bibliography omits several relevant studies related to uncertainty analysis in Brewer spectrophotometer measurements. Notably, valuable insights can be found not only in works addressing UV irradiance, but also in studies focused on trace gas retrieval and aerosol optical depth (AOD) measurements using the Brewer in direct-sun mode. These references often include rigorous uncertainty analyses that could enhance the methodology of this study. The authors should broaden the literature review to incorporate these contributions.

(2) As previously discussed, the measuring procedure is different for global UV irradiance (the diffraction gratings rotate while the slit mask is fixed) and direct-Sun (the diffraction gratings remain fixed while the slit mask rotates) observations. As a result, most of the sources of error affecting these measurements are different. For example, López-Solano et al. (2018) estimate the uncertainty of Brewer AOD by considering the uncertainties arising from ozone optical depth, pressure, and calibration (carried out by transfer or by a Langley plot). None of these sources are relevant in the uncertainty evaluation of UV irradiance (Bernhard and Seckmeyer, 1999; Webb et al., 1998). Nevertheless, there are some uncertainty sources in common, such as stray light, dead time (nonlinearity), or dark signal. Unfortunately, these sources are not as important for direct-Sun measurements as they are for UV observations. As a result, they are sometimes overlooked in the uncertainty evaluation (e.g. Diémoz et al., 2014; López-Solano et al., 2018; Nuñez et al., 2022). Moreover, the number of rigorous uncertainty analyses is very limited for Brewer spectrophotometers. As a result, the literature review could not be greatly broadened. The contributions of these studies have been incorporated in the methodology section whenever possible.

(3) The contribution of other uncertainty evaluation studies has been added in the methodology section as follows:

- Line 167 has been modified to "It is usual to estimate stray light as the average signal recorded below 292 nm (e.g. Arola and Koskela, 2004; Lakkala et al., 2008; Mäkelä et al., 2016) and the correction is carried out by subtracting this average value from the signal measured at all wavelengths".

- "Moreover, the uncertainties estimated for dead time also agree with the ones applied in other uncertainty evaluation studies for Brewer spectrophotometers (Diémoz et al., 2014)" has been added at line 211.

**(1) 7. Section 5**

(1) Section 5 could be significantly shortened or removed entirely. Much of the content is either self-evident or repetitive of points that belong more appropriately in the Introduction.

(2, 3) Section 5 was not in the original version of the paper, but it was specifically demanded by the editor to highlight the positive impacts of the study. Thus, following the reviewer's comment, Section 5 has been summarised and part of its information has been moved to the Conclusion section so as to shorten the section, as follows:

- Moved to the Conclusions' section: "For the ten Brewer spectrophotometers analysed in this study, the average combined standard uncertainty in erythemal spectral irradiance ranges between 2.7 % and 3.9 %, with maximum values varying from 17 % for a single Brewer to 3.4 % for a double Brewer for wavelengths above 310 nm. This variability indicates the need of characterising each Brewer spectrophotometer individually rather than relying on generic values, which may not fully exploit the precision these instruments can achieve (Gröbner et al., 2006). When integrating erythemal spectral irradiance to compute the UV Index (UVI), the resulting uncertainty ranges from 2.7 % to 6.2 %. The UVI, along with cumulative erythemal irradiance doses, represents a fundamental metric for informing the public about the potential adverse effects of UV radiation (Lucas et al., 2019)."

- Summarised section 5: "This study provides an accurate quantification of measurement uncertainties in Brewer spectrophotometer UV data, identifying the main sources of uncertainty and their relative contributions to guide instrumental optimisations. These aspects are of great interest for different studies and fields of work.

  One of the key applications of accurately determining the uncertainties in spectral UV measurements is the computation of effective irradiance for various biological effects, such as erythema, vitamin D synthesis, melanoma risk, and DNA damage, through the integration of the spectral irradiance weighted by different action spectra (Webb et al., 2011).

  The findings benefit regulatory applications, supporting evidence-based UV exposure limits for outdoors workers (Vecchia et al., 2007) and improving standards for sun protection products (Young et al., 2017). The proposed methodology also allows sensitivity analysis to help identify paths for improving instrumentation, measurement procedures, and calibration protocols, which are essential for ensuring the traceability of UV spectroradiometer measurements to international standards (Gröbner et al., 2006). Reliable measurements in the 300–400 nm wavelength range with a relative uncertainty below 4 % are crucial for radiometric networks and studies comparing data from different stations. Ensuring this quality level requires periodic and regular calibrations using lamps

traceable to international standards. For example, the QASUME (Quality Assurance of Solar Ultraviolet Spectral Irradiance Measurements) project has established a European reference standard for UV solar radiation measurements, achieving a global UV irradiance uncertainty of approximately ±4 % in the 300–400 nm range (Gröbner and Sperfeld, 2005) and a direct solar irradiance uncertainty of about 0.7 % (Gröbner et al., 2023). More advanced developments, such as QASUMEII, have further improved accuracy, with a combined uncertainty for global UV measurements of 1.01 % between 310 and 400 nm and 3.67 % at 300 nm (Hülsen et al., 2016).

Furthermore, the uncertainty framework significantly strengthens the validation of satellite-based UV products from instruments such as OMI, TROPOMI, and TEMPO (Klotz et al., 2024; Tanskanen et al., 2007), where ground-based measurements with an uncertainty of less than 5 % are crucial for calibration.

In summary, precise quantification of uncertainty in spectral UV measurements benefits a broad range of scientific, regulatory, and public health applications, reinforcing the need for rigorous uncertainty assessment in Brewer spectrophotometer measurements".

**(1) TECHNICAL REMARKS**

- (1) The terminology used for the Brewer instrument should be standardized throughout the manuscript. The correct and commonly accepted term is "Brewer spectrophotometer". The inconsistent use of terms such as "spectroradiometer" or "spectrometer" is confusing. If the authors want to emphasize the Brewer's capability to measure global irradiance, the more general term Brewer "instrument" may be appropriate in that context.

  (2, 3) Noted. The terms "spectroradiometer" and "spectrometer" have been replaced with "spectrophotometer" throughout the manuscript. Whenever appropriate, the term "Brewer instrument" has been used to emphasise that Brewers can also measure global UV irradiance.

- (1) Line 25: Please note that only a limited number of steps in the algorithm are non-linear, and these are likely to represent minor contributors to the total uncertainty.

  (2, 3) Following the reviewer's comment, line 25 has been modified to "considers the nonlinearity of certain steps in the UV processing algorithm". Then, the effect of these nonlinear steps in the uncertainty evaluation of Brewer spectrophotometers is discussed later in the introduction section (in the abstract the number of words is limited and complicates a proper discussion).

- (1) Line 30: The statement that stray light depends simply or solely on UV intensity is inaccurate.

  (2) In line 30 the text is referring to the irradiance uncertainty produced by the stray light correction, which increases as the wavelength decreases (see Fig. 5). It was never the intention to imply that stray light only depends on UV intensity. As indicated in the methodology, stray light is produced by the scatter inside the instrument and the instrument contamination with dust.

  (3) To avoid confusion, the term "correction of" has been added to indicate that it is not the stray light, but its correction which leads to an irradiance uncertainty that increases as the intensity of the UV irradiance decreases.

- (1) Line 57: The phrase "is trying" is outdated as the COST Action Eubrewnet has concluded. Please revise to reflect the current status.

  (2) EuBrewNet is still working on harmonising the procedures for Brewer QA/QC (both direct-Sun and global UV measurements). The Cost Action 1207 has finished, but its webpage (also named EuBrewNet) and UV processing algorithms are still operational, thanks to AEMET (the State Meteorological Agency from Spain). Any registered user can access, download, and process UV and ozone products from Brewer spectrophotometers that are part of the EuBrewNet community at https://eubrewnet.aemet.es/eubrewnet.

  (3) To clarify this issue, the indicated phrase has been rewritten as: "In this context, EuBrewNet (European Brewer Network), originally developed through COST Action 1207 and currently operational thanks to AEMET (Spanish State Meteorological Agency), is working on harmonising and developing coherent practices for Brewer QA/QC".

- (1) Line 60: Appropriate bibliographic references on the "well-established QA for UV measurements" must be included.

  (2, 3) Following the comments made by Reviewer #2, the phrase "well-established QA for UV measurements" has been replaced with a description of the QA performed by during the RBCC-E intercomparison campaigns and a brief discussion on how this procedure reflects the guidelines of Webb et al. (2003). Furthermore, to take into account the suggestions of Reviewer #3, bibliographic references have been added to back these claims. Therefore, line 60 has been modified to: "These intercomparison campaigns meet the main requirements laid out by Webb et al. (2003), i.e. transparency and objective comparison algorithms (see the campaign reports at the PMOD/WRC website, https://www.pmodwrc.ch/en/world-radiation-center-2/wcc-uv/qasume-site-audits/, the report of the 18th intercomparison campaign, Hülsen, 2023, and an overview of the EuBrewNet's algorithms, López-Solano, 2024)".

- (1) Line 61: Please clarify whether the intended meaning is ozone, UV irradiance, or both.

  (2, 3) QC is a pending task for both ozone and UV irradiance measurements. The previous information has been added to the indicated line: "it remains one of the main challenges for Brewer sites measuring ozone (Fioletov et al., 2008) and UV irradiance".

- (1) Line 96: It should be explicitly stated that a diffuser is used as the global entrance optic for irradiance measurements.

  (2, 3) The following information has been added at line 96: "global UV irradiance enters through the entrance optics, consisting of a Teflon diffuser covered by a quartz dome". Then, on lines 115–121, the different types of entrance optics are described in greater detail.

- (1) Line 99: Please include an explanation that the diffraction grating is rotated during UV spectral scans. Also, provide a description of Filter Wheel #3 (FW#3) for MkIV Brewers.

  (2, 3) Following the reviewer's comment, the explanation of the ozone and UV mode has been added to the manuscript: "At the exit of the spectrometer, there is a cylindrical slit

mask. For ozone, dead time, and dark signal observations, the diffraction grating is fixed while the slit mask rotates, selecting in this way the wavelength. On the other hand, for the measuring of UV irradiance, the slit mask remains fixed, and the diffraction grating rotates (using a micrometre) to select the wavelength.". Furthermore, a description of Filter Wheel #3 has been added as well: "In MkIV Brewers, the emerging light passes through a third filter wheel, which has several filters to block undesired radiation: (1) in the ozone mode, a UG-11/NiSO4 filter combination is used, (2) in the UV mode, the filter switches to a UG-11 filter, and (3) in the NO2 mode, a BG-12 filter is used to block UV radiation (Kipp & Zonen, 2007)".

- (1) Table 1: The column listing operator names should be omitted. Rather, the characteristics of the diffuser used with each Brewer would be more relevant to the technical discussion.

  (2, 3) The information regarding operators in Table 1 has been replaced with the characteristics of the entrance optics of each Brewer, resulting in the following table:

| Brewer | Type (monochromator) | Entrance optics (diffuser) | Institute (Country) |
|---|---|---|---|
| #117 | MkIV (single) | Traditional (flat) | State Meteorological Agency – AEMET (Spain) |
| #150 | MkIII (double) | CMS-Schreder (shaped) | National Institute of Aerospace Technology (Spain) |
| #151 | MkIV (single) | Traditional (flat) | State Meteorological Agency – AEMET (Spain) |
| #158 | MkIII (double) | Traditional (flat) | OTT Hydromet (The Netherlands) |
| #172 | MkIII (double) | Traditional (flat) | University of Manchester (UK) |
| #185 | MkIII (double) | CMS-Schreder (flat) | Izaña Atmospheric Research Center, AEMET (Spain) |
| #186 | MkIII (double) | Traditional (flat) | State Meteorological Agency – AEMET (Spain) |
| #202 | MkIII (double) | Traditional (flat) | Danish Meteorological Institute (Denmark) |
| #228 | MkIII (double) | Traditional (flat) | Danish Meteorological Institute (Denmark) |
| #256 | MkIII (double) | CMS-Schreder (flat) | Izaña Atmospheric Research Center, AEMET (Spain) |

- (1) Line 110: The cosine error can be mitigated through appropriate correction methods, as discussed in Section 3.1.3.

  (2, 3) Following the reviewer's comment, this information has been added in line 110 as: "Therefore, a correction is needed to mitigate such deviation, as will be described later in Section 3.1.3".

- (1) Line 129: Clarify how the angular response characterization was conducted for the two additional Brewers.

  (2) Line 129 contains an error, four Brewers (not three) were calibrated using the Brewer Angular Test. The Brewer remaining (Brewer #158) was calibrated using the operator's lab cosine setup. This consists of an arm mounted on a solar tracker, which is programmed to turn by steps of 5° in sync with the Brewer CRD routine.

  (3) The mistake in line 129 has been corrected and the following information has been added to the text: "Four of them (#150, #185, #186, and #256) were characterised using the Brewer Angular Tester (BAT), described later in Section 3.1.3. The remaining Brewer (#158) was characterised in the laboratory of its operating site, using a lamp mounted on an arm that turns by steps of 5°".

- (1) Line 131: Replace "dawn" with "sun rise".

(2, 3) The term "dawn" has been replaced with "sun rise".

- (1) Lines 141–142: The names of instrument operators should be moved to the Acknowledgements section.

  (2, 3) The names of the Brewer and QASUME operators have been moved to the Acknowledgements section.

- (1) Line 149: Please provide a proper citation or link to the Eubrewnet guidelines referenced.

  (2, 3) The link to the EuBrewNet guidelines has been added to the manuscript as well as the citation of the overview presented by López-Solano et al. (2024) in the Quadrennial Ozone Symposium meeting.

- (1) Lines 154–155: The rationale for completely ignoring certain uncertainty sources is not adequately justified. It is unclear whether this was actually done and, if so, on what basis.

  (2) Lines 154–155 refer to those uncertainty sources that have been included in the study, but their characterisation is not as thorough as the ones carried out in other studies. For instance, the Brewer noise was determined by studying the signal-to-noise-ratio (SNR) from lamp measurements, as Bernhard and Seckmeyer (1999) or Cordero et al. (2012) did for the uncertainty evaluation of their instruments. However, in the RBCC-E campaign, Brewers were calibrated by performing 4 lamp scans. Therefore, the SNR was derived from a limited number of UV scans. Nevertheless, we believe that, rather than ignoring these uncertainty sources, it is preferable to include them in the study, even if their characterisation is limited.

  (3) Following the comments made by Reviewers #1 and #3 and to clarify this issue, the indicated lines have been rewritten as "It should be noted that some of the uncertainties (such as those related to noise, stray light, or radiometric stability) have not been determined thoroughly, as the data used for their estimation are insufficient to obtain appropriate statistics. Nevertheless, in all these cases, uncertainty values have been given and included in the Monte Carlo simulation".

- (1) Line 177: The statement that the coefficients are "deemed reliable" is misleading. The two coefficients referenced were derived for ozone and SO2 retrievals in direct sun mode, not for spectral UV irradiance. Their applicability to spectral UV measurements should be justified.

  (2, 3) Following the reviewer's general comment on the treatment of stray light, the method proposed by Savastiouk et al. (2023) is no longer included in the uncertainty analysis. As discussed above, the stray light is now corrected using the average counts measured from 290 to 292 nm. Therefore, the indicated lines have been removed from the manuscript and a new discussion regarding stray light has been added (see the response to comment "3. Treatment of stray light").

- (1) Lines 201–202: All Brewer measurements, not only global UV scans, collect dark count data. Therefore, the authors could have considered a much larger dataset, potentially an order of magnitude more observations per day.

(2) The dark signal measurements from other Brewer observations (such as direct-Sun measurements) have been included in the uncertainty analysis. Therefore, as the referee states, the dataset considered has increased considerably.

(3) Lines 201–202 have been modified to reflect the changes in the methodology: "The number of available measurements depended on the instrument, but, in total, more than 2500 dark signal measurements were recorded by each Brewer during the intercomparison campaign". Furthermore, Section 4.2 (sensitivity analysis) has also been corrected to reflect the new uncertainties. The analysis in lines 441–442 has been changed to "For larger wavelengths, its impact can be disregarded as dark signal correction leads to irradiance uncertainties of less than 0.06 % in double monochromator Brewers".

- (1) Line 215: Please specify which laboratories or types of labs are being referred to.

(2, 3) Following the reviewer's suggestion, the laboratories and a brief description of their calibration standards have been provided in line 215: "These lamps had been previously calibrated in the laboratories of PMOD/WRC and the Finnish Metrology Research Institute, belonging to Aalto University and MIKES. The calibrations performed by PMOD/WRC are traceable to the primary standard of the Physikalisch-Technische Bundesanstalt (PTB) (Gröbner and Sperfeld, 2005). On the other hand, the Metrology Research Institute is the national standard laboratory for optical quantities in Finland and is part of the CIPM Mutual Recognition Arrangement (CIPM MRA), a framework in which metrology institutes prove the international equivalence of their calibrations and certificates".

- (1) Line 219: The text should elaborate on the effect of the current uncertainty.

(2, 3) The following information has been added in Line 219 regarding the uncertainty caused by the current of the reference lamp: "Based on the findings of Webb et al. (1994), the standard practice is to assume that a 1 % change in the current of the reference lamp leads to a 10 % change in the spectral irradiance measured by the instrument (e.g. Bernhard and Seckmeyer, 1999; Webb et al., 1998). According to the previous rule, the expected change in the irradiance of the Brewers under study would be of 0.125 %, as the electrical current was stabilised to within 0.0125 % during their calibration".

- (1) Line 229: Clarify how the reference plane issue for Brewer #150 was resolved.

(2, 3) Following the reviewer's comment, a description of the determination of the reference plane for Brewer #150 has been added at line 229: "This was carried out by placing an ultrastabilised lamp at several distances and measuring its emitted spectrum. The data showed that the diffuser's reference plane is placed (0.234   0.015) cm below the reference used for calibration, i.e. the metalling ring of the quartz dome covering the Brewer's diffuser (a schematic drawing of this reference can be found in González et al. (2023))".

- (1) Line 242: The statement referencing a "3%" drift lacks a time frame. Please specify the period over which this drift was observed.

(2) The term "drift" was used to describe the uncertainty of the radiometric stability. The reported values (of around 3 %) are the standard deviations from consecutive calibrations, as recommended by Bernhard and Seckmeyer (1999). The data used for this calculation were the available calibration records.

(3) To avoid confusion, the term "drift" has been replaced with "uncertainty" throughout the manuscript. Moreover, following the comments made by the referees, the following information has been added at the end of the indicated paragraph: "For Brewer #150, the radiometric uncertainty was derived using the yearly calibration files from 2005 to 2023, while for Brewer #185 the uncertainty was calculated using the monthly calibration files recorded from 2021 to 2024. As mentioned earlier, no data from prior years could be used as the entrance optics of Brewers #150 and #185 were replaced in 2005 and 2021, respectively".

- (1) Lines 252–253: The wavelength dependency of the stated effect should be included. Please indicate the specific wavelength used in this calculation.

(2, 3) The indicated phrase has been corrected to "a shift of less than 0.05 nm can produce an uncertainty in the UV irradiance of a few percent for wavelengths below 305 nm".

- (1) Lines 271–272: A formula should be included to clarify the described temperature dependence. Specify whether the linear relationship applies to raw count rates or their logarithms.

(2) The linear relationship refers to the responsivity of the instrument, i.e. the signal measured divided by the emitted irradiance of the reference lamp. Brewer #150 was characterised by placing 100 and 1000 W lamps. The instrument temperature increased gradually from 23 to 38 °C while it measured the irradiance emitted by the lamps. Then, the responsivity of the instrument was determined and its change with respect to the internal temperature was studied.

(3) To clarify this issue, the following formula and description has been added at the indicated lines: "Then, the relationship between the internal temperature of Brewer #150 and its change in responsivity with respect to a reference value (31 °C) was studied. The results showed that the instrument's responsivity decreases linearly with temperature, as:

$$r(\lambda, T) = r(\lambda, \mathrm{T_{ref}})[1 + c_\mathrm{T}(T - \mathrm{T_{ref}})] \, , \qquad\qquad (2)$$

where $r(\lambda, T)$ is the responsivity measured at wavelength $\lambda$ and internal temperature $T$, $r(\lambda, \mathrm{T_{ref}})$ is the responsivity measured at the reference temperature $\mathrm{T_{ref}} = 31$ °C, and $c_\mathrm{T}$ is the slope of the linear fit. The latter is the temperature correction factor and for Brewer #150 it has a value of $c_\mathrm{T} = (-0.0016 \pm 0.0002)$ °C$^{-1}$".

- (1) Line 273: According to Eq. 9, the irradiance is divided by $1 + C(T - T\_ref)$, not simply by the correction factor C.

(2, 3) Line 273 has been corrected to "Therefore, the UV measurements of Brewer #150 were corrected for temperature by considering this correction factor and the difference between the temperature of the UV scan and the reference temperature (31 °C), as indicated later in Eq. (9)".

- (1) Line 285: Avoid using the same variable name (C) for both the temperature and cosine corrections.

(2, 3) To avoid confusion, the temperature correction factor has been denoted as "$c_\mathrm{T}$" and the angular response as "$C_\mathrm{R}(\varphi, \theta, \lambda)$".

- (1) Lines 360–364: Describe how systematic error sources were treated for the Brewer instruments that lacked a full calibration.

  (2) Systematic error sources with no characterisation (such as cosine error or temperature dependence for some Brewer spectrophotometers) were not considered in the processing of UV measurements during the intercomparison campaign (Hülsen, 2023). As a result, the global irradiance ratio from these instruments (#117, #151, #172, #202, #228) to the QASUME showed larger deviations from unity than the one from Brewers with all corrections implemented (see the ratios found by Hülsen (2023)).

  (3) As mentioned in responses to comments 4b and 4c, this information has been added at the beginning of the discussion to clearly state that these Brewers have no cosine correction implemented and, as a result, their uncertainty evaluation is limited. Moreover, a comparison with the QASUME has been added to the results section to show that the instruments that lacked a full calibration perform worse than the fully calibrated Brewers.

- Line 372: A plot comparing the ratios between instruments should be included. These ratios should then be compared with the corresponding percentage uncertainties to assess the consistency of the results.

  (2, 3) The instruments have been compared using the ratio to the QASUME, the European reference. As discussed in the response to comment 4b, the agreement between all Brewer spectrophotometers is within 10 % from unity for wavelengths above 310 nm. Furthermore, the uncertainty of the ratio to the QASUME has also been determined, showing that only those Brewers with cosine correction include the ideal value of the ratio (unity) within their uncertainty interval.

- (1) Line 377: The sentence "the uncertainty increases as wavelength grows and SZA decreases" is highly misleading. While the absolute uncertainty may increase, the relative (%) uncertainty generally decreases.

  (2, 3) To clarify this issue, the term "absolute" has been added to differentiate between relative and absolute values. The indicated line has been corrected to "the absolute uncertainty increases as wavelength grows and SZA decreases". The Conclusions section has also been revised to ensure that the discussion regarding absolute and relative uncertainty values is unambiguous.

- (1) Figure 3 and Line 402: The fluctuations observed in the data from Brewers #158 and #151 should be discussed. If these variations are attributable to an outdated or inaccurate dispersion function, then why this function was not updated prior to the analysis?

  (2) The fluctuations of these instruments are indeed caused by wavelength misalignment. As mentioned earlier, the characterisation and uncertainty evaluation of the Brewer spectrophotometers have shown the current limitations of the Brewer network. One of these limitations is the one indicated by the referee: the dispersion function should be updated if fluctuations are detected. A complete and accurate QC procedure as the ones described by Webb et al. (1998) or Bernhard and Seckmeyer (1999) is currently beyond our abilities. However, only if the limitations are identified (i.e. improvement of stray light correction, inclusion of cosine correction, increasing the calibration frequency of the Brewer spectrophotometers, etc.), can they be overcome.

(3) A brief comment has been added to indicate that the discussion regarding wavelength shift is in section 4.3 as "The reason for these behaviours will be described later in section 4.3 (sensitivity analysis)". Furthermore, in the conclusions section the need for updated dispersion function has been added as follows: "Based on the findings of this sensitivity analysis, to reduce the overall uncertainty of a Brewer spectrophotometer, it is recommended to [...] (e) monitor wavelength shifts and reduce them below 0.05 nm through frequent wavelength calibrations and accurate determinations of the instrument wavelength scale".

- (1) Line 461: Clarify whether the term "error" or "uncertainty" is intended here. Also, please verify the stated values using the inverse square law.

  (2) Line 461 can be confusing as it is comparing uncertainty and precision errors. Therefore, it is better to talk about the uncertainty associated with the distance alignment. As for the inverse square law, it was implemented in the Monte Carlo simulation to derive the alignment uncertainty. In each iteration, the distance was varied according to its uncertainty and the irradiance was recalculated as:

  $$E'(\lambda) = E(\lambda) \cdot \left(\frac{d_r}{d'}\right)^2,$$

  where $E(\lambda)$ is the irradiance of the reference lamp, $d_r$ is the reference distance (usually 500 mm for 1000 W lamps), $d'$ is the sampled distance according to its uncertainty, and $E'(\lambda)$ is the resulting irradiance when the distance is modified.

  Nevertheless, the results obtained using the Monte Carlo method can be compared to the GUM uncertainty framework, which propagates the distance uncertainty by obtaining the partial derivative of the inverse square law.

  (3) The errors reported have been replaced with the uncertainties determined (0.59 mm for Brewer #150 and 0.58 mm for the remaining instruments). Moreover, the following discussion regarding the validity of the results obtained has been added to the manuscript: "According to Webb et al. (1998), if the nominal distance is d and its uncertainty $u_d$, the percentage uncertainty can be calculated using the inverse square law ($1/r^2$, where $r$ is the distance between lamp and instrument) as $[(d + u_d)^2 - d^2] * 100 \, / \, d^2$. Therefore, the previous results agree with the formula proposed by Webb et al. (1998)". Additionally, the calibration guidelines established by Webb et al. (1998) have also been mentioned in the sensitivity analysis in line 473 as: "This agrees with the recommendations of Webb et al. (1998). They suggest calibrating the instruments using three reference lamps".

- (1) Lines 518–520: The statement is misleading. Total ozone column (TOC) is typically derived from direct-sun observations, not from spectral UV irradiance measurements.

  (2, 3) Following the comments made by all reviewers, the indicated paragraph has been deleted from the manuscript.

- (1) Lines 542–549: Ozone is only one of several atmospheric factors that influence long-term changes in UV irradiance. Other contributors can also introduce significant variability and trends.

  (2, 3) Following the reviewer's general comment ("7. Section 5"), the indicated lines have been removed from the manuscript to shorten Section 5.

- (1) Line 622: Calibration should be conducted more frequently than once per year. Critically, it should be stated explicitly that the calibration lamps themselves must be periodically recalibrated, and up-to-date certificates must be used.

  (2) Indeed, the frequency of calibration should be higher than once a year. However, most of the Brewer spectrophotometers used in this study were calibrated once every two years. That is why in the conclusions section we emphasise it. As for the reference lamps, it should be stated that they also need to be periodically calibrated.

  (3) Following the reviewer's comment, the information regarding lamp calibration has been added in former line 622 as "(b) calibrate the reference lamps periodically to ensure up-to-date calibration certificates".

- (1) Lines 622–623: The cosine error should be characterized and corrected systematically. If uncorrected, replacing the diffuser could introduce a step-change in the time series data, undermining the consistency of long-term measurements.

  (2) Indeed, the cosine correction should be characterised and corrected on a regular basis. However, the results from this study show that the traditional optics lead to higher uncertainties than other entrance optics. Therefore, to reduce the overall UV uncertainty of a Brewer spectrophotometer, the traditional optics could be replaced. This will lead to a change in the responsivity of the instrument, but this doesn't have to affect long-term monitoring or UV trend calculation provided that the data is re-evaluated and its QC is revisited (e.g. https://doi.org/10.5194/acp-16-2493-2016).

  (3) The previous information has been added to the text as follows: "Although replacing the entrance optics will modify the responsivity of the instrument, this change will not affect the calculation of UV trends or long-term monitoring as long as the data is re-evaluated and its QC revisited (Fountoulakis et al., 2016a)".

---

## Author Comment (AC4)

**Response to Anonymous Reviewer #1**

Authors' response to Reviewer #1 comments on "Evaluation of the uncertainty of the spectral UV irradiance measured by double- and single-monochromator Brewer spectroradiometers". The authors thank the Reviewer for the additional revision of the manuscript as well as for their suggestions for improving the clarity of the text and reply to all comments below.

The answer is structured as follows: (1) comments from Reviewer #1, (2) authors' response and (3) authors' change in the manuscript.

(1) 153–155: I would rephrase the sentence as follows: It should be noted that some of the uncertainties (such as those related to noise, stray light, or radiometric stability) have not been determined thoroughly, as the data used for their estimation are insufficient to obtain appropriate statistics.

(2, 3) Noted. Lines 153–155 have been rephrased following the reviewer's suggestion.

(1) 359: please rephrase to: methodology for calculating the cosine correction and noise must be adapted accordingly.

(2, 3) Done.

(1) 359: Not clear what is meant here. Maybe: For example, by studying the variability of groups of data measured very close in time.

(2, 3) Following the reviewer's comment, the phrase has been modified to "by studying the variability of groups of data measured very close in time".

(1) 371: Do you plan to include this figure in the revised manuscript? I suggest to do so.

(2, 3) Following the comments made by all reviewers, a section has been added to the manuscript to compare all Brewer spectrophotometers with the QASUME, the European reference unit. As a result, the following information has been added to the text:

"The corrections applied to the irradiance measured (described in section 3.2) are recommended by numerous studies to improve the quality of the measurements (e.g. Fountoulakis et al., 2016b; Garane et al., 2006; Kerr, 2010; Lakkala et al., 2008, 2018). This was also verified during the 18th RBCC-E campaign, as the results show that including the cosine correction improves considerably the comparison to the QASUME (Hülsen, 2023). Although the campaign report shows the ratio of each participating Brewer to the QASUME (see Hülsen (2023)), it is interesting to represent the ratio of all studied Brewers together. In this way, Fig. 4 displays the global irradiance ratio to the QASUME obtained from dividing the irradiances shown in Fig. 1 to the irradiance recorded by the QASUME unit.

[Figure]

**Figure 4. Global irradiance ratio to the QASUME recorded on 13 September at 14:00 UTC. (a) First group (double Brewers with cosine correction). (b) Second group (two single and three double Brewers with no cosine correction implemented).**

Figure 4 shows the effectiveness of the cosine correction, since Brewers with such correction implemented (Fig. 4a) report irradiances more similar to the one measured by the QASUME. Nevertheless, the agreement between all Brewer spectrophotometers is within ±10 % from unity for wavelengths above 310 nm.

Furthermore, the irradiance uncertainty found for each Brewer in the previous section can be used to derive the uncertainty of their ratio to the QASUME. Table 3 summarises the combined standard uncertainty of the average Brewer/QASUME ratio measured on 13 September at three different wavelengths. These uncertainties were computed by combining the irradiance uncertainty of each Brewer and the one from the QASUME, calculated by Hülsen et al. (2016).

**Table 3. Number of simultaneous scans, mean ratio to the QASUME and its combined standard uncertainty (both absolute and relative) determined between 310 and 360 nm on 13 September.**

| Brewer ID | N | Ratio to the QASUME (310–360 nm) | | |
|---|---|---|---|---|
| | | Mean value | Combined standard uncertainty | Relative standard uncertainty (%) |
| #117 | 19 | 0.927 | 0.034 | 3.7 |
| #150 | 20 | 1.035 | 0.035 | 3.4 |
| #151 | 24 | 0.914 | 0.033 | 3.6 |
| #158 | 17 | 0.972 | 0.036 | 3.7 |
| #172 | 19 | 0.947 | 0.033 | 3.5 |
| #185 | 18 | 0.978 | 0.030 | 3.1 |
| #186 | 15 | 1.003 | 0.043 | 4.3 |
| #202 | 19 | 0.928 | 0.033 | 3.6 |
| #228 | 19 | 0.937 | 0.033 | 3.5 |
| #256 | 19 | 1.003 | 0.037 | 3.7 |

Table 3 shows that only those Brewer spectrophotometers with a cosine correction implemented (#150, #158, #185, #186, and #256) include the ideal value of the ratio (unity) within their uncertainty interval. The remaining Brewers underestimate the UV irradiance and deviate from unity. This is likely caused by the cosine and temperature errors of the instruments, which couldn't be corrected (there was no available information regarding their characterisation). Therefore, to improve the performance of these uncorrected Brewers these two sources must be characterised and corrected".

---

## Author Response (AR2)

**Response to Anonymous Reviewer #1**

Authors' response to Reviewer #1 comments on "Evaluation of the uncertainty of the spectral UV irradiance measured by double- and single-monochromator Brewer spectrophotometers". The authors thank the Reviewer for the additional revision of the manuscript as well as for their suggestions for improving the clarity of the text and reply to all comments below.

The answer is structured as follows: (1) comments from Reviewer #1, (2) authors' response and (3) authors' change in the manuscript.

- (1) Line numbers refer to the version with tracked changes
- (1) 101: Replace the text in parentheses with "such as stray light and wavelength alignment". I suggest removing "wavelength accuracy" as it is related to alignment.
- (2, 3) Following the reviewer's comment, the term "wavelength accuracy" has been removed from the manuscript and the text in parentheses has been replaced with "such as stray light and wavelength alignment".
- (1) 119: Keep only the word "iris" and remove "diaphragm".
- (2, 3) Done.
- (1) 124: Replace "spectrometer" with "spectrometer's entrance slit".
- (2, 3) Done.
- (1) 155: Replace "at zenith" with "towards the UVB port"
- (2, 3) Done.
- (1) 212: I suggest rephrasing "uncertainty values have been given" to "typical uncertainty estimates have been assumed"
- (2, 3) Done.
- (1) 226: Replace "are" with "is"
- (2, 3) Done.
- (1) 275: Replace "in total" with "on average", since you say "by each Brewer"
- (2, 3) The term "in total" has been replaced with "on average".
- (1) 324: Replace "metalling" with "metal"
- (2, 3) Done.
- (1) 420: Replace "those Brewer" with
- (2, 3) The reviewer's sentence is incomplete. Maybe they were referring to the typo found in the term "those Brewer". Therefore, the previous term has been replaced with "those Brewers", in plural.

- (1) 455: I suggest removing this sentence because the state of the atmosphere has nothing to do with the performance of the instrument. The reason stated in the previous sentence for using only spectra at SZA

Figure 3. Relative combined standard uncertainties of all UV irradiances measured on 13 September 2023 at 305 nm. (a) First group (double Brewers with cosine correction). (b) Second group (two single and three double Brewers with no cosine correction implemented).

Figure 7. Relative standard relative uncertainty on 13 September 2023 caused by the cosine correction implemented. (a) Spectral dependency at 12:30 UTC. (b) SZA dependency at 305 nm.

(1) 6. SZA dependence of the uncertainty: The results in Figure 3b, indicating that the relative uncertainty does not vary with solar zenith angle (SZA), are highly suspicious. At least, measurement noise and wavelength misalignments, particularly at shorter wavelengths, should result in increased uncertainty at higher SZAs, as also acknowledged in lines 527–529. Stray light should also play a more significant role under these conditions, and the uncertainty associated with its correction is expected to increase, as stated in lines 569–570. In my opinion, the key limitation of Fig. 3 is that the choice of 335 nm does not reflect the most critical effects. This also leads to the possibly wrong conclusion that "For half of the Brewers studied, the relative uncertainty shows no spectral nor angular dependency", as the behaviour at shorter wavelengths could differ substantially.

- (2) Indeed, all the mentioned uncertainty sources cause the relative combined standard uncertainty to increase with SZA at shorter wavelengths. At larger wavelengths, their impact is reduced and the relative uncertainty levels off. Nevertheless, we agree with the reviewer that shorter wavelengths can be more interesting as they are important for ozone and biological studies.
- (3) Following the reviewer's comment, Figure 3 has been updated to show the relative combined standard uncertainty at 305 nm (see the previous response). The discussion regarding this figure (lines 486–490) has been changed to "Regarding the angular dependency, Figure 3 represents all the relative uncertainty values derived on 13 September at 305 nm. This wavelength was selected as its analysis is interesting for studying ozone variability and biological effects. Figure 3 shows that the relative combined standard uncertainty of the Brewers studied increases with SZA at short wavelengths. This increase is especially marked for single Brewers, due to the uncertainty in stray light correction". Furthermore, Section 6 has also been revised to remove any wrong conclusions. As a result, lines 700–705 have been modified to "For the Brewers studied, the relative uncertainty increases with SZA at short wavelengths. This behaviour is linked to wavelength shifts, noise, and the correction of stray light, cosine error, and dark counts".
- (1) 7. Clear-sky conditions: It should be clearly stated throughout the paper that this uncertainty assessment applies only under clear-sky conditions.
- (2, 3) The conditions in which the UV measurements were performed have been added in the introduction and conclusions section. They have also been discussed at the beginning of the methodology section and in greater detail in Section 3.1.3. Furthermore, they were also stated in the results section following the comments made by Reviewers #1 and #3 in the first round of revision.
- (1) 8. Section 5: This section still appears unnecessary and rather forced. Some of its content could be incorporated into the Introduction as the rationale of the study, while the remainder could be moved to the Conclusions. Removing this section entirely would improve the overall structure and flow of the paper.
- (2) Section 5 plays a key role in justifying the relevance of this work for publication in ACP, as it is necessary to highlight the importance of Brewer uncertainty evaluation. However, we agree with the reviewer that the section can be revised and enhanced to better align with the overall flow of the paper.
- (3) Following the reviewer's comment, Section 5 has been revised to improve its clarity and relevance, as follows:

"In the previous section, the combined standard uncertainty and the main sources of uncertainty of single and double Brewer spectrophotometers have been determined. These aspects are of great interest for identifying the type of studies for which Brewers are most suitable, given that the required uncertainty in global UV irradiance measurements depends on the intended use of the data.

One of the key applications of spectral UV measurements is the computation of effective irradiance for various biological effects, such as erythema, vitamin D synthesis, melanoma risk, and DNA damage, through the integration of the spectral irradiance weighted by different action spectra (Webb et al., 2011). The findings benefit regulatory applications, supporting evidence-based UV exposure limits for outdoors workers (Vecchia et al., 2007) and improving standards for sun protection products (Young et al., 2017). For these types of studies, the standard procedure is to use instruments with relative irradiance and erythemal uncertainties of less than 7 % in the UV-B region (e.g. Bilbao and de Migue,

2020; Cede et al., 2002; McKenzie et al., 1991). Therefore, double Brewer spectrophotometers are suitable for biological studies. However, single Brewers might be limited as their relative uncertainty rises rapidly with SZA (see Fig. 3), up to 30 % at 302 nm. This increase is mainly produced by the uncertainty in stray light correction (see Fig. 5 and Table 4), which indicates that the five-wavelength method implemented (described in Section 3.1.1) might need improvement.

Furthermore, spectral UV measurements are also used for the validation of satellite-based UV products from instruments such as OMI, TROPOMI, and TEMPO. In these validation studies, the standard uncertainty of the ground-based instruments used is of the order of 4–10 % (e.g. Klotz et al., 2025; Weish et al., 2008; Zempila et al., 2016). Consequently, the UV data from both MkIV and MkIII Brewer spectrophotometers are reliable for satellite-validation for wavelengths above 305 and 302 nm, respectively.

The more demanding application of spectral UV measurements is trend detection. It requires high-quality and long-term measurements (Bernhard, 2011; Glandorf et al., 2005; Weatherhead et al., 1998) as the trends are expected to be small. For example, between the 1990s and 2010s, long-term trends in UVI, global UV, and erythemal irradiance have typically been a few percent per decade, generally from 2 to 10 % per decade (Bernhard and Stierle, 2020; Bilbao et al., 2011; De Bock et al., 2014; Fitzka et al., 2012; Fountoulakis et al., 2016a, 2018). This indicates that double Brewers are able to detect the current changes in UV irradiance above 305 nm as long as they are properly maintained and their irradiances corrected. On the other hand, single Brewers can also reliably detect these changes at SZAs below 70° and wavelengths above 305 nm.

Finally, an even more ambitious goal is to detect the long-term change in spectral UV irradiance caused by a 1 % change in ozone (Seckmeyer et al., 2001). Bernhard and Seckemyer (1999) calculated that a 1 % change in ozone results in a 4 % change in global UV irradiance at 300 nm (30° SZA and 300 DU). Since the expanded uncertainty (k = 2) of the Brewers studied (single and double) ranges from 5.5 % to 7.8 % at 33° and 300 nm, the detection of such a trend might be beyond their capabilities. This holds even if the Brewer spectrophotometer is fully corrected and properly maintained, indicating a substantial reduction of its uncertainty is required (the recommendations for this task are outlined in Section 6). If the trend detection threshold were relaxed from a 1 % to a 3 % change in total ozone, then the instruments studied in the first group (cosine correction implemented) would be able to reliably detect the resulting change in spectral UV irradiance at 300 nm and SZAs below 70°. The remaining Brewers would be able to detect such change at 300 nm for SZAs below 60°".

- (1) 9. Emphasis on replacing traditional optics: The paper puts strong emphasis on replacing traditional optics (e.g., lines 32–33 and 723–724). However, I believe the primary message should be to first reprocess existing data, taking angular errors into account. Only after this step, should optical replacement be considered, in order to avoid introducing step changes in the long-term data records. Although this is briefly mentioned in lines 726–728, it is not sufficiently highlighted and does not come across as a key conclusion.
- (2) Brewer spectrophotometers with traditional optics have worse angular response while the new optics improve both the response and the overall uncertainty of the instrument. Nevertheless, we agree with the reviewer that such change should not be recommended without first studying the cosine response of the instrument. Moreover, replacing the entrance optics is not an easy task and it could require substantial development and changes in the instrument design (Bais et al., 2005).

(3) Following the reviewer's comment, lines 32–33 and 723–724 have been removed from the manuscript. Furthermore, the discussion carried out in lines 726–728 has been replaced with the following information "Regarding the angular response of the instrument, it should be studied and the existing data reprocessed with the corresponding cosine correction. This correction is necessary to ensure the reliability of the Brewer UV measurements as Brewer spectrophotometers with no cosine correction underestimate the spectral UV irradiance. Furthermore, this study is also essential to accurately determine the overall measurement uncertainty and to guarantee the instrument's suitability for detecting long-term UV trends".

**(1) TECHNICAL REMARKS**

- (1) Line 21: The phrase "due to the difficulties involved in the uncertainty propagation" is misleading. This study itself demonstrates that the main challenges are from the characterisation of the instrument, not from the propagation of uncertainty. This is also acknowledged at lines 75–76.
- (2, 3) Following the reviewer's comment, the phrase "due to difficulties involved in the uncertainty propagation" at line 21 has been replaced with "due to difficulties involved in characterising the instrument".
- (1) Line 30: I still believe that stray light is more closely related to the spectral shape than to the absolute intensity.
- (2, 3) Lines 30–31 have been modified to reflect the relationship between stray light and spectral shape as follows: "As the measured wavelength decreases, the correction of dark signal, stray light (for single Brewers), and noise become the dominant sources of uncertainty".
- (1) Line 51: The bibliographic entry for Webb et al., 2003 is missing and should be included in the reference list.
- (2, 3) The missing reference has been included in the references section.
- (1) Line 64: There is still no proper reference to protocols concerning UV calibration and measurements within EUBREWNET. Redondas et al., 2018 and Rimmer et al., 2018 focus mainly on ozone measurements. López-Solano et al., 2024 addresses the processing of UV irradiance but does not include protocols or recommendations for measurements or calibrations. The web page https://eubrewnet.aemet.es/dokuwiki/doku.php?id=codes:uvaccess refers instead to UV database access functions and not to formal protocols.
- (2) There are no protocols developed for UV or ozone in EuBrewNet. At the moment, only the processing algorithms are available. We are now aware that the manuscript may cause confusion and part of the text needs rephrasing.
- (3) The manuscript has been revised to remove any reference to "guidelines" or "protocols" regarding EuBrewNet. Now, only the processing algorithms are mentioned throughout the manuscript.
- (1) Line 115: Please rephrase as: "using a set of filters installed on two wheels".
- (2, 3) Done.
- (1) Line 118: Replace "filter wheels" with simply "filters".

- (2, 3) Done.
- (1) Line 147: It is unclear what calibration is being referred to here. For global irradiance measurements, the input optics is the UV diffuser, not the zenithal prism.
- (2, 3) The calibration mentioned is the one performed for global UV measurements. Following the reviewers' comment, line 147 has been modified to "with the input optics positioned towards the UVB port".
- (1) Lines 203–205 and Sections 3.1.1 / 3.1.3: The distinction between these two categories is not entirely clear. All the sources discussed (wavelength misalignments, temperature dependence, cosine error) affect the raw signal in some way. Perhaps the title of Section 3.1.3 could be revised or expanded to better reflect its specific focus and how it differs from Section 3.1.1.
- (2, 3) To further differentiate Sections 3.1.1. and 3.1.3. their titles have been modified to "Uncertainty in Brewer raw signal (counts)" and "Uncertainty in absolute irradiance", respectively.
- (1) Line 214: It would be helpful to specify that the authors are referring to dust particles inside the monochromator, rather than in the atmosphere.
- (2, 3) Noted. The term "inside the spectrometer" has been added in line 214 to specify that the dust particles are in the air within the spectrometer and not in the atmosphere.
- (1) Line 286 (" $500.0 \pm 0.6$  mm"): What is the difference between this value and the one reported in line 285?
- (2) The value in line 285 is 0.58 mm and the one in line 286 is 0.59 mm.
- (3) To avoid confusion, two decimals have been used to express the uncertainty in the distance adjustment. As a result, line 286 has been corrected to " $(500.00 \pm 0.59)$  mm" and line 285 to " $(500.00 \pm 0.58)$  mm"
- (1) Lines 329–332: The issue discussed in this paragraph appears to concern accuracy, not precision. Please reformulate accordingly to avoid confusion.
- (2) The issue discussed in the indicated lines is related to the smallest wavelength increment that the micrometre that rotates the grating can perform. Therefore, "precision" is not the best term to describe it, but "resolution".
- (3) The phrase "This precision" in line 330 has been replaced with "This resolution (smallest wavelength increment)". Furthermore, in line 329, the term "precision" has been replaced with "resolution", as well.
- (1) Line 336: This statement applies only to the temperature dependence in global UV measurements. Please make this limitation explicit.
- (2, 3) As the reviewer states, this limitation only affects global UV measurements as TOC observations do have a standard methodology for the temperature correction. The term "in global UV irradiance measurements" has been added in line 336.

- (1) Lines 412–413: The influence of this effect seems negligible compared to the other factors considered in the study. Since it is also somewhat off-topic, I suggest removing this sentence.
- (2, 3) Following the comments made by all reviewers, lines 412–413 have been removed from the manuscript.
- (1) Line 449: Remove the word "likely". These uncertainties are certainly highly underestimated.
- (2, 3) The word "likely" has been removed from the manuscript.
- (1) Line 460: Is the correct term here accuracy or precision?
- (2, 3) In the indicated context, accuracy is the correct term. Therefore, the word "precision" has been replaced with "accuracy".
- (1) Lines 462–463: Although it may be technically correct to state that uncertainty increases with increasing wavelength and decreasing SZA, this is highly misleading. The more relevant quantity is the relative uncertainty, which behaves oppositely. I strongly recommend rephrasing this, and the same applies to lines 695–696.
- (2, 3) Noted. To avoid confusion between absolute and relative uncertainty values, the indicated lines have been rephrased by adding the units of the absolute and relative combined standard uncertainty.
- (1) Line 508: Is the 10% "target" agreement based on a specific recommendation or official guideline? If so, please provide a reference.
- (2) The 10 % target is usually used in intercomparison campaigns to assess the agreement between instruments (e.g. Bais et al., 2001; Diémoz et al., 2011).
- (3) Following the reviewer's comment, references have been added in line 508 to provide context for the 10 % target used.
- (1) Line 530: Dead time does not appear to be a dominant factor for most of the instruments listed, perhaps only for #202.
- (2, 3) As the reviewer states, it is only important for Brewer #202. Therefore, the term "dead time correction" has been removed from line 530.

---

## Author Response (AR3)

**Response to Editor**

Authors' response to Editor comments on "Evaluation of the uncertainty of the spectral UV irradiance measured by double- and single-monochromator Brewer spectrophotometers". The authors thank the Editor for their insight as well as their careful and constructive examination of the manuscript and reply to all comments below.

The answer is structured as follows: (1) comments from Editor, (2) authors' response and (3) authors' change in the manuscript.

- (1) Dear authors, I think the paper has been improved after the two rounds of revisions. It would be more easy and more valid if a few very well characterized (including the characterization related uncertainties) have been investigated and instrument with additional issues (lack of characterization and also measurement issues) were not included. However this is the reality for various operational Brewers. In a way you can mention such things in the discussion and try to talk about the analysis of well characterized instruments and others that their measurement deviations from reference instrument could be expected larger for the additional above mentioned reasons.
- (2, 3) We agree with the Editor that the paper can be further improved. Consequently, a discussion has been added at the beginning of Section 4 (results) to explain why some of the studied Brewers have a cosine correction implemented, while others are lacking a characterisation of their angular response. Furthermore, Sections 4.1 and 4.2 have been modified to compare the performance and uncertainties of well characterised Brewers with those of instruments with a less complete characterisation. Moreover, the sensitivity analysis (Section 4.3) has been broadened to include remarks on the importance of maintenance and calibration, which can result in larger uncertainties if the frequency of such procedures is inadequate. Finally, the conclusions (Section 6) have also been modified to reflect the changes made in the analysis of the results.

**(1) Concerning the responses**

- (1) I would include a summary of the discussion you include in the response of comment 1 reviewer 3 (page 3 in the author responses document), in the revised document.
- (2, 3) A summary of the response given to comment 1 of Reviewer #3 has been added by modifying former lines 448–454 as follows:

"To present the results, the Brewers studied have been separated into two groups, depending on their degree of characterisation. Ideally, all instruments would be fully characterised. Unfortunately, this is not the case as there are no standard protocols for studying the cosine error and temperature dependence of the global UV irradiance measured by Brewer spectrophotometers. Consequently, most Brewers worldwide lack both temperature and angular characterisation. This situation is expected to improve as the Brewer community has recognised the importance of temperature and cosine errors and is making efforts to systematically correct these two sources (e.g. Lakkala et al., 2018).

Within this framework, the first group used in this work includes the five Brewers whose angular responses had been measured (#150, #158, #185, #186, and #256). The remaining five Brewers were gathered in a second set as their characterisation is less complete. Therefore, the second

group has two single (#117 and #151) and three double (#172, #202, and #228) Brewers. The uncertainty evaluation of the Brewer spectrophotometers in this second group is limited as the cosine correction, which is one of the key uncertainty sources in solar radiometry, is missing. As a result, the uncertainties determined are underestimated. Nevertheless, the inclusion of these Brewers is of interest as it allows both the comparison with fully corrected ones and the determination of uncertainties that, although underestimated, are representative for most Brewers worldwide".

- (1) Also for comment 4 (page 5 of the responses) I would discuss a bit more on the basis of my comments above and the more detailed comments of the reviewer 3 and I would add a paragraph discussing the uncertainty estimation of well characterized instruments operating according to certain protocols (e.g. calibration frequency, wavelength calibration accuracy, etc), compared with instruments that in addition to this uncertainty, could have additional problems that can be randomly low or high according to their additional operational issues.
- (2, 3) The sensitivity analysis has been revised to acknowledge the problematic uncertainty sources for those Brewers that do not follow strict calibration and maintenance protocols. Moreover, the text has been modified to highlight the need for better characterisation of certain uncertainty sources (especially noise) and for defining calibration protocols and their frequency (to reduce the uncertainty in wavelength shifts, radiometric stability, the uncertainty of the reference lamp, and the dead time).

On the other hand, a paragraph comparing instruments that follow calibration protocols with those that can have additional problems have been added in the conclusions section as follows: "Regarding the relative values (the absolute combined standard uncertainty divided by the UV irradiance measured) of all Brewers (single and double), it is instrument specific and depends on both the instrument individual characteristics as well as its calibration and maintenance protocols. Thus, a well-maintained Brewer can reach relative uncertainties of 2–3 % for wavelengths above 310 nm. On the other hand, instruments not following strict calibration and maintenance protocols can have large fluctuations in their uncertainties, due to operational issues. As a result, they show lager uncertainties, of 3–5 % for wavelengths above 310 nm, and higher deviations from the QASUME (the European reference unit)". Moreover, to include the concerns expressed by Reviewer #3 the following information has been added regarding wavelength shifts (in line 720, Section 6): "Thus, it is important to monitor the wavelength calibration of the instrument to keep wavelength shifts below 0.1 nm as this shift lead to uncertainties of 3 % in the UV-A region".